# One-Step is Enough: Sparse Autoencoders for Text-to-Image Diffusion Models

**Viacheslav Surkov**[*,1]     **Chris Wendler**[*,2]     **Antonio Mari**[1]     **Mikhail Terekhov**[1]
**Justin Deschenaux**[1]     **Robert West**[1]     **Caglar Gulcehre**[1]     **David Bau**[2]
[1]EPFL     [2]Northeastern University     [*]equal contribution

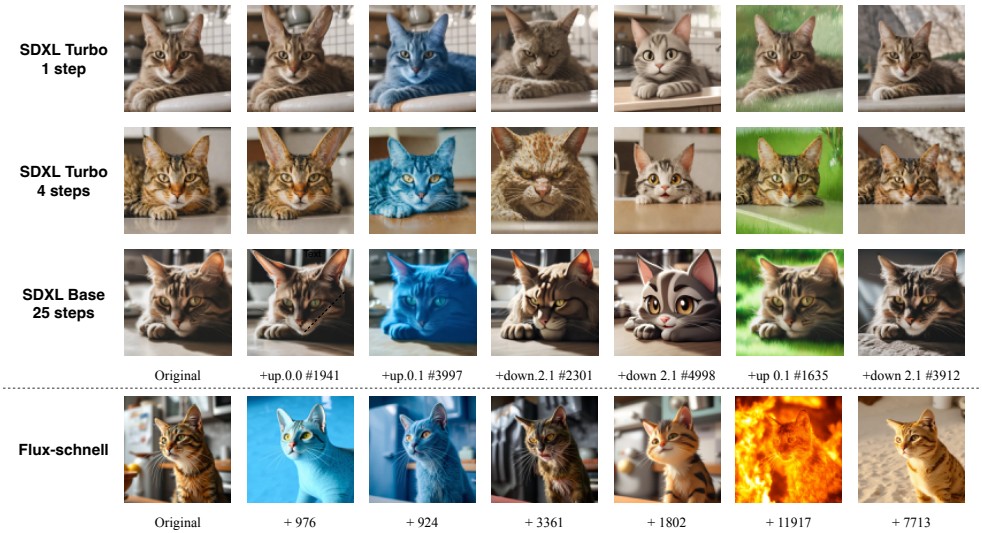

Figure 1: **Enabling features learned by sparse autoencoders results in interpretable changes** in SDXL Turbo's, SDXL's, and Flux-schnell's image generation processes. The image captions correspond to feature codes comprised of transformer block name and feature index.

## Abstract

For large language models (LLMs), sparse autoencoders (SAEs) have been shown to decompose intermediate representations that often are not interpretable directly into sparse sums of interpretable features, facilitating better control and subsequent analysis. However, similar analyses and approaches have been lacking for text-to-image models. We investigate the possibility of using SAEs to learn interpretable features for SDXL Turbo, a few-step text-to-image diffusion model. To this end, we train SAEs on the updates performed by transformer blocks within SDXL Turbo's denoising U-net in its 1-step setting. Interestingly, we find that they generalize to 4-step SDXL Turbo and even to the multi-step SDXL base model (i.e., a different model) without additional training. In addition, we show that their learned features are interpretable, causally influence the generation process, and reveal specialization among the blocks. We do so by creating RIEBench, a representation-based image editing benchmark, for editing images while they are generated by turning on and off individual SAE features. This allows us to track which transformer blocks' features are the most impactful depending on the edit category. Our work is the first investigation of SAEs for interpretability in text-to-image diffusion models and our results establish SAEs as a promising approach for understanding and manipulating the internal mechanisms of text-to-image models.

39th Conference on Neural Information Processing Systems (NeurIPS 2025).

# 1  Introduction

Text-to-image generation is a rapidly evolving field. The DALL-E model first captured public interest [46], combining learned visual vocabularies with sequence modeling to produce high-quality images based on user input prompts. Today's best text-to-image models are largely based on text-conditioned diffusion models [50, 51, 43, 52, 3, 42]. This can be partially attributed to the stable training dynamics of diffusion models, which makes them easier to scale than previous approaches such as generative adversarial neural networks [16]. As a result, they can be trained on internet scale image-text datasets like LAION-5B [53] and learn to generate photorealistic images from text.

However, the underlying logic of the neural networks that enable the text-to-image pipelines we have today, due to their black-box nature, is not well understood. Unfortunately, this lack of interpretability is typical in the deep learning field. For example, advances in image recognition [28] and language modeling [15, 6] come mainly from scaling models [21], rather than from an improved understanding of their internals. Recently, the emerging field of mechanistic interpretability has sought to alleviate this limitation by reverse engineering visual models [37] and transformer-based LLMs [45]. At the same time, diffusion models have remained underexplored.

This work focuses on SDXL Turbo, a recent open-source few-step text-to-image diffusion model. We import methods from a toolbox originally developed for language models, which allows inspection of the intermediate results of the forward pass [8, 20, 11, 5]. Moreover, some of these methods even enable reverse engineering of the entire task-specific subnets [35]. In particular, *sparse autoencoders (SAEs)* [62, 11, 5] are considered a breakthrough in interpretability for LLMs. They have been shown to decompose intermediate representations of the LLM forward pass – often difficult to interpret due to *polysemanticity* [1] – into sparse sums of interpretable and monosemantic features. These features are learned in an unsupervised way, can be automatically annotated using LLMs [7], and facilitate subsequent analysis, for example, circuit extraction [35].

## 1.1  Contributions

In this work, we investigate whether we can use SAEs to localize semantics to vector representations at specific layers of SDXL Turbo, a recent open-source few-step text-to-image diffusion model.

**SDLens.** To facilitate our analysis, we developed a library called *SDLens* that allows us to cache and manipulate intermediate results of SDXL Turbo's forward pass. We use our library to create a dataset of SDXL Turbo's intermediate feature maps of several transformer blocks inside SDXL Turbo's U-net on 1.5M LAION-COCO prompts [53, 54]. We then use these feature maps to train multiple SAEs for each transformer block. We open-source the code of SDLens. [2]

**SAEPaint.** To qualitatively assess the role and the impact of features learned at different transformer blocks in SDXL Turbo we create feature visualization techniques based on examples in which the features are highly active and on various interventions. To interactively perform interventions, e.g., adding a feature on a spatial region (at a specific transformer block) or subtracting it while generating an image, we build an app *SAEPaint* (Appendix A). Our qualitative analysis (Appendix G and Appendix H) suggests that the blocks specialize into a "composition," a "detail," and a "style" block. [3]

**RIEBench.** In order to quantify the strong causal effects that we qualitatively observe when interacting with SDXL Turbo and SDXL via SAEPaint, we create a new *representation-based image editing benchmark (RIEBench)*, for our setting of turning on and off features as we generate new images. We do so by leveraging the nine edit categories including prompts from PIEBench [26] and combining them with grounded SAM2 [49] to compute semantic segmentations, which allows for the selection of the constituting features. We find that SAE features allow for fine-grained transfer of visual features across different denoising processes matching neuron baselines while requiring multiple orders of magnitudes less features.

By analyzing the features selected with regard to the edit category we quantify the specialization of SDXL Turbo's transformer blocks that we also observe when interacting with SDXL through SAE-

---

[1] A phenomenon where a single neuron or feature encodes multiple, unrelated concepts [17].

[2] Our project page links to all mentioned repositories and resources `https://sdxl-unbox.epfl.ch`.

[3] Our "composition" and "style" blocks have been already known in the community [57].

Paint. In particular, the "detail" block was most effective for adding/deleting/changing objects/details and the "style" block for changing color/background and style.

**Generalization across steps and architectures.** Despite training on the one-step process we find that our learned features – without requiring additional training – generalize to the SDXL Turbo's four-step process and even vanilla SDXL's multi-step process. We find that one-step denoising is enough for training meaningful SAE features applicable to multi-step denoising. Our one-step SAEs not only retain much of their reconstruction quality but also their learned features retain their semantic meaning and causal influence on the multi-step generation (see Fig. 1 or Appendix E). We quantify this by analyzing reconstruction performance and feature overlap across multiple denoising steps.

Additionally, we train SAEs that can be used to manipulate the generative process of FLUX Schnell [30] (also Fig. 1). Again we observe that training on the one step generation process is enough. We consider this a crucial result because it demonstrates that meaningful, interpretable features can be extracted with significantly lower computational resources by training on the distilled model and then effectively deployed to understand and edit more powerful, multi-step models.

Thus, we show that SAEs learn interpretable features that causally the image generation process. By open-sourcing our library and SAEs, we lay the foundation for further research in this area.

## 2 Background

### 2.1 Sparse Autoencoders

Let $h(x) \in \mathbb{R}^d$ be an intermediate result of a forward pass of a neural network on input $x$. In a fully connected neural network, $h(x)$ could correspond to a vector of neuron activations. In transformers, which are neural network architectures that combine attention with fully connected layers and residual connections, $h(x)$ could either refer to the content of the residual stream after a layer, an update to the residual stream by a layer, or a vector of neuron activations within a fully connected layer.

It has been shown [62, 11, 5] that in many neural networks, especially LLMs, intermediate representations can be well approximated by sparse sums of $n_f \in \mathbb{N}$ learned feature vectors, i.e.,

$$h(x) \approx \sum_{\rho=1}^{n_f} s_\rho(x) \mathbf{f}_\rho, \tag{1}$$

where $s_\rho(x)$ are the input-dependent coefficients, most of which are equal to zero and $\mathbf{f}_1, \ldots, \mathbf{f}_{n_f} \in \mathbb{R}^d$ is a learned dictionary of feature vectors. Importantly, the features are usually interpretable.

**Sparse autoencoders.** To implement the sparse decomposition from equation 1, the vector $s$ containing the $n_f$ coefficients of the sparse sum, is parameterized by a single linear layer followed by an activation function, called the *encoder*,

$$s = \text{ENC}(h) = \sigma(W^{\text{ENC}}(h - b_{\text{pre}}) + b_{\text{act}}), \tag{2}$$

in which $h \in \mathbb{R}^d$ is the latent that we aim to decompose, $\sigma(\cdot)$ is an activation function, $W^{\text{ENC}} \in \mathbb{R}^{n_f \times d}$ is a learnable weight matrix and $b_{\text{pre}}$ and $b_{\text{act}}$ are learnable bias terms. We omitted the dependencies $h = h(x)$ and $s = s(h)$, which are clear from the context.

Similarly, the learnable features are parametrized by a single linear layer called *decoder*,

$$h' = \text{DEC}(s) = W^{\text{DEC}}s + b_{\text{pre}}, \tag{3}$$

in which $W^{\text{DEC}} = (\mathbf{f}_1|\cdots|\mathbf{f}_{n_f}) \in \mathbb{R}^{d \times n_f}$ is a learnable matrix. Its columns take the role of learnable features and $b_{\text{pre}}$ is a learnable bias term. [4]

### 2.2 Few-Step Diffusion Models: SDXL Turbo

SDXL Turbo [52] is a distilled version of Stable Diffusion XL [43], a powerful latent diffusion model. SDXL Turbo allows high-quality sampling in as few as 1-4 steps. It employs a denoising network implemented using a U-net similar to [50]. For an architecture diagram consult Appendix D Fig. 10.

---

[4]An extended version of this section, including training details, is in Appendix K and Appendix L.

The U-net consists of a down-sampling path, a bottleneck, and an up-sampling path, each comprising one or more U-net blocks connected via up- and down-samplers. U-net blocks are built from residual networks, and some blocks incorporate multiple cross-attention transformer blocks while others do not. We refer to these transformer blocks by their short names (e.g., down.2.1). Each transformer block is composed of several basic transformer layers, including self-attention, cross-attention, and MLP layers. Importantly, the text conditioning is achieved via cross-attention to text embeddings performed by 11 transformer blocks embedded in the down-, up-sampling paths, and the bottleneck. An architecture diagram displaying the relevant blocks can be found in additional material.

## 3 Sparse Autoencoders for SDXL Turbo

With the necessary definitions at hand, in this section we show a way to apply SAEs to SDXL Turbo.

**Where to apply the SAEs.** Since we want to localize blocks' responsibilities, we apply SAEs to the updates performed by the transformer blocks containing the cross-attention layers responsible for incorporating the text prompt (for details consider Appendix D). Each of these blocks consists of multiple transformer layers, which attend to all spatial locations (self-attention) and to the text prompt embeddings (cross-attention).

Formally, the $\ell$th cross-attention transformer block updates its inputs in the following way

$$D[\ell]_{ij}^{out} = D[\ell]_{ij}^{in} + \mathcal{T}[\ell](D[\ell]^{in}, c)_{ij}, \tag{4}$$

in which $D[\ell]^{in}, D[\ell]^{out} \in \mathbb{R}^{h \times w \times d}$ denote the residual stream before and after application of the $\ell$-th cross-attention transformer block respectively. The transformer block itself calculates the function $\mathcal{T}[\ell] : \mathbb{R}^{h \times w \times d} \to \mathbb{R}^{h \times w \times d}$. Note that we omitted the dependence on input noise $z_t$ and text embedding $c$ for both $D[\ell]^{in}(z_t, c)$ and $D[\ell]^{out}(z_t, c)$.

We train SAEs on the residual updates $\mathcal{T}[\ell](D[\ell]^{in}, c)_{ij} \in \mathbb{R}^d$ denoted by

$$\Delta D[\ell]_{ij} := \mathcal{T}[\ell](D[\ell]^{in}, c)_{ij} = D[\ell]_{ij}^{out} - D[\ell]_{ij}^{in}. \tag{5}$$

That is, we jointly train one encoder $\text{ENC}[\ell]$ and decoder $\text{DEC}[\ell]$ pair per transformer block $\ell$ and share it over all spatial locations $i, j$. We do this for the 4 (out of 11) transformer blocks[5] that we found have the highest impact on the generation, namely, down.2.1, mid.0, up.0.0 and up.0.1.

For the sake of simplicity, we omit the transformer block index $\ell$ in the remainder of the paper.

**Feature maps.** We refer to $\Delta D \in \mathbb{R}^{h \times w \times d}$ as dense feature map and applying $\text{ENC}$ to all image locations results in the *sparse feature map* $S \in \mathbb{R}^{h \times w \times n_f}$ with entries $S_{ij} = \text{ENC}(\Delta D_{ij})$.

We refer to the feature map of the $\rho$th learned feature using $S^\rho \in \mathbb{R}^{h \times w}$. This feature map $S^\rho$ contains the spatial activations of the $\rho$th learned feature. Its associated feature vector $\mathbf{f}_\rho \in \mathbb{R}^d$ is a column in the decoder matrix $W^{\text{DEC}} = (\mathbf{f}_1 | \cdots | \mathbf{f}_{n_f}) \in \mathbb{R}^{d \times n_f}$. Using this notation, we can represent each element of the dense feature map as a sparse sum

$$\Delta D_{ij} \approx \sum_{\rho=1}^{n_f} S_{ij}^\rho \mathbf{f}_\rho, \text{ with } S_{ij}^\rho = 0 \text{ for most } \rho \in \{1, \ldots, n_f\}. \tag{6}$$

**Training.** In order to train an SAE for a transformer block, we collected dense feature maps $\Delta D_{ij}$ from SDXL Turbo one-step generations on 1.5M prompts from the LAION-COCO [54]. Each feature map has dimensions of $16 \times 16$, resulting in a training dataset of 384M dense feature vectors per transformer block. For the SAE training process, we followed the methodology described in [19], using the TopK activation function and an auxiliary loss to handle dead features. For more details on the SAE training and for training metrics, consider the supplementary material. In order to find good *hyperparameters*, we perform a line search for an optimal sparsity parameter $k$ while keeping $n_f$ fixed to 5120 (expansion factor 4) and a line search for for $n_f$ where we keep $k$ fixed at 10. As can be seen in Fig. 2, the explained variance strictly increases when increasing $k$ and a similar trend holds for the expansion factor, proportional to the number of features $n_f$.

---

[5]We provide an architecture diagram in Appendix D Fig. 10.

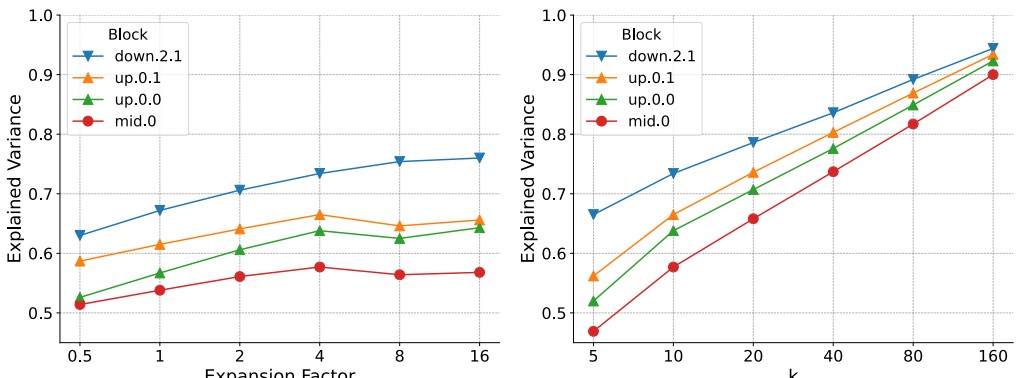

Figure 2: **The explained variance increases with both the expansion factor and $k$.** The left shows the dynamics of explained variance for SAEs with varying expansion factors and a fixed $k = 10$, while the right panel illustrates the same for varying $k$ values with a fixed expansion factor of 4.

**Generalization study.** We trained the SAEs on the 1-step generation process of SDXL Turbo. However, most diffusion models (including SDXL Turbo) require multiple denoising steps to create high quality images. To assess our SAEs' potential for application across multiple denoising steps, we compute their explained variance across the 100 randomly generated images for SDXL Turbo's 4-step setting and the vanilla SDXL base model's multi-step setting (see Fig. 3 left).[6] One can see that the explained variance remains high across the entire denoising process, which suggests that our SAEs are also applicable for the 4-step generation process and even the base model's 20-step process.

This suggests that our feature intervention (formal definition below) should causally influence the generated image in multi-step diffusion in the same way as it does in the 1-step process where it was trained. This is visualized in the first three rows of Fig. 1: for example, adding the 2301st feature vector to the forward pass consistently makes the output look more "evil." The 4th row of Fig. 1 shows steering examples for Flux-schnell [30] as well, using features obtained with an SAE trained on 1-step activations of layer 18. This shows the generalizability of our approach to recent diffusion-transformers [41]. Additional examples of FLUX interventions are presented in Appendix C. Further examples of vanilla SDXL interventions and the effect of intervening only on subsets of the denoising steps are shown in Appendix E.

On the right of Fig. 3 we visualize how the SAE features change across denoising steps. We do so by computing the cosine similarity between the SAE coefficients of adjacent timesteps. Interestingly, the set of features stabilizes rapidly and subsequent denoising steps' features stay highly overlapping for most of the denoising process. *We therefore proceed by using SDXL Turbo's 4-step process from now.*

**Feature interventions.** We design interventions to turn on and off the $\rho$-th feature. Specifically, we achieve this by adding or subtracting it across spatial locations $i, j$ weighted by $A \in \mathbb{R}^{h \times w}$.

$$\Delta D'_{ij} = \Delta D_{ij} + A_{ij} \mathbf{f}_\rho, \tag{7}$$

in which $\Delta D_{ij}$ is the update performed by the transformer block before the intervention and $\Delta D'_{ij}$— after the intervention, and $\mathbf{f}_\rho$ is the $\rho$-th learned feature vector. Our examples in Fig. 1 are obtained by drawing binary masks in SAEPaint and multiplying them with an scalar specifying the intervention strength, i.e. $A = \alpha M$ in which $\alpha \in \mathbb{R}$ and $M \in \{0, 1\}^{h \times w}$.

Note that naive application of equation 7 is sufficient when using SDXL Turbo with default settings, that is *without classifier-free guidance*. However, with classifier-free guidance (which is enabled in the SDXL base model), we add the features only during the text-conditioned forward pass and subtract the same features during the unconditional one. When adding the same features during both forward passes, the features' effects inhibit each other (see Appendix B Fig. 7).

## 4  Analysis of SDXL Turbo

In this section we leverage our SAEs to better understand how SDXL Turbo generates images.

---

[6]This results in 25600 latents for SDXL Turbo and 102400 latents for SDXL.

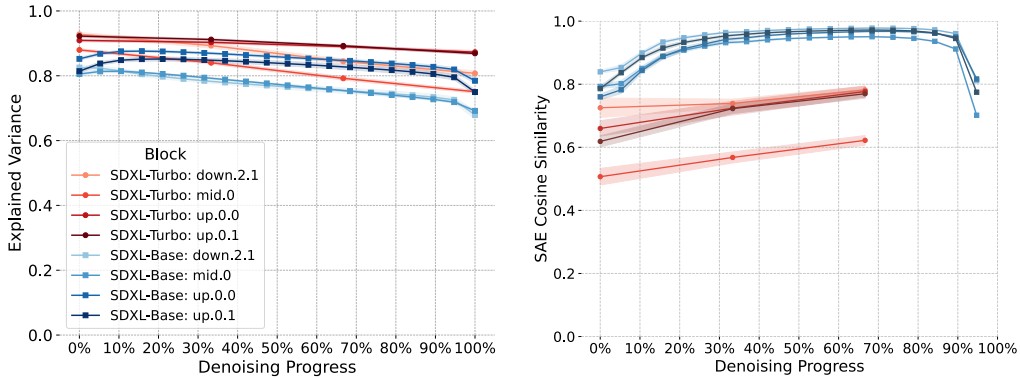

Figure 3: For our best SAEs $k = 160$, $n_f = 5120$, we compute how their explained variance changes across denoising steps in the multi-step setting (for which they were not specifically trained) on the left. On the right, we compute SAE feature overlap by computing cosine similarities of the sparse $n_f$ dimensional SAE feature vectors. **The SAE feature overlap is remarkably high across timesteps**. The plot ends early because we compute overlaps between $t$ and $t - 1$ respectively.

## 4.1 RIEBench: Representation-Based Image Editing Benchmark

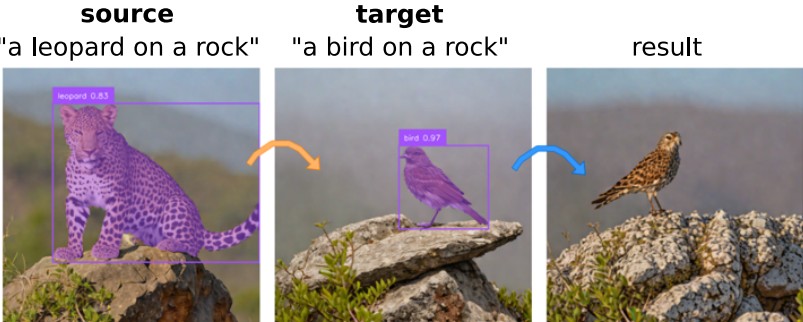

Figure 4: **We perform a feature transfer experiment**, computing semantic image segmentation masks using grounded SAM2 [49] for a source and a target image. We collect features inside the mask of the source image at SDXL Turbo's intermediate layers and insert them (via addition) during a new forward pass with the same prompt as the target one. We also subtract features collected in the target forward pass. The resulting image is a blend of the two forward passes.

In order to quantify the causal impact of SAE features on SDXL Turbo's generation, we introduce *RIEBench*, a representation-based image editing benchmark. To generate RIEBench samples, we first create two parallel forward passes (source and target forward pass) that receive the same random input noise and mostly parallel prompts except that they are differing in one visual aspect. Next, we transfer features from the source forward pass into the target forward pass. E.g., in Fig. 4 the prompts are "a *leopard* standing on top of a rock" and "a *bird* standing on top of a rock." In order to do so, we select spatial locations using grounded SAM2 [49] masks, obtained by prompting grounded SAM2 with "leopard" and "bird." Finally, we select a subset of the features that fall inside these masks in our four blocks and create a new forward pass with the target prompt in which we subtract some of the target features and add some of the source features. This results in the new image on the right that depicts visual features of both the leopard and the bird.

We sample our source and target prompts from PIEBench [26], a prompt-based image editing benchmark for diffusion inversion methods. PIEBench contains the "original prompts" (target of the transfer) and the "edit prompts" (source of the transfer) for 10 different edit categories. We omit the examples from the *random* category and implement specialized interventions for the other edit categories (Appendix F). For each edit category, we manually create grounded SAM2 prompts for the first 50 examples. For each of these examples, we fix a random seed, generate images for source

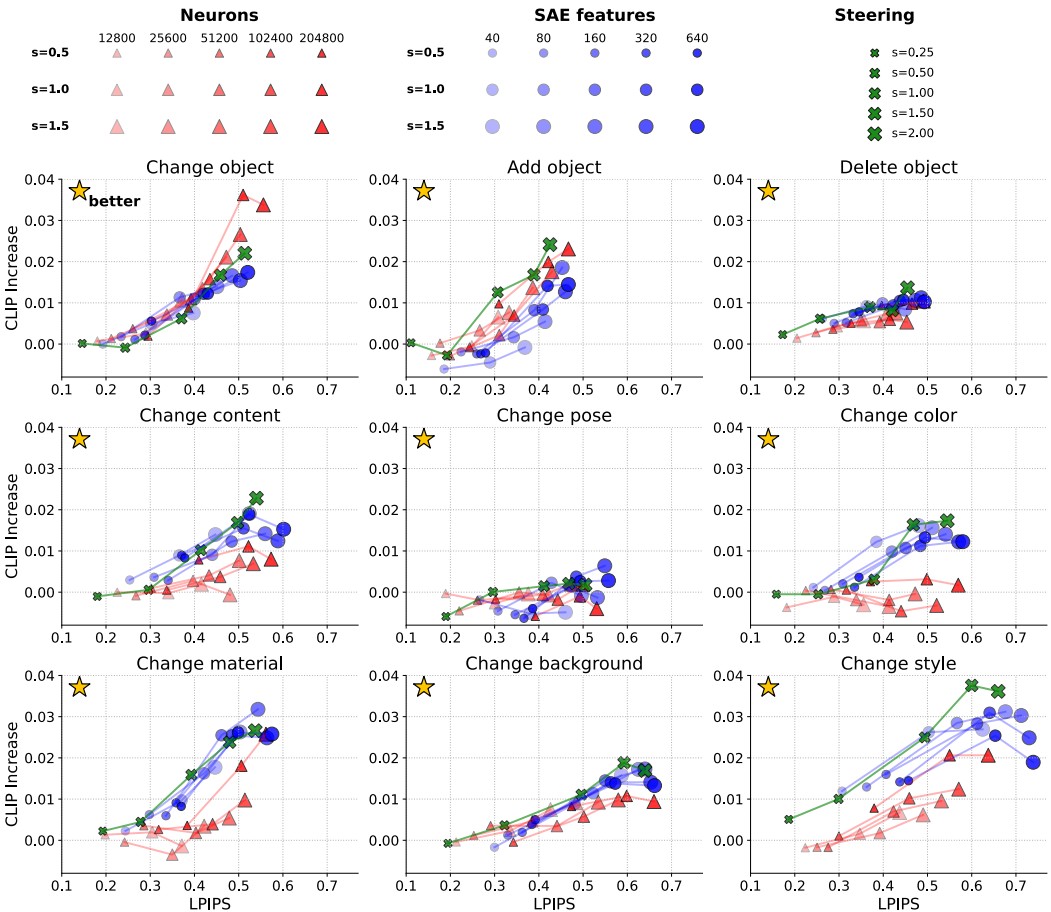

Figure 5: **LPIPS with original image (x-axis) versus increase in CLIP similarity with the edit prompt (y-axis) from feature-transfer interventions using SAE features, neurons, and activations across nine edit categories from PIEBench** [26]. Features from the edit prompt (within SAM2 segmentation mask) are transferred into the original prompt's forward pass. SAE experiments are blue, neurons red, and steering green. We plotted one line per number of features/neurons transported with increasing opacity proportional to the number. For SAEs we transport 40, 80, 160, 320, 640 features at a time and for neurons 12,800 25,600 51,200 102,400 204,800. For SAEs/neurons we use strengths 0.5, 1.0, 1.5 and for steering 0.25, 0.5, 1.0, 1.5, 2.0.

and target prompt, and manually inspect whether all relevant visual features are present and whether they are selected by the grounded SAM2 masks and only those.[7]

**Metrics.** From an interpretability standpoint, our benchmark jointly measures features' sensitivity, specificity, and causality. Visual features of the source forward pass can only be detected if their corresponding representations (e.g., SAE features) are sensitive. On the other hand, editing the generated image to be closer to the source image via interpretable visual changes requires both specificity and causality of the features. When a feature is active during the generation, it should ideally result in a corresponding interpretable visual feature and minimally interfere with the remaining ones. We quantify this aspect by transferring varying numbers of features between two parallel forward passes and measuring the LPIPS [64] distance between the intervened image and the original image and the increase in CLIP [44] similarity of the resulting intervened image with the edit prompt. Ideally, by transporting a varying number of features and increasing/decreasing their strength, it should be possible to smoothly interpolate between the relevant aspects of the source and the target image.

---

[7]After filtering, the edit categories and the corresponding numbers of selected examples were: change object (36), add object (27), delete object (43), change content (21), change pose (20), change color (30), change material (26), change background (46), change style (47).

**Methods.** We compare our best SAEs ($k = 160$ and $n_f = 5120$) with neurons of their corresponding transformer blocks. Since there are 10 MLP layers within each block that we decompose, there are 51200 neurons per block in total. Additionally, we create a simple method similar to activation steering [39], which adds the $\Delta D$ within the mask from the source forward pass to the target forward pass and subtracts the $\Delta D$ within the target forward pass' mask from the target forward pass. To mimic interventions that can be performed in SAEPaint we aggregate feature values/neurons/activations over spatial locations before adding them to masked area. Furthermore, all of our interventions are timestep step specific, i.e., the updates performed to the masked areas vary across denoising steps.

**Feature selection.** For our feature selection step, we first aggregate the sparse feature maps by taking the mean over timesteps. Let $S[\ell]^{\rho,t} \in \mathbb{R}^{h \times w}$ denote a sparse feature map at layer $\ell$ and timestep $t$, we use $\bar{S}[\ell]^\rho = \frac{1}{T} \sum_{t=1}^{T} S[\ell]^{\rho,t}$. Then to determine which features to transport, we rank each feature $\rho$ according to a score $\gamma[\ell]_\rho$ that quantifies the difference in the magnitude of $\rho$ across the source and target, aggregated over the respective grounded SAM2 masks and normalized by the sum of all feature magnitudes (c.f. Cywiński and Deja [12]):

$$\gamma[\ell]_\rho = \frac{s[\ell]^{\text{src}}_\rho}{\sum_{\rho'=1}^{n_f} s[\ell]^{\text{src}}_{\rho'}} - \frac{s[\ell]^{\text{tgt}}_\rho}{\sum_{\rho'=1}^{n_f} s[\ell]^{\text{tgt}}_{\rho'}}, \tag{8}$$

where $s[\ell]^{\text{src}}_\rho = \frac{1}{|M^{\text{src}}|} \sum_{i,j \in M^{\text{src}}} \bar{S}[\ell]^\rho_{ij}$ and $s[\ell]^{\text{tgt}}_\rho = \frac{1}{|M^{\text{tgt}}|} \sum_{i,j \in M^{\text{tgt}}} \bar{S}[\ell]^\rho_{ij}$. To select features among multiple layers we simply concatenate the layer specific score vectors $\gamma[\ell] \in \mathbb{R}^{n_f}$, i.e., $\gamma = \gamma[\text{down.2.1}] \cdot \gamma[\text{mid.0}] \cdot \gamma[\text{up.0.0}] \cdot \gamma[\text{up.0.1}] \in \mathbb{R}^{4n_f}$, where $\cdot$ denotes concatenation, before ranking.

For the neurons, we perform a similar ranking (see Appendix F for details).

**Results.** We show the results in Fig. 5. Quantitatively, in most categories there are no significant differences between the considered intervention types. Given an LPIPS budget, they all achieve similar increases in CLIP similarity (top left is better) in all categories except *change object* in which neurons dominate for higher LPIPS values, and, *change content, change color, change material and change style* in which both steering and SAEs are significantly more effective than neurons. While neurons also cover a good range of LPIPS distances / CLIP similarities, SAEs come with the benefit of requiring several orders of magnitude fewer features to do so. Interestingly, the simple steering baseline that works directly with the activations is highly effective as well. It's worth noting that when sufficiently increasing the number of transported features the SAE intervention approaches the steering intervention up to the reconstruction error introduced by the SAE. On the *change pose* task none of the considered methods works well. The change pose tasks is hard to achieve using our current RIEBench setup that in essence consists of adding constant update vectors to a masked areas.

*We also provide qualitative examples and the same evaluation for FLUX Schnell in Appendix F.*

### 4.2 Specialization Among Blocks

Thanks to our automated feature extraction with grounded SAM2 and our importance ranking, we can investigate which transformer blocks were relevant in terms of the number of features selected from that block depending on the task (see Fig. 6). It stands out that `mid.0` does not have a high causal impact on the generation. Similarly, `down.2.1` does not seem to get selected much for the edits considered. This is surprising given its strong impact in our qualitative experiments using our app (e.g., Fig. 1 #2301, #4998, #3912). The `up.0.1` block is most relevant for changing color, material, background, and style editing categories. For the other categories, `up.0.0` is the most impactful. We also observed qualitatively that `up.0.0` is good for editing local details as long as the context is relevant (e.g., Fig. 1 #1941), which we think happens in this setting due to the paired prompts. We have an extended analysis of the roles of the blocks in Appendix G and Appendix I.

## 5 Related Work

**Layer specialization in diffusion models.** Similar to our findings on the roles of SDXL Turbo's transformer blocks, [59, 1, 65] observe specializations among the layers and denoising steps of text-to-image diffusion models as well. Voynov et al. [59] introduce layer-specific embeddings for the text conditioning and find that different sets of layers are more effective for influencing the generation of specific concepts such as styles or objects. For a selected set of image attributes (e.g., color, object,

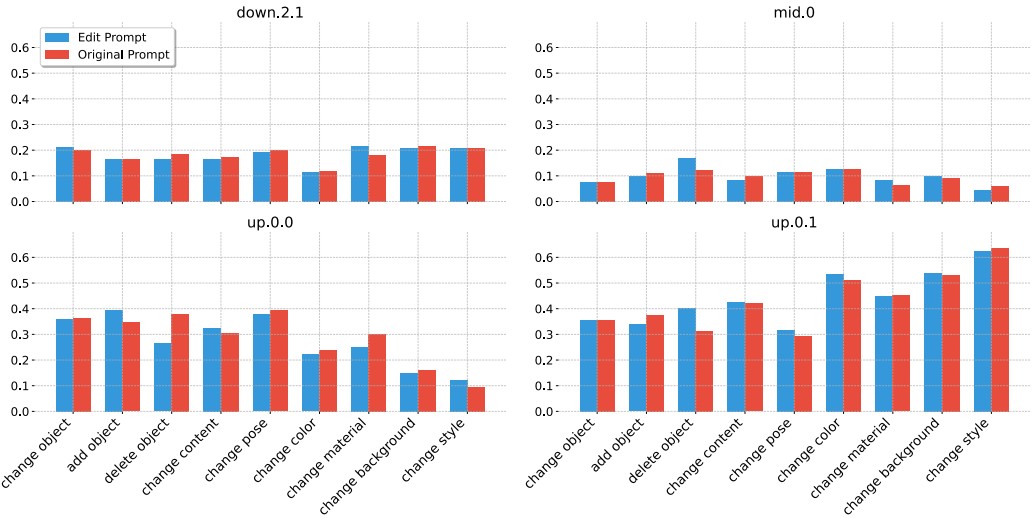

Figure 6: We count how often a feature of a block is selected for each task category.

layout, style), Agarwal et al. [1] analyze which attributes are captured in which timestamp and layer. They also find that a subset of these attributes is often captured in the same layer and across the same denoising step. Zhang et al. [65] observe that during the denoising process of diffusion models first the layout forms (in early timestamps), then the content, and finally the material and style.

In contrast to these prior works, our approach takes a fundamentally different perspective and does not rely on either handcrafted attributes or prompts. Instead, we introduce a new lens for analyzing SDXL Turbo's transformer blocks, which reveals specialization among the blocks as well. Interestingly, our findings on SDXL Turbo, which is a distilled, few-step diffusion model parallel [65]'s observations, which identify that composition precede material and style. In SDXL Turbo, this progression occurs across the layers instead of the denoising timestamps.

**Analyzing the latent space of diffusion models.** Kwon et al. [29] show that diffusion models have a semantically meaningful latent space. Park et al. [40] analyze the latent space of diffusion models using Riemannian geometry. Li et al. [31] and Dalva and Yanardag [13] present self-supervised methods for finding semantic directions. Similarly, Gandikota et al. [18] show that the attribute variations lie in a low-rank space by learning LoRA adapters [23] on top of pre-trained diffusion models. Brack et al. [4] and Wang et al. [60] demonstrate effective semantic vector algebraic operations in the latent space of DMs, as observed by Mikolov et al. [36]. However, none of those works train SAEs to interpret and control the latent space.

**Mechanistic interpretability.** Sparse autoencoders have recently been popularized by [5], in which they show that it is possible to learn interpretable features by decomposing neuron activations in MLPs in 2-layer transformer language models. At the same time, a parallel work decomposed the elements of the residual stream [11], which followed up on [55]. To our knowledge, the first work that applied sparse autoencoders to transformer-based LLM was [62], which learned a joint dictionary for features of all layers. Recently, sparse autoencoders have gained much traction, and many have been trained even on state-of-the-art LLMs [19, 58, 32]. In addition, great tools are available for inspection [33] and automatic interpretation [7] of learned features. [35] have shown how to use SAE features to facilitate automatic circuit discovery.

The studies most closely related to our work are [2], [24] and [14]. Ismail et al. [24] apply concept bottleneck methods [27] that decompose latent concepts into vectors of interpretable concepts to generative image models, including diffusion models. Unlike the SAEs that we train, this method requires labeled concept data. Daujotas [14] decomposes CLIP [44, 9] vision embeddings using SAEs and use them for conditional image generation with a diffusion model called Kandinsky [47]. Importantly, using SAE features, they are able to manipulate the image generation process in interpretable ways. In contrast, in our work, we train SAEs on intermediate representations of the forward pass of SDXL Turbo. Consequently, we can interpret and manipulate SDXL Turbo's forward pass on a finer granularity, e.g., by intervening on specific transformer blocks and spatial positions.

Another closely related work to ours is [2], in which neurons in generative adversarial neural networks are interpreted and manipulated. The interventions in [2] are similar to ours, but on neurons instead of sparse features. To identify neurons for a semantic concept, [2] require semantic segmentation maps.

# 6   Conclusion and Discussion

We trained SAEs on SDXL Turbo's opaque intermediate representations. This study is the first in the academic literature to mechanistically interpret the intermediate representations of a modern text-to-image model. Our findings demonstrate that SAEs can extract interpretable features and have a significant causal effect on the generated images. Importantly, the learned features provide insights into SDXL Turbo's forward pass, revealing that transformer blocks fulfill specific and varying roles in the generation process. In particular, our results clarify the functions of `down.2.1`, `up.0.0`, and `up.0.1`. However, the role of `mid.0` remains less defined; it seems to encode more abstract information and interventions are less effective.

We follow up with a discussion of the results and their implications for future research. Based on our observations, we suggest a preliminary hypothesis about SDXL Turbo's generation process: `down.2.1` decides on top-level composition, `mid.0` encodes low-level semantics, `up.0.0` adds details based on the two above, and `up.0.1` fills in color, texture, and style.

Our work focuses on analyzing SDXL Turbo's intermediate representations. As a relatively compact, few-step diffusion model with a small number of naturally partitioned components, SDXL Turbo turned out to be convenient to analyze with SAEs. However, the application of the proposed techniques to larger and more complex text-to-image diffusion models with alternative architectures represents a promising direction for further research. We provide some motivational results on Flux-schnell [30] in the supplementary material. Additionally, we observe that SAE features learned on SDXL Turbo's one-step generation are applicable to 4-step and to vanilla SDXL multi-steps generations. We observe that for FLUX training on one step is enough as well.

This fact suggests that our SDXL SAEs are also applicable to most of the 8,700 SDXL adaptations available on huggingface[8]. The same should hold for our FLUX SAEs, which should generalize to most of the 36,148 FLUX adaptations of on huggingface. This type of generalization is similar to what has been observed in the classic model merging literature [61], where averaging the weights of finetunes of the same base model has been found to improve model performance. Recently, this compatibility between different finetunes of the same base model has been rediscovered and expanded in weight-space learning, where probes that take model weights as inputs are trainable and generalize within a collection of finetunes of the same base model (but not across different seeds) [22].

**Future directions.** Analyzing larger diffusion models with higher number of diffusion steps would benefit from advanced interpretability techniques capable of capturing connections between its components and across denoising steps. For example, such techniques are explored in [35, 34]. Marks et al. [35] compute circuits showing how different layers and attention heads wire together and Lindsey et al. [34] introduce cross-coders, a variation of SAEs that allows to learn a shared set of features over latents corresponding to different layers.

**Broader impact.** This work explores the applicability of SAEs on text-to-image models to encourages disentangled and human-interpretable feature representations. As such models become increasingly powerful and widely adopted for image synthesis, they may be misused for generating deceptive content (e.g., deepfakes). Thus, understanding their internal representations is critical and improving interpretability can help mitigate such misuse by enabling researchers to detect manipulation or unintended behaviors within these models. Overall, our work advances the development of interpretable, responsible, and human-centered AI systems.

**Limitations.** Our current method focuses on individual blocks and may miss features that require combinations of multiple transformer blocks. The success rate and disentanglement quality of edits varies depending on the feature type and target image content. While we demonstrate local changes effectively (like skin color modifications), the identification and control of more global and compositional features remains challenging. We focused on SDXL and FLUX, thus, generalization to other models requires further validation. The hyperparameter choices for SAE training impact the types and granularity of discovered features, requiring tuning.

---

[8]Based on stabilityai/stable-diffusion-xl-base-1.0 and black-forest-labs/FLUX.1-dev as of 7th of October 2025.

## Acknowledgements

The authors thank Danila Zubko for the initial contribution, Peter Baylies, Rohit Gandikota, Rudy Gilman, Bartosz Cywiński, Julian Minder, Imke Grabe, Niv Cohen, Gytis Daujotas, Tim R. Davidson and Alexander Sharipov for the valuable discussions and feedback. We also express our gratitude to EPFL RCP and IC clusters maintainers. Robert West's lab is partly supported by grants from the Swiss National Science Foundation (200021_185043, TMSGI2_211379), Swiss Data Science Center (P22_08), H2020 (952215), Microsoft, and Google. Caglar Gulcehre's lab is supported by nimble.ai. Chris Wendler is supported by NSF grant #2403304.

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

# A SAEPaint: Our Feature Editor Application

We host our SAE-based feature editing app for you to try adding and subtracting SAE features during a forward pass. To find interesting features to try, you can read the rest of the supplementary material or you can try some of the ones of the ones we list on the app landing page. Alternatively, what works best if you want to explore the features yourself is to generate images for which you believe your features of interest should be on and have a look at their activation masks in the "Generate" tab. For

**SDXL Turbo App:** `https://huggingface.co/spaces/surokpro2/Unboxing_SDXL_with_SAEs`

**SDXL Base App:** `https://huggingface.co/spaces/surokpro2/sdxl-sae-multistep`

**Flux App:** `https://huggingface.co/spaces/surokpro2/sae_flux`

Notice that while we trained on SDXL Turbo 1-step mode the same features without additional training also work for 4 steps and even for the base model with, e.g., 25 steps.

# B SDXL Base

SDXL's default setting leverages classifier-free guidance to condition the generated image on a text prompt. We found that in this setting turning on SAE features works best, when adding them to the text-conditioned forward pass and subtracting them from the unconditional forward pass, see Fig. 7.

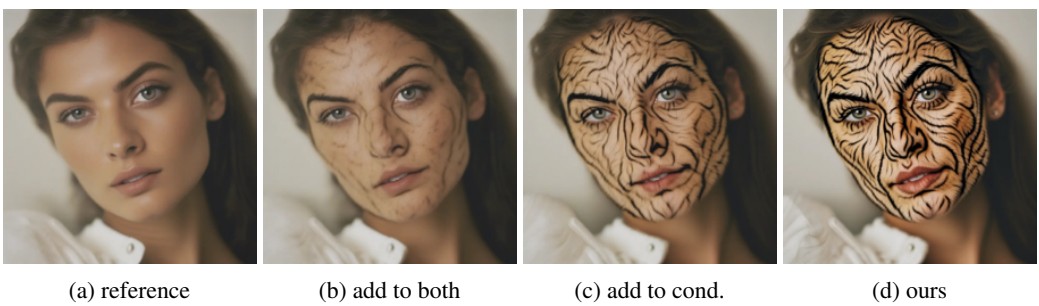

| (a) reference | (b) add to both | (c) add to cond. | (d) ours |

Figure 7: When using SDXL with classifier-free guidance and adding the #4977 "tiger texture" feature naively during both the conditional and unconditional forward pass its effects inhibit each other, see (b). In (b) we used twice the intervention strength as in (c) and (d), yet the tiger texture corresponding to feature #4977 is not visible. In (c) we add it only to the conditional forward pass, which works. In (d) we add the feature to the conditional forward pass and subtract it from the unconditional one, which we found works best.

# C Flux

**Training settings** We train a SAE on layer 18 activations of Flux-schnell 1-step. We choose layer 18 because we empirically find that its activations have higher norms than other layers. Additionally, all other exploratory experiments that we tried on FLUX, e.g., ablating layers, patching activations, simple activation steering, all consistently showed that layer 18 is a high impact layer. To train the SAE, we sample 1 million prompts from LAION-5B [53] and input them to Flux-schnell, then we randomly sample 10% of the activations (output - input) in the image stream of layer 18 (so for each prompt we get $\lceil 64 \times 64 \times 0.1 \rceil$ 3072-dimensional vectors).

The trained SAE has an expansion factor of 4 (thus its hidden dimension is 12288) and $k = 20$. All other hyperparameters and the training loss are detailed in K.

**Features injection** Features learned on Flux-schnell 1-step can be used on Flux-schnell 4-steps as well as Flux-dev (we report examples with 25 steps). To inject a feature in a new generation, we simply rescale it by a strength factor and add it to the output of every layer starting from layer 18, finding that this way we can achieve high-quality results. Figures 8 and 9 show some examples of feature injections with varying strengths.

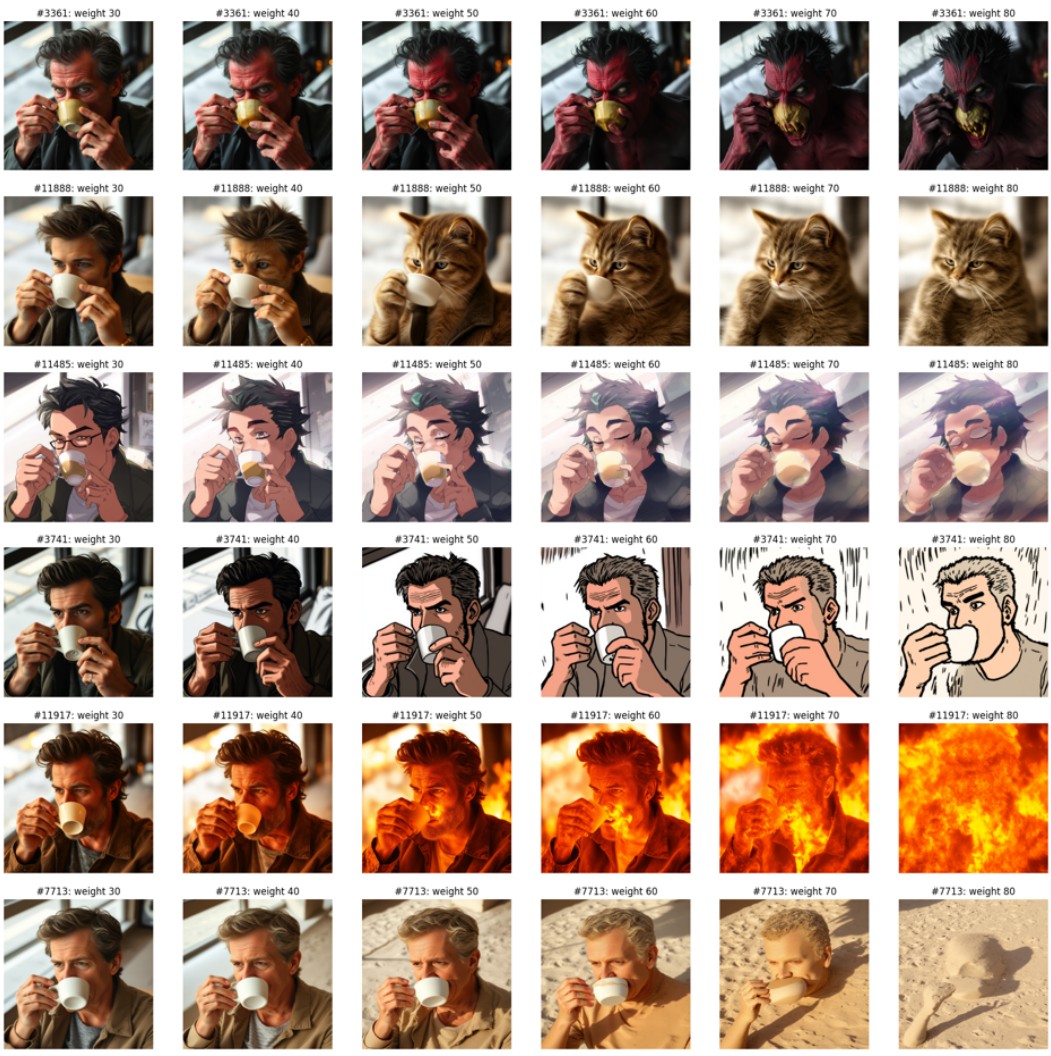

Figure 8: Feature injections on Flux-schnell 4-steps generations.

# D    Finding Causally Influential Transformer Blocks

We narrow down design space of the 11 cross-attention transformer blocks (see Fig. 10) to those with the highest causal impact on the output. In order to assess their causal impact on the output we qualitatively study the effect of individually ablating each of them (see Fig. 11). As can be seen in Fig. 11 each of the middle blocks `down.2.1`, `mid.0`, `up.0.0`, `up.0.1` have a relatively high impact on the output respectively. In particular, the blocks `down.2.1` and `up.0.1` stand out. It seems like most colors and textures are added in `up.0.1`, which in the community is already known as "style" block [57]. Ablating `down.2.1`, which is also already known in the community as "composition" block, impacts the entire image composition, including object sizes, orientations and framing. The effects of ablating other blocks such as `mid.0` and `up.0.0` are more subtle. For `mid.0` it is difficult to describe in words and `up.0.0` seems to add local details to the image while leaving the overall composition mostly intact.

We also provide a quantitative version of this experiment in Tab. 1. As can be seen some of the resnet blocks also exhibit similarly strong effects. We added this experiment during our NeurIPS rebuttal and think it is a promising future direction to investigate these blocks as well.

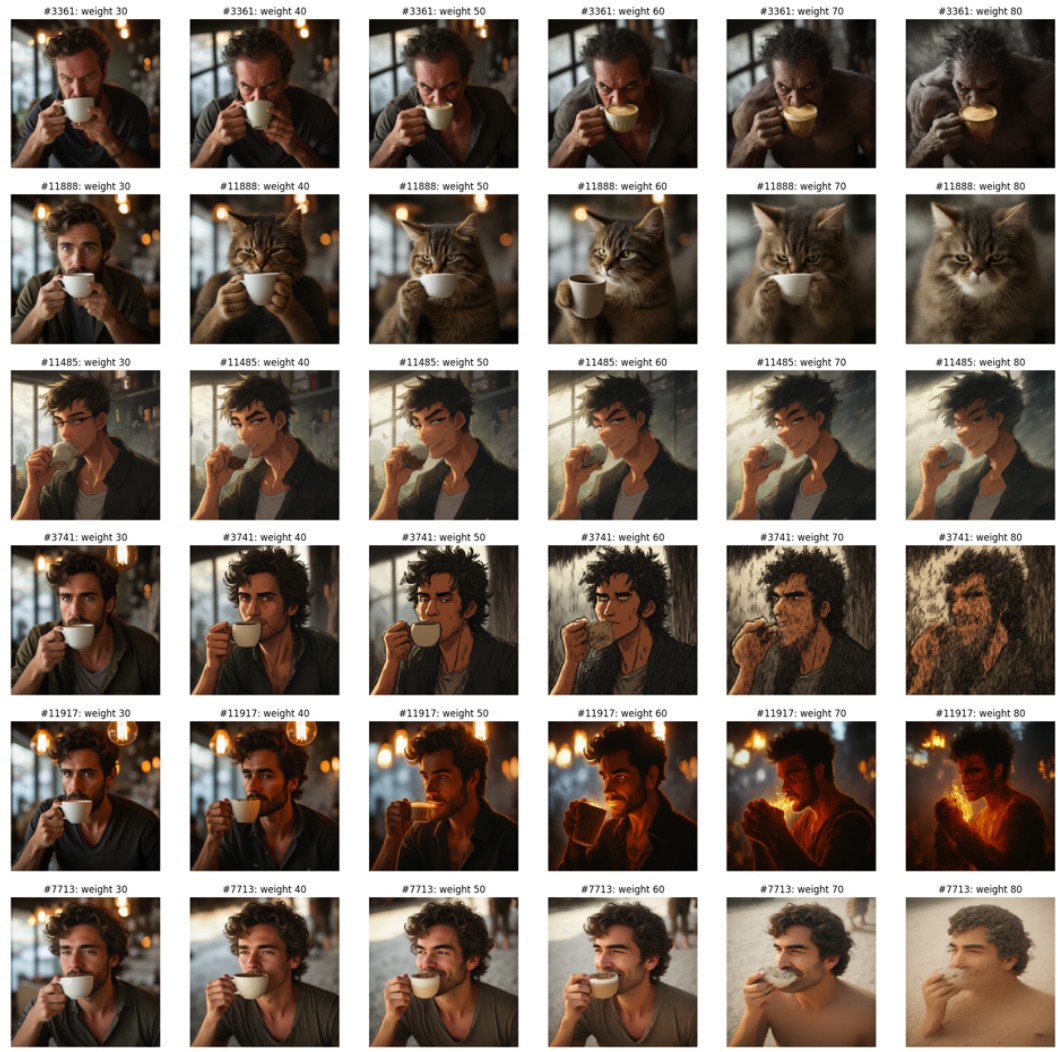

Figure 9: Feature injections on Flux-dev 25-steps generations.

# E  Interventions in the Multi-Step Setting

In addition to our quantitative analysis from the main paper showing that features stabilize fast and are relatively shared across timesteps, here, we performed a series of experiments to also qualitatively assess the impact of performing interventions across multiple timesteps and also on subsets of timesteps. See Fig. 12, 19, 20, 21, 22, 23, 24, and 25.

Broadly, these results are aligned with what one would expect. Intervening from the beginning to the end leads to big perturbations of the original generation. Starting the interventions at later denoising steps keeps more of the original generated image intact. Interestingly, the sliding window of interventions shows that the different transformer blocks can have different effective ranges, e.g., up.0.1 features start working later than down.2.1 features.

# F  RIEBench: Representation-based Image Editing Benchmark

For each of our PIEBench adaptation's edit categories, we implement corresponding feature transport interventions. When selecting feature indices in SDXL Turbo, we aggregate them across spatial positions and timesteps by taking the mean across these dimensions. In FLUX Schnell, we only considered the one-step setting and thus only have to aggregate over spatial locations. The different

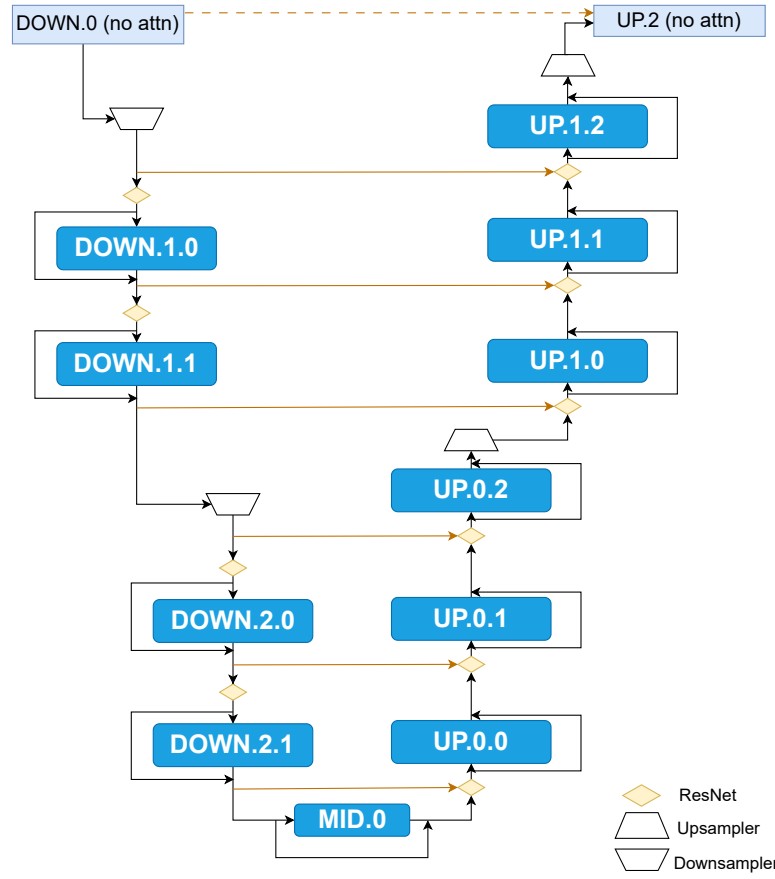

Figure 10: Cross-attention transformer blocks in SDXL's U-net.

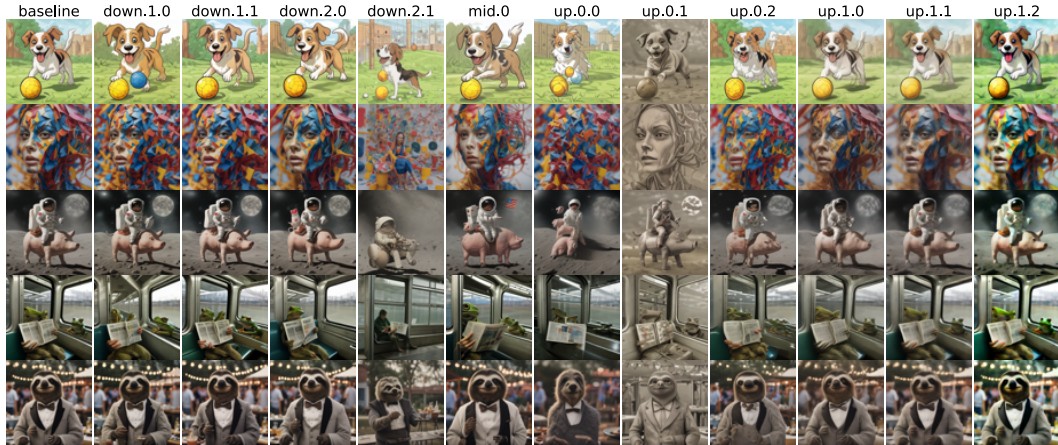

Figure 11: We generate images for the prompts "A dog playing with a ball cartoon.", "A photo of a colorful model.", "An astronaut riding on a pig on the moon.", "A photograph of the inside of a subway train. There are frogs sitting on the seats. One of them is reading a newspaper. The window shows the river in the background." and "A cinematic shot of a professor sloth wearing a tuxedo at a BBQ party." while ablating the updates performed by different cross-attention layers (indicated by the titles). The title "baseline" corresponds to the generation without interventions.

Table 1: **Causal impact of individual resnet and attention blocks.** Each block is ablated independently, and the resulting image is compared to the unperturbed output using the LPIPS distance. We report the mean over 20 randomly generated prompts. Higher values indicate greater causal influence.

| Block Type | Block Name | LPIPS Score |
| --- | --- | --- |
| ResNet | down_blocks.0.resnets.0 | 0.733 |
| ResNet | down_blocks.0.resnets.1 | 0.415 |
| Attention | down_blocks.1.attentions.0 | 0.185 |
| Attention | down_blocks.1.attentions.1 | 0.161 |
| ResNet | down_blocks.1.resnets.0 | 0.457 |
| ResNet | down_blocks.1.resnets.1 | 0.306 |
| Attention | down_blocks.2.attentions.0 | 0.194 |
| Attention | down_blocks.2.attentions.1 | 0.512 |
| ResNet | down_blocks.2.resnets.0 | 0.523 |
| ResNet | down_blocks.2.resnets.1 | 0.270 |
| Attention | mid_block.attentions.0 | 0.345 |
| ResNet | mid_block.resnets.0 | 0.240 |
| ResNet | mid_block.resnets.1 | 0.132 |
| Attention | up_blocks.0.attentions.0 | 0.395 |
| Attention | up_blocks.0.attentions.1 | 0.521 |
| Attention | up_blocks.0.attentions.2 | 0.281 |
| ResNet | up_blocks.0.resnets.0 | 0.348 |
| ResNet | up_blocks.0.resnets.1 | 0.170 |
| ResNet | up_blocks.0.resnets.2 | 0.157 |
| Attention | up_blocks.1.attentions.0 | 0.252 |
| Attention | up_blocks.1.attentions.1 | 0.217 |
| Attention | up_blocks.1.attentions.2 | 0.254 |
| ResNet | up_blocks.1.resnets.0 | 0.199 |
| ResNet | up_blocks.1.resnets.1 | 0.168 |
| ResNet | up_blocks.1.resnets.2 | 0.201 |
| ResNet | up_blocks.2.resnets.0 | 0.275 |
| ResNet | up_blocks.2.resnets.1 | 0.703 |
| ResNet | up_blocks.2.resnets.2 | 0.851 |

interventions for the different categories mainly differ in where the features are collected and whether they are inserted using the target mask or the source mask. We don't preserve spatial information when adding and subtracting feature coefficients / neuron activations / block activations, i.e., at each location within the respective mask the same update is performed. We refer to the mask computed on the target forward pass as target mask and the one computed on the source forward pass as source mask. We split the interventions into three different types:

1. **Change interventions:** We add the top features with coefficients obtained from the source forward pass using also the source mask and we subtract the bottom features with coefficients from the target forward pass using the target mask. *Change object* (SDXL Turbo Fig. 26 and Fig. 13; FLUX Fig. 35), *content* (SDXL Turbo Fig. 29; FLUX Fig. 38), *pose* (SDXL Turbo Fig. 30; FLUX Fig. 39), *color* (SDXL Turbo Fig. 31; FLUX Fig. 40), *material* (SDXL Turbo Fig. 32; FLUX Fig. 41), *background* (SDXL Turbo Fig. 33; FLUX Fig. 42), *style* (SDXL Turbo Fig. 34; FLUX Fig. 43) fall within this category.

2. **Add object:** The *add object* intervention (SDXL Turbo Fig. 27; FLUX Fig. 36) requires special treatment. Here we use the source mask in both forward passes both to select and subsequently add the top and subtract the bottom features.

3. **Delete object:** The *delete object* intervention (SDXL Turbo Fig. 28; FLUX Fig. 37) also requires special treatment. Here we use the target mask in both forward passes both to select and subsequently add the top and subtract the bottom features.

**Neuron selection.** Let $H[\ell]^{\rho,t} \in \mathbb{R}^{h \times w}$ denote a neuron activation map for neuron $\rho$ at layer $\ell$ and timestep $t$. First, to make layers comparable to each other we normalize by L2 norm $\tilde{H}[\ell]_{ij} =$

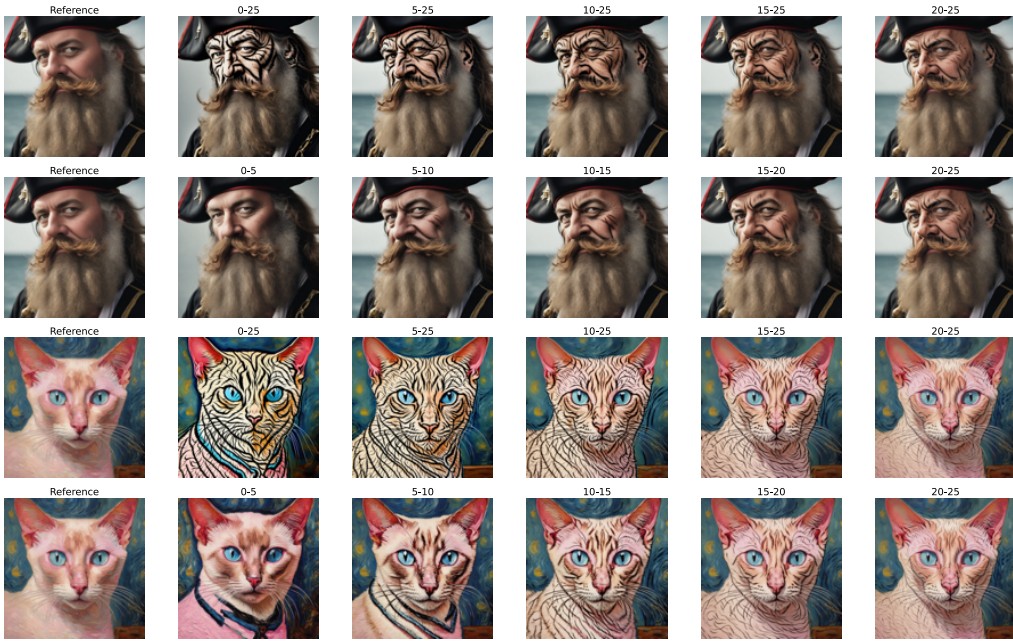

Figure 12: Performing interventions across different time intervals. For each prompt there are two rows, the first row contains ranges 0-25, 5-25, 10-25, 15-25, 20-25 and the second one 0-5, 5-10, 10-15, 15-20, 20-25. We would describe this feature as "tiger texture feature". We intervened with this feature across the entire face of the pirate and across the entire cat except its ears. *These results are from our first working SAE's with $k = 10$ and $n_f = 5120$.* **This is a preview, please find the remaining multi-step intervention figures at the bottom of the document after the text.**

$\frac{H[\ell]_{ij}}{\|H[\ell]_{ij}\|_2}$. Then, similar as in our our feature selection step, we aggregate the neuron activations by taking the mean over timesteps, i.e., $\bar{H}[\ell]^\rho = \frac{1}{T}\sum_{t=1}^T \tilde{H}[\ell]^{\rho,t}$. To determine which neurons to transport, we rank each neuron $\rho$ according to a score $\gamma[\ell]_\rho$ that quantifies the absolute difference in the magnitude of $\rho$ across the source and target, aggregated over the respective grounded SAM2 masks and normalized by the sum of all feature magnitudes:

$$\gamma[\ell]_\rho = \left| \frac{h[\ell]_\rho^{\mathrm{src}}}{\sum_{\rho'=1}^{n_f} h[\ell]_{\rho'}^{\mathrm{src}}} - \frac{h[\ell]_\rho^{\mathrm{tgt}}}{\sum_{\rho'=1}^{n_f} h[\ell]_{\rho'}^{\mathrm{tgt}}} \right|, \tag{9}$$

where $h[\ell]_\rho^{\mathrm{src}} = \frac{1}{|M^{\mathrm{src}}|}\sum_{i,j \in M^{\mathrm{src}}} \bar{H}[\ell]_{ij}^\rho$ and $h[\ell]_\rho^{\mathrm{tgt}} = \frac{1}{|M^{\mathrm{tgt}}|}\sum_{i,j \in M^{\mathrm{tgt}}} \bar{H}[\ell]_{ij}^\rho$. In equation 9 we use absolute difference because GEGLU [56] neurons can take positive and negative values (in contrast to SAE feature coefficients that are always positive). Again, to select neurons among multiple layers we simply concatenate the layer specific score vectors $\gamma[\ell] \in \mathbb{R}^{n_f}$, i.e., $\gamma = \gamma[\ell_1] \cdots \gamma[\ell_{40}]$, where $\cdot$ denotes concatenation, before ranking. For neurons $n_f = 5,120$ for a single layer and there are 40 layers in total within our 4 considered transformer blocks, resulting in $\gamma \in \mathbb{R}^{204,800}$.

**FLUX.** As mentioned in App. C for FLUX we found that interventions are more effective when performing the same update across multiple layers. Thus, in the FLUX figures Fig. 35–Fig. 43 we always have multi-layer interventions (top rows with y-label "multi") and single-layer interventions (bottom rows with y-label "single"). The multi-layer interventions are performed from layer 18 to 56, i.e., for 39 layers. They are effective using small intervention strengths. In contrast, the single-layer interventions in FLUX require big intervention strengths to have a causal influence on the output. While we need to further investigate this our best guess is that multi-layer interventions interact more gracefully with normalization layers than single layer ones and thus introduce less artifacts while achieving a similar effect to using very high intervention strength at a single layer. Further, we omit neuron interventions from the qualitative examples because they did not work in FLUX when performed only using neurons from layer 18.

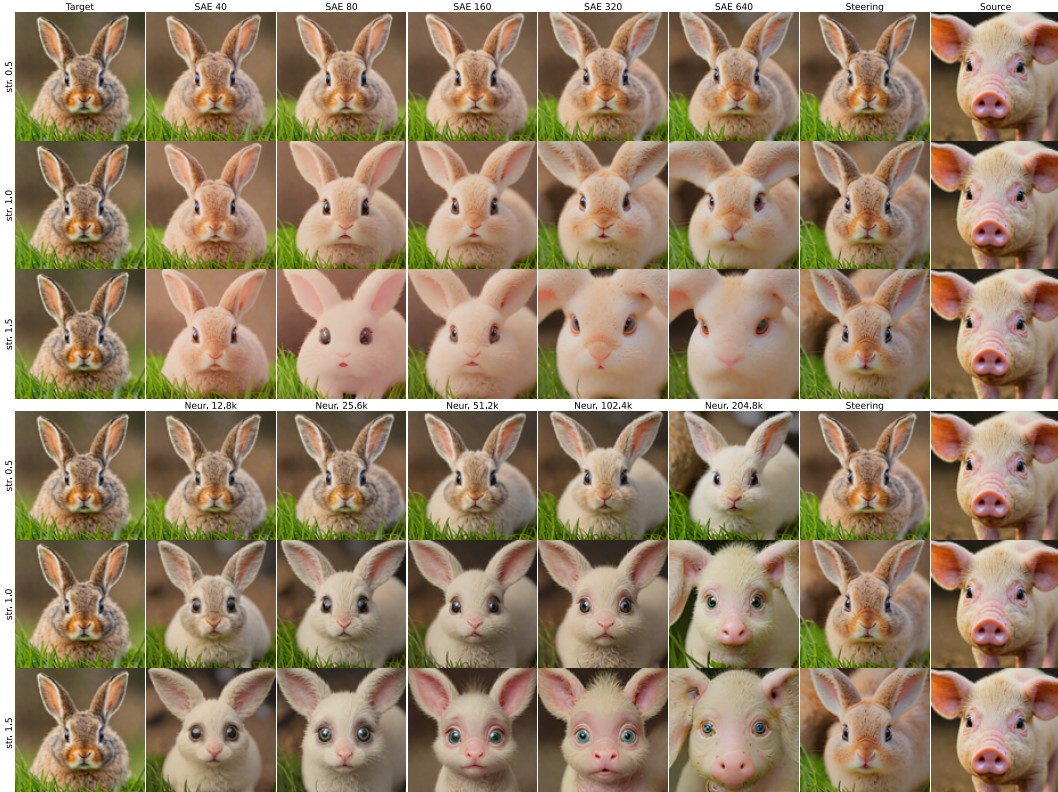

Figure 13: Example for edit category 1: "change object". Original prompt (target): "a cute little bunny with big eyes", edit prompt (source): "a cute little pig with big eyes". Source and target refers to from where we extract features (source) and where we insert them (target). Grounded SAM2 masks used to collect the features are not shown but in this example they would select the entire foreground objects respectively. **This is a preview, please find the figures for the remaining edit categories at the bottom of this document after the text.**

Fig. 14 shows our quantitative evaluation of our FLUX SAE interventions. We omitted neurons form this plot because our neuron interventions based on the neurons from layer 18 did not significantly increase CLIP similarity with the edit prompt in any of the categories. As can be seen, the multi-layer interventions outperform the single-layer ones both for steering as well as for SAE interventions on *change object, add object, change content, color, material, background, style* tasks. On the *delete object and change pose* tasks none of the considered methods works well. The *change pose* tasks is hard to achieve using our current RIEBench setup that in essence consists of interventions adding an update vector to a masked area. We think the bad performance on the *delete object* task can be overcome by improving the delete interventions. E.g., by improving the feature selection strategy or by subtracting the relevant features everywhere in the image and not just locally.

### F.1 Feature Visualization Techniques

We introduce our methods used for feature visualization used in Fig. 15. Informally, given a feature, *spatial activations* (denoted by `hmap`) we highlight the regions of an image where the feature activates during the generation process. *Activation modulation* (A. columns) refers to the intervention process in which the feature activations are enhanced or diminished. This technique is used to demonstrate how the manipulation of a feature's value affects the generated image. Finally, *empty-prompt interventions* (B. column) illustrate the isolated role of the feature by disabling all other features during generation conditioned on an empty prompt. In the remainder of this section, we provide formal definitions and details.

**Spatial activations.** We visualize a sparse feature map $S^\rho \in \mathbb{R}^{h \times w}$ containing activations of a feature $\rho$ across the spatial locations by up-scaling it to the size of the generated images and overlaying it as

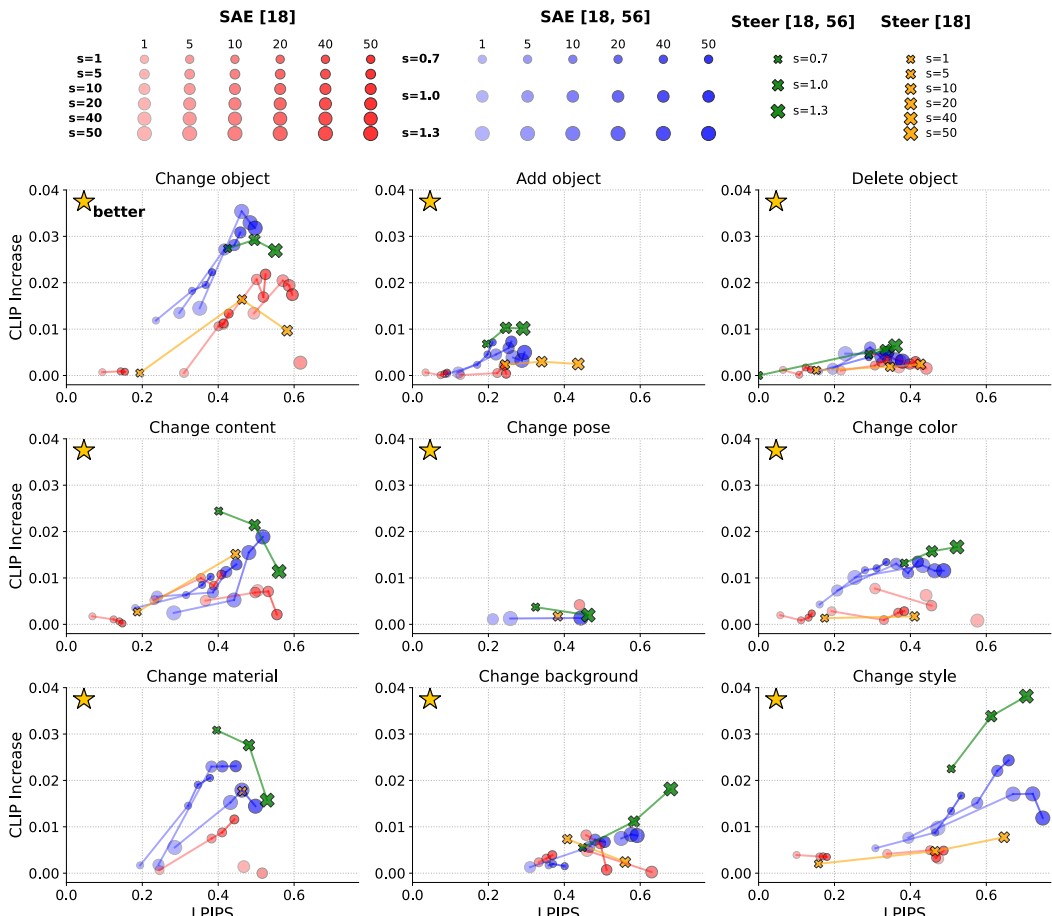

Figure 14: **FLUX Schnell one-step evaluation on RIEBench; LPIPS with original image (x-axis) versus increase in CLIP similarity with the edit prompt (y-axis) from feature-transfer interventions using SAE features, and activations across nine edit categories from PIEBench [26]**. We omitted neuron experiments because they failed to increase CLIP similarity with edit prompt. Features from the edit prompt (within SAM2 segmentation mask) are transferred into the original prompt's forward pass. We experiment with adding SAE/steering based updates to a single layer (SAE red, steering yellow), which is the standard way to do this, as well as adding the same feature on multiple layers (SAE blue, steering green). Strengths (s=1,...,s=50 for single-layer and s=0.7,s=1.0,s=1.3 for multi-layer) and numbers of features transported (1, ..., 50) are indicated in the legend. *We omit all settings that fail to increase CLIP similarity.*

a heatmap over the generated images. In the heatmap, red indicates the highest feature activation, and blue represents the lowest non-zero one.

**Top dataset examples.** For a given feature $\rho$, we sort dataset examples according to their average spatial activation

$$a_\rho = \frac{1}{wh} \sum_{i=1}^{h} \sum_{j=1}^{w} S_{ij}^\rho \in \mathbb{R}. \tag{10}$$

We use equation 10 to define the top dataset examples and to sample from the top 5% quantile of the activating examples ($a_\rho > 0$). We will refer to them as top 5% images for a feature $\rho$.

Note that $S_{ij}^\rho$ always depends on an embedding of the input prompt $c$ and input noise $z_1$, via $S_{ij}(c, z_1) = \text{ENC}(\Delta D_{ij}(c, z_1))$, which we usually omit for ease of notation. As a result, $a_\rho$ also depends on $c$ and $z_1$. When we refer to the top dataset examples, we mean our $(c, z_1)$ pairs with the largest values for $a_\rho(c, z_1)$.

**Activation modulation.** We design interventions that allow us to modulate the strength of the $\rho$th feature. Specifically, we achieve this by adding or subtracting a multiple of the feature $\rho$ on all of the spatial locations $i, j$ proportional to its original activation $S_{ij}^{\rho}$

$$\Delta D_{ij}' = \Delta D_{ij} + \beta S_{ij}^{\rho} \mathbf{f}_{\rho}, \tag{11}$$

in which $\Delta D_{ij}$ is the update performed by the transformer block before and $\Delta D_{ij}'$ after the intervention, $\beta \in \mathbb{R}$ is a modulation factor, and $\mathbf{f}_{\rho}$ is the $\rho$th learned feature vector. In the following, we will refer to this intervention as *activation modulation intervention*.

Note that $S_{ij}^{\rho}$ can be also freely defined allowing for the application of sparse features to arbitrary images and spatial positions (refer to Fig. 1 for examples).

**Activation on empty context.** Another way of visualizing the causal effect of features is to activate them while doing a forward pass on the empty prompt $c(\text{""})$. To do so, we turn off all other features at the transformer block $\ell$ of intervention and turn on the target feature $\rho$. Formally, we modify the forward pass by setting

$$D_{ij}^{out'} = D_{ij}^{in} + \gamma k \mu_{\rho} \mathbf{f}_{\rho}, \tag{12}$$

in which $D_{ij}^{out'}$ replaces residual stream plus transformer block update, $D_{ij}^{in}$ is the input to the block, $\mathbf{f}_{\rho}$ is the $\rho$th learned feature vector, $\gamma \in \mathbb{R}$ is a hyperparameter to adjust the intervention strength, and $\mu_{\rho}$ is a feature-dependent multiplier obtained by taking the average activation across positive activations of $\rho$ (collected over a subset of 50.000 dataset examples). Multiplying it by $k$ aims to recover the coefficients lost by setting the other features to zero. Further in the text, we will refer to this intervention as *empty-prompt intervention*, and the images generated using this method with $\gamma$ set to 1, as *empty-prompt intervention images*.

Note that we directly added/subtracted feature vectors to the dense vectors for both intervention types instead of encoding, manipulating sparse features, and decoding. This approach helps mitigate side effects caused due to reconstruction loss (see App. L).

## G Case Study: Most Active Features on a Prompt

Combining all our feature visualization techniques, in Fig. 15, we depict the features with the highest average activation when processing the prompt: "A cinematic shot of a professor sloth wearing a tuxedo at a BBQ party". We discuss the transformer blocks in order of decreasing interpretability.

**Down.2.1** seems to contribute towards the image composition. Several features seem to relate directly to phrases of the prompt: 4539 "professor sloth", 4751, 1226, "wearing a tuxedo", 2881, 567, 3119, 2345 "party".

Turning off features (A. -6.0 column) removes elements and changes elements in the scene in ways that align with heatmap (hmap column) and the top examples (C columns): 1674 *removes* the light chains in the back, 4608 the umbrellas/tents, 4539 the 3D animation-like sloth face, 567 people in the background, 3119, 2345 some of the light chains, and, 4751 *changes* the type of suit, 1226 the shirt. Similarly, enhancing the same features (A. 6.0 column) enhances the corresponding elements and sometimes changes them.

Activating the features on the empty prompt often creates related elements. Note that, for the fixed random seed we use, the empty prompt itself looks like a painting of a piece of nature with a lot of green and brown. Therefore, while the prompt is empty the features active during the forward pass are not and due to the layers that we don't intervene on still contribute to the images.

While top dataset examples (C.0, C.1 columns) and also empty-prompt intervention (B. column) mostly agree with the feature activation heatmaps (hmap column), some of them add additional insight, e.g., 2881, which activates on the suit, seems to correspond to (masqueraded) characters in a (festive) scene, 3119 seems to be about party decorations in general and not just light chains, 2345 seems to react to other celebration backgrounds as well.

**Up.0.1** transformer block indeed seems to contribute substantially to the style of the image. They are hard to relate directly to phrases in the prompt, yet indirectly they do relate. E.g., the illumination (2727) and shadow (500, 1700) effects probably have something to do with "a cinematic shot" and

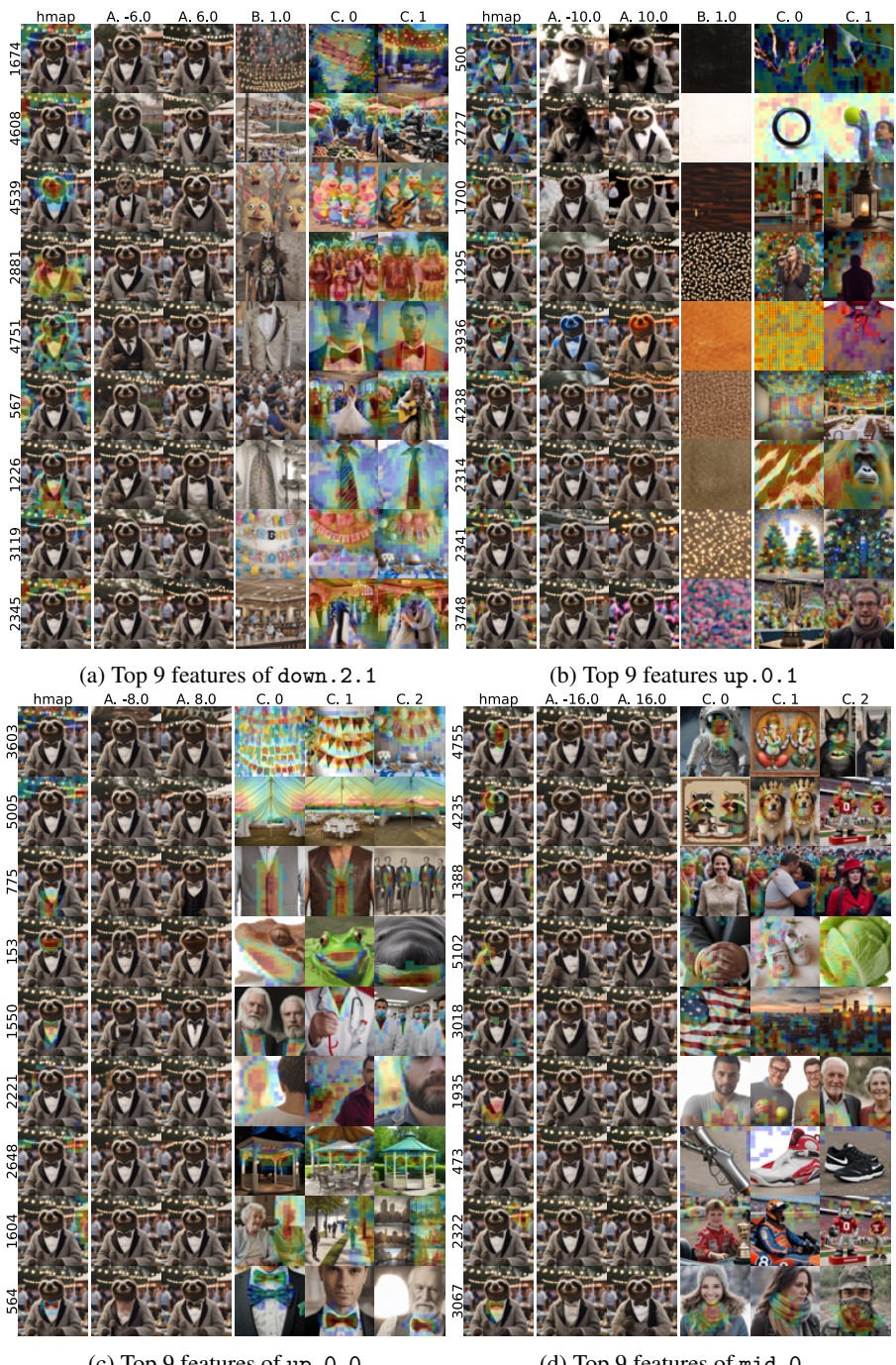

(a) Top 9 features of down.2.1

(b) Top 9 features up.0.1

(c) Top 9 features of up.0.0

(d) Top 9 features of mid.0

Figure 15: The top 9 features of down.2.1 (a), up.0.1 (b), up.0.0 (c) and mid.0 (d) for the prompt: "A cinematic shot of a professor sloth wearing a tuxedo at a BBQ party." Each row represents a feature. The first column depicts a feature heatmap (highest activation red and lowest nonzero one blue). The column titles containing "A" show feature modulation interventions, the ones containing "B" the intervention of turning on the feature on the empty prompt, and the ones containing "C" depict top dataset examples. Floating point values in the title denote $\beta$ and $\gamma$ values. *These results are from our first working SAE's with $k = 10$ and $n_f = 5120$.*

the animal hair texture (2314) with "sloth". Beyond that several features seem to mainly contribute to the glowing lights in the background (1295, 4238, 2341).

Interestingly, turning on the `up.0.1` features on the entire empty prompt (B. column) results in texture-like images. In contrast, when activating them locally (A. columns) their contribution to the output is highly localized and keeps most of the remaining image largely unchanged. For the `up.0.1` we find it remarkable that often the ablation and amplification are counterparts: 500 (light, shadow), 2727 (shadow, light), 3936 (blue, orange), 2314 (less grey hair, more brown hair).

**Up.0.0.** First, we observe that `up.0.0` features act very locally and we think that it often requires relevant other features from the previous and subsequent transformer blocks effectively influence the image. For the empty prompt, activating these features results in abstract looking images, which are hard to relate to the other columns. Thus, we excluded this visualization technique and instead added one more example.

Most top dataset examples and their activations (C columns) are highly interpretable: 3603 party decoration, 5005 upper part of tent, 775 buttons on suit, 153 lower animal jaw, 1550 collars, 2648 pavilions, 1604 right part of the image, 564 bootie. Many of the features have a expected causal effect on the generation when ablating/enhancing (B. columns): 3603, 5005, 775, 153, 1550, 564, but not all: 2221, 2648, 1604. To sum up, this transformer block seems to mostly add local details to the generation and when interventions are performed locally they are effective.

**Mid.0.** Again to the best of our knowledge, `mid.0`'s role is also not well understood. We find it harder to interpret because most interventions on the `mid.0` have very subtle effects. Similar to `up.0.0`, we did not include the results of empty-prompt interventions.

While effects of interventions are subtle, dataset examples (C. columns) and heatmap (hmap column) all mostly agree with each other and are specific enough to be interpretable: 4755 bottom right part of faces, 4235 left part of (animal) faces, 1388 people in the background, 1935 is active on chests, 473 mostly active on the image border, 2322 again seems to have to do with backgrounds that also contain people, 3067 active on the neck or neck accessories, and, 5102 outlines the left border of the main object in the scene. The feature 3018 is difficult to interpret.

Our observations indicate that `mid.0`'s features encode more abstract concepts. Particularly, some of them are activated at specific spatial locations within images[9] and other features potentially signify how image objects relate to each other.

# H   Case Study: Random Features

In this case study, we explore the learned features independently of any specific prompt. *We moved the many and large figures corresponding to this section to the end of the supplementary material.* In Fig. 44 and Fig. 45, we demonstrate the first 5 and last 5 learned features for each transformer block (since SAEs were initialized randomly before training, we can treat these features as a random sample). As SAEs are randomly initialized before the training process, these sets can be considered as random samples of features. Each feature visualization consists of 3 images of top 5% images for this feature, and their perturbations with activation modulation interventions. For `down.2.1` and `up.0.1`, we also include the empty-prompt intervention images. Additionally, we provide visualizations of several selected features in App. H Fig. 46 and demonstrate the effects of their forced activation on unrelated prompts in App. H Fig. 47.

**Feature plots.** We provide the same plots as in Fig. 44 but for the last six feature indices of each transformer block in Fig. 45 and the corresponding prompts in Table 7. Additionally, provide some selected features for `down.2.1` and `up.0.1` in Fig. 46 and the corresponding prompts in Table 8.

**Intervention plots.** Additionally, we provide plots in which we turn on features from Fig. 46 but in unrelated prompts (as opposed to top dataset example prompts that already activate the features by themselves). For simplicity here we simply turn on the features across all spatial locations, which does not seem to be a well suitable strategy for `up.0.1`, which usually acts locally. To showcase, the difference we created one example image in Fig. 16, in which we manually draw localized masks to turn on the corresponding features.

---

[9]SDXL Turbo does not utilize positional encodings for the spatial locations in the feature maps. Therefore, we did a brief sanity check and trained linear probes to detect $i, j$ given $D_{ij}^{in}$. These probes achieved high accuracy on a holdout set: $97.9\%, 98.48\%, 99.44\%, 95.57\%$ for `down.2.1`, `mid.0`, `up.0.0`, `up.0.1`.

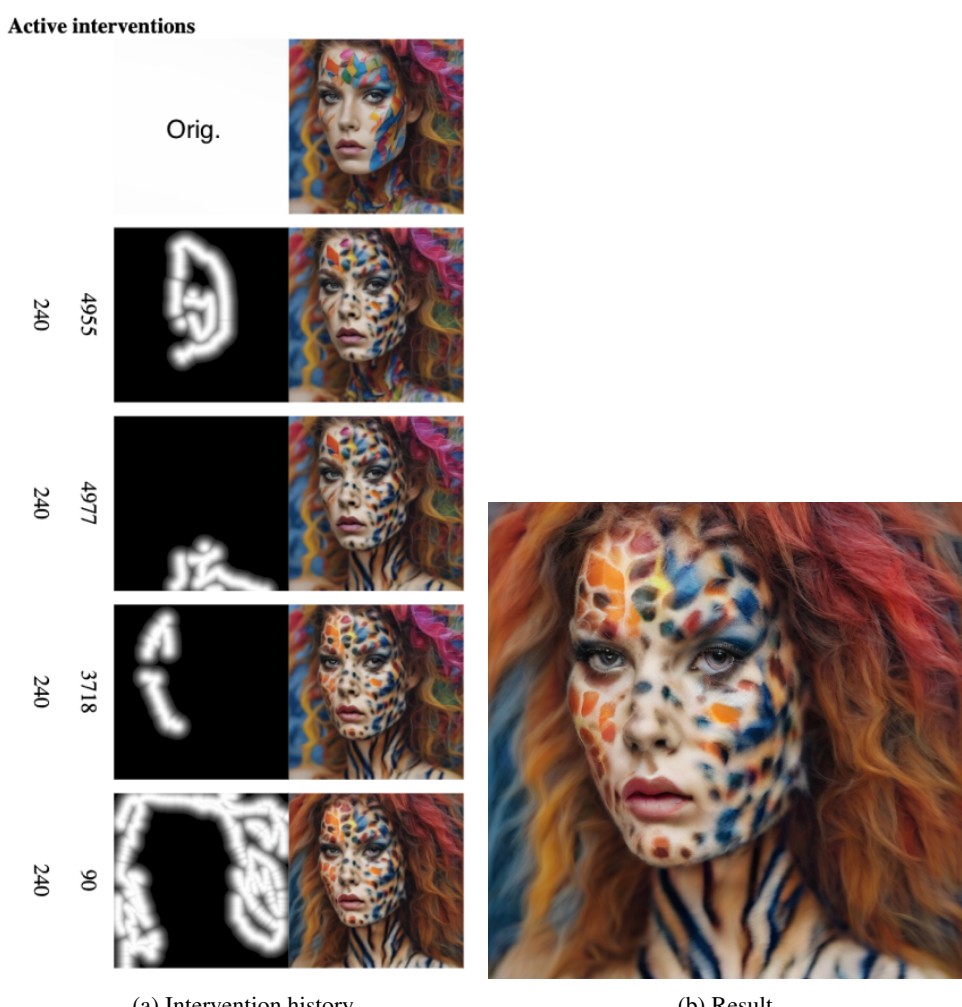

(a) Intervention history        (b) Result

Figure 16: Local edits showcase `up.0.1`'s ability to locally change textures in the image without affecting the remaining image. Multiple consecutive interventions are possible (a). The first in (a) row depicts the original image and each subsequent row we add an intervention by drawing a heatmap with a brush tool and then turning on the feature labelling the row only on that area. The other number (240) is the absolute feature strength of the edit. Figure (b) shows the final result in full resolution (512x512). *These results are from our first working SAE's with $k = 10$ and $n_f = 5120$.*

# I    Quantitative Evaluation of the Roles of the Blocks

In this section, we follow up on qualitative insights by collecting quantitative evidence.

## I.1    Annotation Pipeline

Feature annotation with an LLM followed by further evaluation is a common way to assess feature properties such as specificity, sensitivity, and causality [7]. We found it applicable to the features learned by the `down.2.1` transformer block, which have a strong effect on the generation. Thus, they are amendable to automatic annotation using visual language models (VLMs) such as GPT-4o [38]. In contrast, for the features of other blocks with more subtle effects, we found VLM-generated captions to be unsatisfactory. In order to caption the features of `down.2.1`, we prompt GPT-4o with a sequence of 14 images. The first five images are irrelevant to the feature (i.e., the feature was inactive during the generation of the images), followed by a progression of 4 images with increasing average activation values, and finished by five images with the highest average activation values. The last nine images are provided alongside their so-called "coldmaps": a version of an image with weakly

active and inactive regions being faded and concealed. The prompt template and examples of the captions can be found in the App. J.

### I.2   Experimental Details

We perform a series of experiments to get statistical insights into the features. We report the majority of the experimental scores in the format $M(S)$. When the score is reported in the context of a SDXL Turbo transformer block, it means that we computed the score for each feature of the block and set $M$ and $S$ to mean and standard deviation across the feature scores. Note that $S$ does not represent the error margin of $M$, as the actual error margin is much lower.[10] Therefore, almost all the differences in the reported means are statistically significant. For the baselines, we calculate the mean and standard deviation across the scores of a 100-element sample.

Table 2: Specificity, texture score, and color activation for different blocks and baselines.

| Block | Specificity | Texture | Color |
|---|---|---|---|
| Down.2.1 SAE | 0.76 (0.10) | 0.16 (0.02) | 86.2 (14.9) |
| Down.2.1 Neurons | 0.65 (0.09) | | |
| Down.2.1 PCA all | 0.58 (0.06) | | |
| Down.2.1 PCA 50 | 0.66 (0.08) | | |
| Down.2.1 PCA 100 | 0.64 (0.08) | | |
| Down.2.1 PCA 500 | 0.61 (0.07) | | |
| Mid SAE | 0.70 (0.10) | 0.14 (0.01) | 84.7 (16.3) |
| Mid Neurons | 0.67 (0.07) | | |
| Up.0.0 SAE | 0.74 (0.10) | 0.18 (0.03) | 86.3 (16.5) |
| Up.0.0 Neurons | 0.67 (0.07) | | |
| Up.0.1 SAE | 0.73 (0.09) | 0.20 (0.02) | 73.8 (20.6) |
| Up.0.1 Neurons | 0.66 (0.08) | | |
| Random | 0.57 (0.09) | 0.13 (0.02) | 90.7 (54.9) |
| Same Prompt | 0.89 (0.06) | – | – |
| Textures | – | 0.18 (0.02) | – |

**Interpretability.** Features are usually considered interpretable if they are sufficiently specific, i.e., images exhibiting the feature share some commonality. In order to measure this property, we compute the similarity between images on which the feature is active. High similarity between these images is a proxy for high specificity. For each feature, we collect 10 random images among top 5% images for this feature and calculate their average pairwise CLIP similarity [44, 9]. This value reflects how semantically similar the contexts are in which the feature is most active. We display the results in the first column of Table 2, which shows that the CLIP similarity between images with the feature active is significantly higher then the random baseline (CLIP similarity between random images) for all transformer blocks. This suggests that the generated images share similarities when a feature is active.

For `down.2.1` we compute an additional *interpretability* score by comparing how well the generated annotations align with the top 5% images. The resulting CLIP similarity score is $0.21\ (0.03)$ and significantly higher then the random baseline (average CLIP similarity with random images) $0.12\ (0.02)$. To obtain an upper bound on this score we also compute the CLIP similarity to an image generated from the feature annotation, which is $0.25\ (0.03)$.

**Causality.** We can use the feature annotations to measure a feature's causal strength by comparing the empty prompt intervention images with the caption.[11] The CLIP similarity between intervention images and feature caption is $0.19\ (0.04)$ and almost matches the annotation-based interpretability score of $0.21\ (0.03)$. This suggests that feature annotations effectively describe to the corresponding empty-prompt intervention images. Notably, the annotation pipeline did not use empty-prompt intervention images to generate captions. This fact speaks for the high causal strength of the features learned on `down.2.1`.

**Sensitivity.** A feature is considered sensitive when activated in its relevant context. As a proxy for the context, we have chosen the feature annotations obtained with the auto-annotation pipeline. For each learned feature, we collected the 100 prompts from a 1.5M sample of LAION-COCO with

---

[10]Given that $M$ is computed over a sample of 1280 elements, the confidence interval of $M$ can be estimated as $M \pm S \cdot 0.055$.

[11]We require feature captions for the causality and sensitivity analyses, we only have them for `down.2.1`.

the highest sentence similarity based on sentence transformer embeddings of `all-MiniLM-L6-v2` [48]. Next, we run SDXL Turbo on these prompts and count the proportion of generated images in which the feature is active on more than 0%, 10%, 30% of the image area, resulting in 0.60 (0.32), 0.40 (0.34), 0.27 (0.30) respectively, which is much higher than the random baseline, which is at 0.06 (0.09), 0.003 (0.006), 0.001 (0.003). However, the average scores are $< 1$ and thus not perfect. This may be caused by incorrect or imprecise annotations for subtle features and, therefore, hard to annotate with a VLM and SDXL Turbo failing to comply with some prompts.

**Relatedness to texture.** In Fig. 15 the empty prompt interventions of the `up.0.1` features resulted in texture-like pictures. To quantify whether this consistently happens, we design a simple texture score by computing CLIP similarity between an image and the word "texture". Using this score, we compare empty-prompt interventions of the different transformer blocks with each other and real-world texture images. The results are in the second column of Table 2 and suggest that empty-prompt intervention images of `up.0.1` and `up.0.0` resemble textures and some of the `down.2.1` images look like textures as well. For `up.0.0`, we did not observe any connection of these images to the top activating images. Interestingly, the score of `up.0.1` is higher than the one of the real-world textures dataset (Cimpoi et al. [10]).

**Color sensitivity.** In our qualitative analysis, we suggested that the features learned on `up.0.1` relate to texture and color. If this holds, the image regions that activate a feature should not differ significantly in color on average. To test that, we calculate the "average" color for each feature: this is a weighted average of pixel colors with the feature activation values as weights. To determine the average color of a each feature we compute it over a sample of 10 images of the feature's top 5% images. Then, we calculate Manhattan distances between the colors of the pixels and the "average" color on the same images (the highest possible distance is $3 \cdot 255 = 765$). Finally, we take a weighted average of the Manhattan distances using the same weights. We report these distances for different transformer blocks and for the images generated on random prompts from LAION-COCO. We present the results in the third column of Table 2. The average distance for the `up.0.1` transformer block is, in fact, the lowest.

Table 3: Manhattan distances between original and intervened images at varying intervention strengths outside/inside of the feature's activation map.

| Block | -10 | -5 | 5 | 10 |
|---|---|---|---|---|
| Down.2.1 | 148.2 / 116.0 | 124.2 / 94.4 | 101.4 / 78.7 | 128.9 / 105.60 |
| Mid | 69.2 / 32.2 | 39.4 / 18.5 | 33.2 / 15.2 | 59.9 / 29.82 |
| Up.0.0 | 105.3 / 38.4 | 77.7 / 23.7 | 63.6 / 23.3 | 88.6 / 37.08 |
| Up.0.1 | 125.0 / 26.8 | 73.1 / 16.4 | 68.6 / 21.9 | 98.9 / 34.74 |

**Intervention locality.** We suggested that features learned on `up.0.0` and `up.0.1` primarily influence local regions of the generation, with minimal effect outside the active areas. To test this, we measure changes in the top 5% images inside and outside the active regions while performing activation modulation interventions. To exclude weak activation regions from consideration, a pixel is considered inside the active area if the corresponding patch has an activation value larger than 50% of the image patches, and it is outside the active area if the corresponding patch has zero activation. Table 3 reports Manhattan distances between the original images and the intervened images outside and inside the active areas for activation modulation intervention strengths -10, -5, 5, 10. The features for `up.0.0` and `up.0.1` have a stronger effect inside the active area than outside, unlike `down.2.1` where the difference is smaller.

# J   Annotation Pipeline Details

We used GPT-4o to caption learned features on `down.2.1`. For each feature, the model was shown a series of 5 unrelated images, a progression of 9 images, the $i$-th of those corresponds to $\sim i \cdot 10\%$ average activation value of the maximum. Finally, we show 5 images corresponding to the highest average activations. Since some features are active on particular parts of images, the last 9 images are provided alongside their so-called "coldmaps": a version of an image with weakly active and inactive regions being faded and concealed.

The images were generated by 1-step SDXL Turbo diffusion process on $50'000$ random prompts of LAION-COCO dataset.

### J.1 Textual Prompt Template

Here is the prompt template for the VLM.

> **System.** You are an experienced mechanistic interpretability researcher that is labeling features from the hidden representations of an image generation model.
>
> **User.** You will be shown a series of images generated by a machine learning model. These images were selected because they trigger a specific feature of a sparse auto-encoder, trained to detect hidden activations within the model. This feature can be associated with a particular object, pattern, concept, or a place on an image. The process will unfold in three stages:
>
> 1. **Reference Images:** First, you'll see several images *unrelated* to the feature. These will serve as a reference for comparison.
>
> 2. **Feature-Activating Images:** Next, you'll view images that activate the feature with varying strengths. Each of these images will be shown alongside a version where non-activated regions are masked out, highlighting the areas linked to the feature.
>
> 3. **Strongest Activators:** Finally, you'll be presented with the images that most strongly activate this feature, again with corresponding masked versions to emphasize the activated regions.
>
> Your task is to carefully examine all the images and identify the thing or concept represented by the feature. Here's how to provide your response:
>
> - **Reasoning:** Between '<thinking>' and '</thinking>' tags, write up to 400 words explaining your reasoning. Describe the visual patterns, objects, or concepts that seem to be consistently present in the feature-activating images but not in the reference images.
>
> - **Expression:** Afterward, between '<answer>' and '</answer>' tags, write a concise phrase (no more than 15 words) that best captures the common thing or concept across the majority of feature-activating images.
>
> Note that not all feature-activating images may perfectly align with the concept you're describing, but the images with stronger activations should give you the clearest clues. Also pay attention to the masked versions, as they highlight the regions most relevant to the feature.
>
> **User.** These images are not related to the feature: {Reference Images}
>
> **User.** This is a row of 9 images, each illustrating increasing levels of feature activation. From left to right, each image shows a progressively higher activation, starting with the image on the far left where the feature is activated at 10% relative to the image that activates it the most, all the way to the far right, where the feature activates at 90% relative to the image that activates it the most. This gradual transition highlights the feature's growing importance across the series. {Feature-Activating Images}
>
> **User.** This row consists of 9 masked versions of the original images. Each masked image corresponds to the respective image in the activation row. Areas where the feature is not activated are completely concealed by a white mask, while regions with activation remain visible.) {Feature-Activating Images Coldmaps}
>
> **User.** These images activate the feature most strongly. {Strongest Activators}
>
> **User.** These masked images highlight the activated regions of the images that activate the feature most strongly. The masked images correspond to the images above. The unmasked regions are the ones that activate the feature. {Strongest Activators Coldmaps}

### J.2 Example of Prompt Images

The images used to annotate feature 0 are shown in Fig. 17.

### J.3 Examples of Generated Captions

We present the captions generated by GPT-4o for the first and last 10 features in Table 4.

Table 4: `down.2.1` first 10 and last 10 feature captions.

| Block | Feature | Caption |
|-------|---------|---------|
| `down.2.1` | 0 | Organizational/storage items for documents and office supplies |
| | 1 | Luxury kitchen interiors and designs |
| | 2 | Architectural Landmarks and Monumental Buildings |
| | 3 | Upper body clothing and attire |
| | 4 | Rustic or Natural Wooden Textures or Surfaces |
| | 5 | Intricately designed and ornamental brooches |
| | 6 | Technical diagrams and instructional content |
| | 7 | Feature predominantly activated by visual representations of dresses |
| | 8 | Home decor textiles focusing on cushions and pillows |
| | 9 | Eyewear: glasses and sunglasses |
| | 5110 | Concept of containment or organized enclosure |
| | 5111 | Groups of people in collective settings |
| | 5112 | Modern minimalist interior design |
| | 5113 | Indoor plants and greenery |
| | 5114 | Feature sensitivity focused on sneakers |
| | 5115 | Handling or manipulating various objects |
| | 5116 | Athletic outerwear, particularly zippered sporty jackets |
| | 5117 | Spectator Seating in Sporting Venues |
| | 5118 | Textiles and clothing materials, focus on textures and folds |
| | 5119 | Yarn and Knitting Textiles |

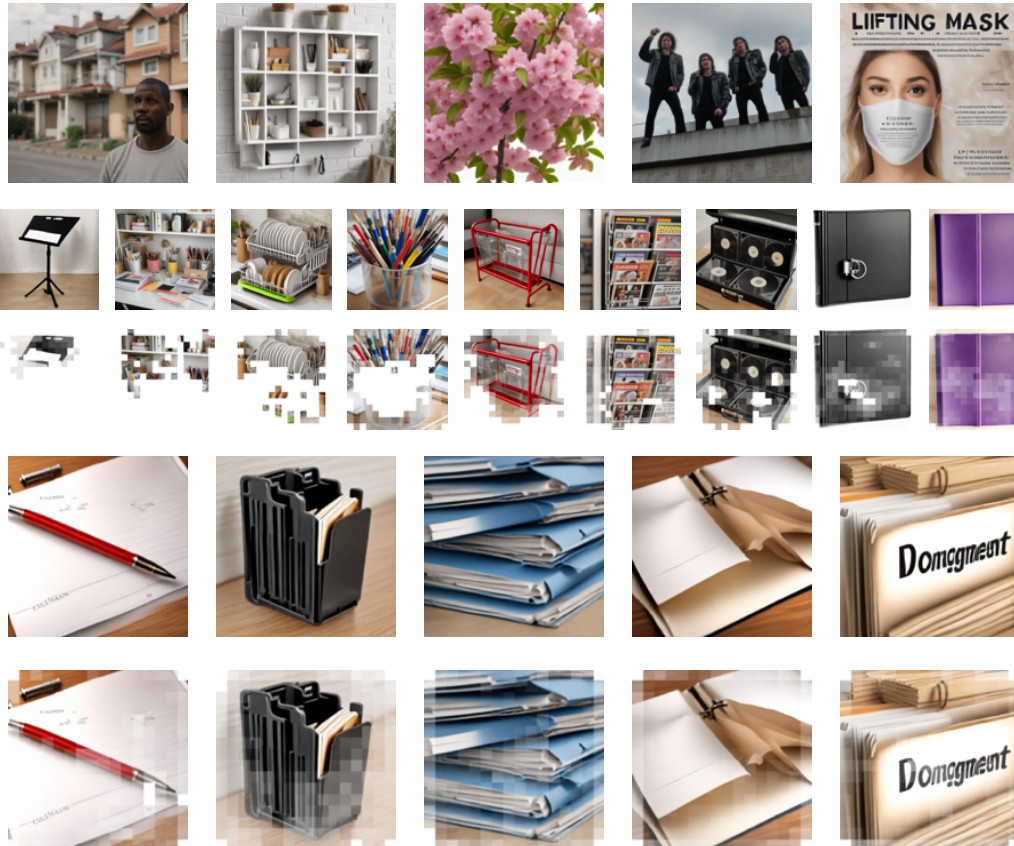

Figure 17: The images used by GPT-4o to generate captions for feature 0. From top to bottom: irrelevant images to feature 0; image progression from left to right, showing increasing activation of SAE feature 0, with low activation on the left and high activation on the right; "Coldmaps" representing the image progression; images corresponding to the highest activation of feature 0; "Coldmaps" corresponding to these highest activation images.

## K   Sparse Autoencoders and Superposition

This is an extended version of Sparse Autoencoders subsection of background section.

Let $h(x) \in \mathbb{R}^d$ be some intermediate result of a forward pass of a neural network on the input $x$. In a fully connected neural network, the components $h(x)$ could correspond to neurons. In transformers, which are residual neural networks with attention and fully connected layers, $h(x)$ usually either refers to the content of the residual stream after some layer, an update to the residual stream by some layer, or the neurons within a fully connected block. In general, $h(x)$ could refer to anything, e.g., also keys, queries, and values. It has been shown [62, 11, 5] that in many neural networks, especially LLMs, intermediate representations can be well approximated by sparse sums of $n_f \in \mathbb{N}$ learned feature vectors, i.e.,

$$h(x) \approx \sum_{\rho=1}^{n_f} s_\rho(x) \mathbf{f}_\rho, \tag{13}$$

where $s_\rho(x)$ are the input-dependent[12] coefficients most of which are equal to zero and $\mathbf{f}_1, \dots, \mathbf{f}_{n_f} \in \mathbb{R}^d$ is a learned dictionary of feature vectors.

Importantly, these learned features are usually highly *interpretable* (specific), *sensitive* (fire on the relevant contexts), *causal* (change the output in expected ways in intervention) and usually do not correspond directly to individual neurons. There are also some preliminary results on the universality of these learned features, i.e., that different training runs on similar data result in the corresponding models picking up largely the same features [5].

**Superposition.** By associating task-relevant features with directions in $\mathbb{R}^d$ instead of individual components of $h(x) \in \mathbb{R}^d$, it is possible to represent many more features than there are components, i.e., $n_f >> d$. As a result, in this case, the learned dictionary vectors $\mathbf{f}_1, \dots, \mathbf{f}_{n_f}$ cannot be orthogonal to each other, which can lead to interference when too many features are on (thus the sparsity requirement). However, it would be theoretically possible to have exponentially (in $d$) many almost orthogonal directions embedded in $\mathbb{R}^d$.[13]

Using representations like this, the optimization process during training can trade off the benefits of being able to represent more features than there are components in $h$ with the costs of features interfering with each other. Such representations are especially effective if the real features underlying the data do not co-occur with each other too much, that is, they are sparse. In other words, in order to represent a single input ("Michael Jordan") only a small subset of the features ("person", ..., "played basketball") is required [17, 5].

The phenomenon of neural networks that exploit representations with more features than there are components (or neurons) is called superposition [17]. Superposition can explain the presence of polysemantic neurons. The neurons, in this case, are simply at the wrong level of abstraction. The closest feature vector can change when varying a neuron, resulting in the neuron seemingly reacting to or steering semantically unrelated things.

**Sparse autoencoders.** In order to implement the sparse decomposition from equation 13, the vector $s$ containing the $n_f$ coefficients of the sparse sum is parameterized by a single linear layer followed by an activation function, called the *encoder*,

$$s = \text{ENC}(h) = (W^{\text{ENC}}(h - b_{\text{pre}}) + b_{\text{act}}), \tag{14}$$

in which $h \in \mathbb{R}^d$ is the latent that we aim to decompose, $\sigma(\cdot)$ is an activation function, $W^{\text{ENC}} \in \mathbb{R}^{n_f \times d}$ is a learnable weight matrix and $b_{\text{pre}}$ and $b_{\text{act}}$ are learnable bias terms. We omitted the dependencies $h = h(x)$ and $s = s(h)$ that are clear from context.

Similarly, the learnable features are parametrized by a single linear layer, called *decoder*,

$$h' = \text{DEC}(s) = W^{\text{DEC}} s + b_{\text{pre}}, \tag{15}$$

in which $W^{\text{DEC}} = (\mathbf{f}_1 | \cdots | \mathbf{f}_{n_f}) \in \mathbb{R}^{d \times n_f}$ is a learnable matrix of whose columns take the role of learnable features and $b_{\text{pre}}$ is a learnable bias term.

---

[12]In the literature this input dependence is usually omitted.

[13]It follows from the Johnson-Lindenstrauss Lemma [25] that one can find at least $\exp(d\epsilon^2/8)$ unit vectors in $\mathbb{R}^d$ with the dot product between any two not larger than $\epsilon$.

**Training.** The pair ENC and DEC are trained in a way that ensures that $h'$ is a sparse sum of feature vectors. Given a dataset of latents $h_1, \ldots, h_n$, both encoder and decoder are trained jointly to minimize a proxy to the loss

$$\min_{\substack{W^{\text{ENC}}, W^{\text{DEC}} \\ b_{\text{pre}}, b_{\text{act}}}} \sum_{i=1}^{n} \|h'_i - h_i\|_2^2 + \lambda \|s_i\|_0, \tag{16}$$

where $h_i = h(x_i)$, $s_i = \text{ENC}(h(x_i))$ (when we refer to components of $s$ we use $s_\rho$ instead), the $\|h'_i - h_i\|_2^2$ is a reconstruction loss, $\|s_i\|_0$ a regularization term ensuring the sparsity of the activations and $\lambda$ the corresponding trade-off term.

In practice, $\|s_i\|_0$ cannot be efficiently optimized directly, which is why it is usually replaced with $\|s_i\|_1$ or other proxy objectives.

**Technical details.** In our work, we make use of the top-$k$ formulation from [19], in which $\|s_i\|_0 \leq k$ is ensured by introducing the a top-$k$ function TopK into the encoder:

$$s = \text{ENC}(h) = \text{RELU}(\text{TopK}(W^{\text{ENC}}(h - b_{\text{pre}}) + b_{\text{act}})). \tag{17}$$

As the name suggests, TopK returns a vector that sets all components except the top $k$ ones to zero.

In addition [19] use an auxiliary loss to handle dead features. During training, a sparse feature $\rho$ is considered *dead* if $s_\rho$ remains zero over the last 10M training examples.

The resulting training loss is composed of two terms: the $L_2$-reconstruction loss and the top-auxiliary $L_2$-reconstruction loss for dead feature reconstruction. For a single latent $h$, the loss is defined

$$L(h, h') = \|h - h'\|_2^2 + \alpha \|h - h'_{\text{aux}}\|_2^2 \tag{18}$$

In this equation, the $h'_{\text{aux}}$ is the reconstruction based on the top $k_{\text{aux}}$ dead features. This auxiliary loss is introduced to mitigate the issue of dead features. After the end of the training process, we observed none of them. Following [19], we set $\alpha = \frac{1}{32}$ and $k_{\text{aux}} = 256$, performed tied initialization of encoder and decoder, normalized decoder rows after each training step. The number of learned features $n_f$ is set to 5120, which is four times the length of the input vector. The value of $k$ is set to 10 as a good trade-off between sparsity and reconstruction quality. Other training hyperparameters are batch size: 4096, optimizer: Adam with learning rate: $10^{-4}$ and betas: $(0.9, 0.999)$.

## L   SAE Training Results

We trained several SAEs with different sparsity levels and sparse layer sizes and observed no dead features. To assess reconstruction quality, we processed 100 random LAION-COCO prompts through a one-step SDXL Turbo process, replacing the additive component of the corresponding transformer block with its SAE reconstruction.

The explained variance ratio and the output effects caused by reconstruction are shown in Table 5. Fig. 18 presents random examples of reconstructions from an SAE with the following hyperparameters: $k = 10, n_f = 5120$, trained on down.2.1. The reconstruction causes minor deviations in the images, and the fairly low LPIPS [63] and pixel distance scores also support these findings. However, to prevent these minor reconstruction errors from affecting our analysis of interventions, we decided to directly add or subtract learned directions from dense feature maps.

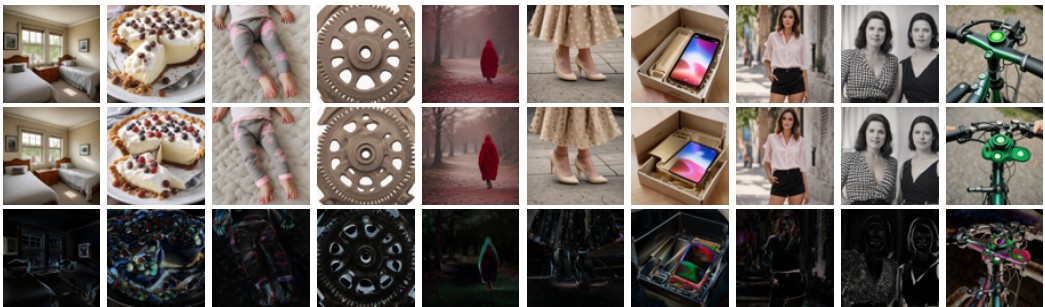

Figure 18: Images generated from 10 random prompts taken from the LAION-COCO dataset are shown in the first row. In the second row, `down.2.1` updates are replaced by their SAE reconstructions ($k = 10, n_f = 5120$). The third row visualizes the differences between the original and reconstructed images.

Table 5: Distances and explained variance ratio in generated images. "Mean" represents the average pixel Manhattan distance between original and reconstruction-intervened images, with a maximum possible value of 765. "Median" represents the median Manhattan distance per pixel, averaged over all images. 'LPIPS' refers to the average LPIPS score, measuring perceptual similarity. "Explained variance ratio" denotes the ratio of variance explained by the trained SAEs to the total variance.

| $k$ | $n_f$ | Configuration | Mean | Median | LPIPS | EV (%) |
|---|---|---|---|---|---|---|
| 5 | 640 | down.2.1 | 83.29 | 50.04 | 0.3383 | 56.0 |
| | | mid.0 | 52.64 | 26.82 | 0.2032 | 43.4 |
| | | up.0.0 | 55.89 | 30.69 | 0.2276 | 44.8 |
| | | up.0.1 | 52.67 | 34.53 | 0.2073 | 50.3 |
| | 5120 | down.2.1 | 74.68 | 41.49 | 0.3036 | 67.8 |
| | | mid.0 | 48.82 | 24.60 | 0.1845 | 50.8 |
| | | up.0.0 | 49.19 | 25.86 | 0.1969 | 57.2 |
| | | up.0.1 | 47.50 | 31.11 | 0.1775 | 59.5 |
| 10 | 640 | down.2.1 | 73.65 | 41.79 | 0.2893 | 62.8 |
| | | mid.0 | 46.80 | 23.10 | 0.1772 | 51.5 |
| | | up.0.0 | 48.43 | 25.80 | 0.1908 | 52.5 |
| | | up.0.1 | 43.06 | 26.85 | 0.1638 | 58.7 |
| | 5120 | down.2.1 | 64.97 | 34.77 | 0.2582 | 73.7 |
| | | mid.0 | 44.02 | 21.72 | 0.1627 | 58.8 |
| | | up.0.0 | 42.08 | 21.54 | 0.1624 | 64.2 |
| | | up.0.1 | 39.77 | 24.84 | 0.1453 | 67.1 |
| 20 | 640 | down.2.1 | 59.29 | 31.47 | 0.2291 | 69.9 |
| | | mid.0 | 39.95 | 19.44 | 0.1459 | 60.0 |
| | | up.0.0 | 40.15 | 21.06 | 0.1499 | 60.9 |
| | | up.0.1 | 31.97 | 18.15 | 0.1196 | 66.7 |
| | 5120 | down.2.1 | 56.37 | 29.04 | 0.2190 | 78.8 |
| | | mid.0 | 37.28 | 17.82 | 0.1328 | 66.5 |
| | | up.0.0 | 35.73 | 18.03 | 0.1302 | 70.6 |
| | | up.0.1 | 30.31 | 17.22 | 0.1104 | 74.2 |

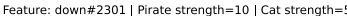

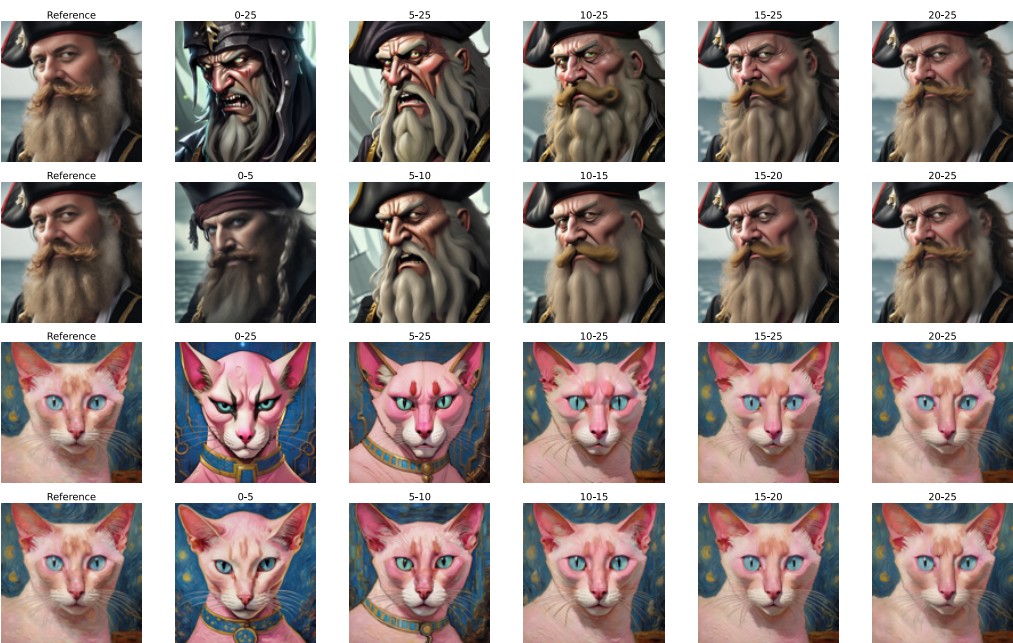

Figure 19: Performing interventions across different time intervals. For each prompt there are two rows, the first row contains ranges 0-25, 5-25, 10-25, 15-25, 20-25 and the second one 0-5, 5-10, 10-15, 15-20, 20-25. We would describe this feature as "evil feature". We intervened with this feature across the entrire spatial grid. *These results are from our first working SAE's with $k = 10$ and $n_f = 5120$.*

Feature: down#4998 | Pirate strength=5 | Cat strength=5

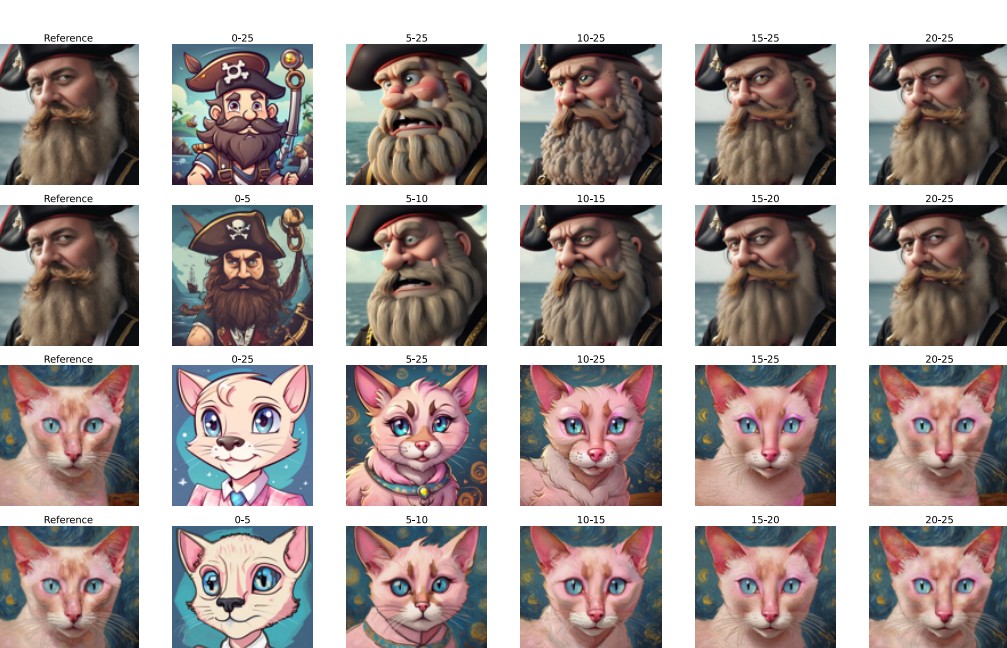

Figure 20: Performing interventions across different time intervals. For each prompt there are two rows, the first row contains ranges 0-25, 5-25, 10-25, 15-25, 20-25 and the second one 0-5, 5-10, 10-15, 15-20, 20-25. We would describe this feature as "cartoon feature". We intervened with this feature across the entire spatial grid. *These results are from our first working SAE's with $k = 10$ and $n_f = 5120$.*

Feature: up0#1941 | Pirate strength=5 | Cat strength=3

Figure 21: Performing interventions across different time intervals. For each prompt there are two rows, the first row contains ranges 0-25, 5-25, 10-25, 15-25, 20-25 and the second one 0-5, 5-10, 10-15, 15-20, 20-25. We would describe this feature as "ear feature". We intervened with this feature on the ears. *These results are from our first working SAE's with $k = 10$ and $n_f = 5120$.*

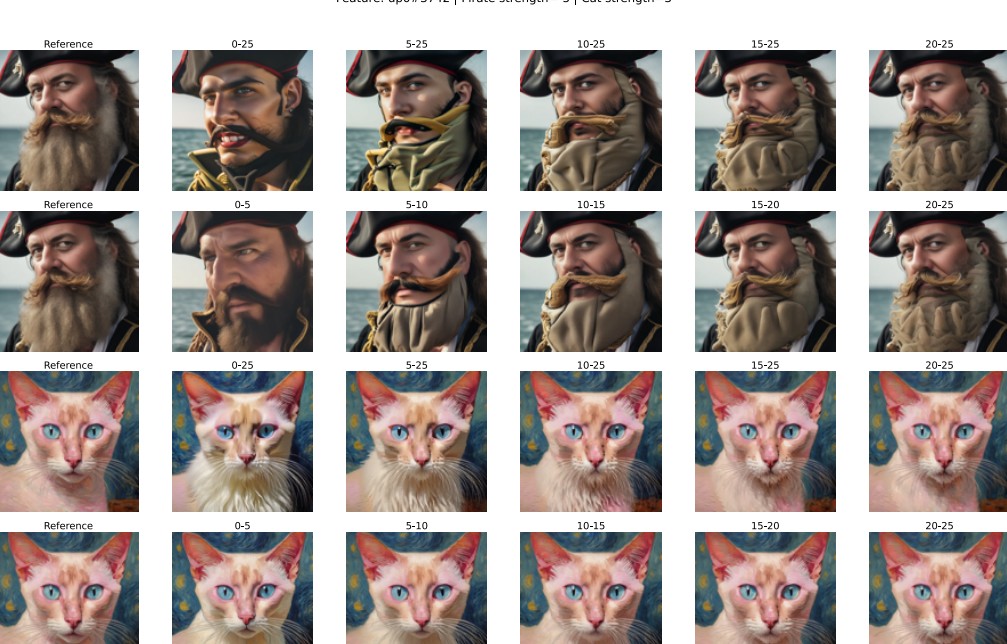

Feature: up0#3742 | Pirate strength=-5 | Cat strength=3

Figure 22: Performing interventions across different time intervals. For each prompt there are two rows, the first row contains ranges 0-25, 5-25, 10-25, 15-25, 20-25 and the second one 0-5, 5-10, 10-15, 15-20, 20-25. We would describe this feature as "beard feature". We intervened with this feature on the chin/beard area. In the pirate we subtracted this feature. *These results are from our first working SAE's with $k = 10$ and $n_f = 5120$.*

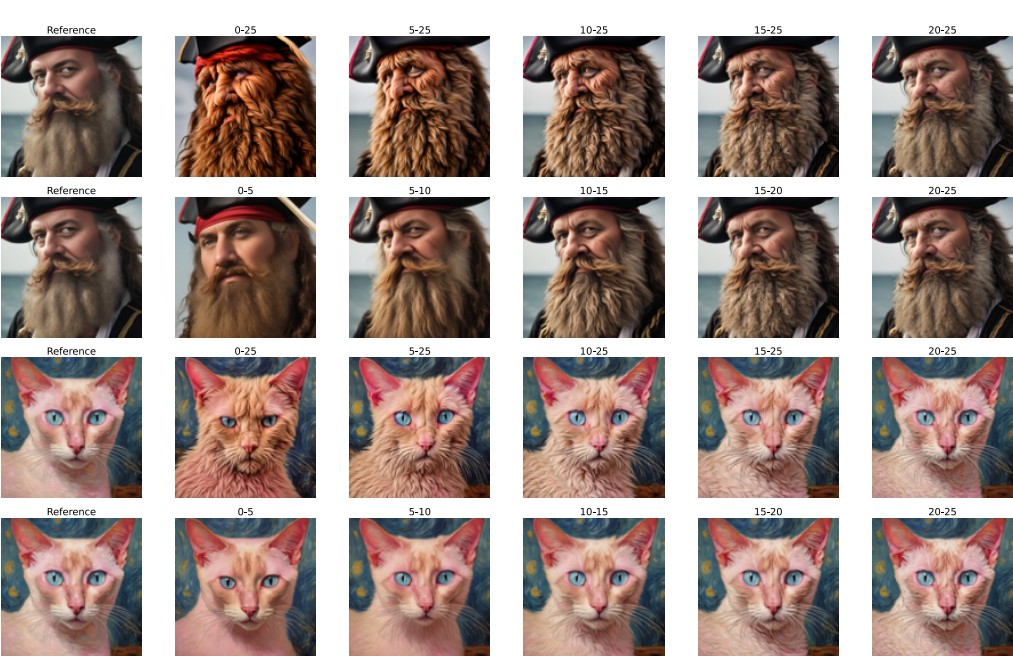

Figure 23: Performing interventions across different time intervals. For each prompt there are two rows, the first row contains ranges 0-25, 5-25, 10-25, 15-25, 20-25 and the second one 0-5, 5-10, 10-15, 15-20, 20-25. We would describe this feature as "furry feature". We intervened with this feature across the entire beard and face of the pirate and across the entire cat except its ears. *These results are from our first working SAE's with $k = 10$ and $n_f = 5120$.*

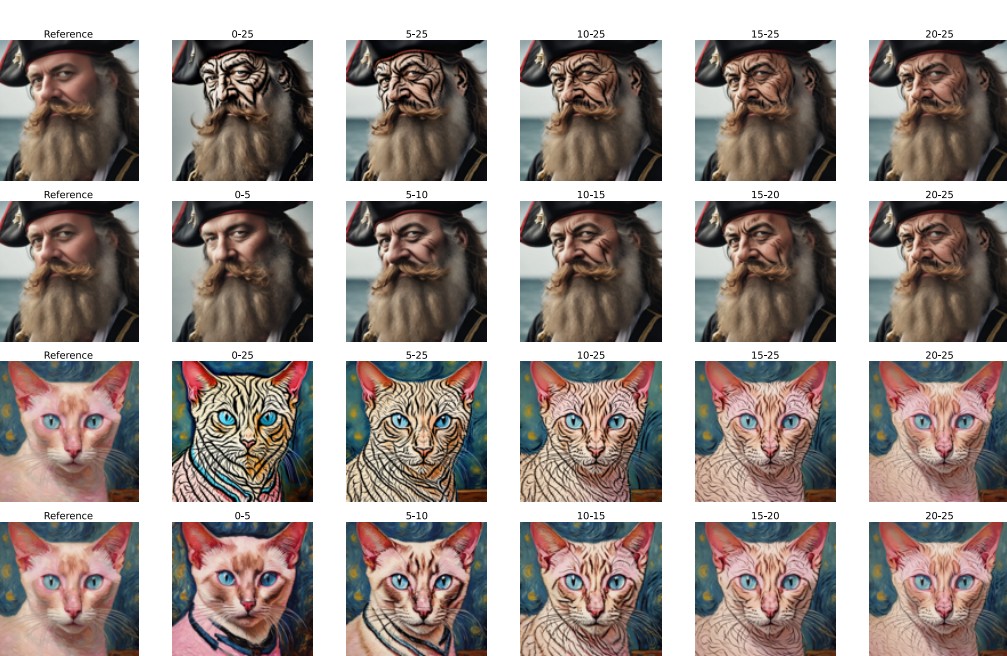

Figure 24: Performing interventions across different time intervals. For each prompt there are two rows, the first row contains ranges 0-25, 5-25, 10-25, 15-25, 20-25 and the second one 0-5, 5-10, 10-15, 15-20, 20-25. We would describe this feature as "tiger texture feature". We intervened with this feature across the entire face of the pirate and across the entire cat except its ears. *These results are from our first working SAE's with $k = 10$ and $n_f = 5120$.*

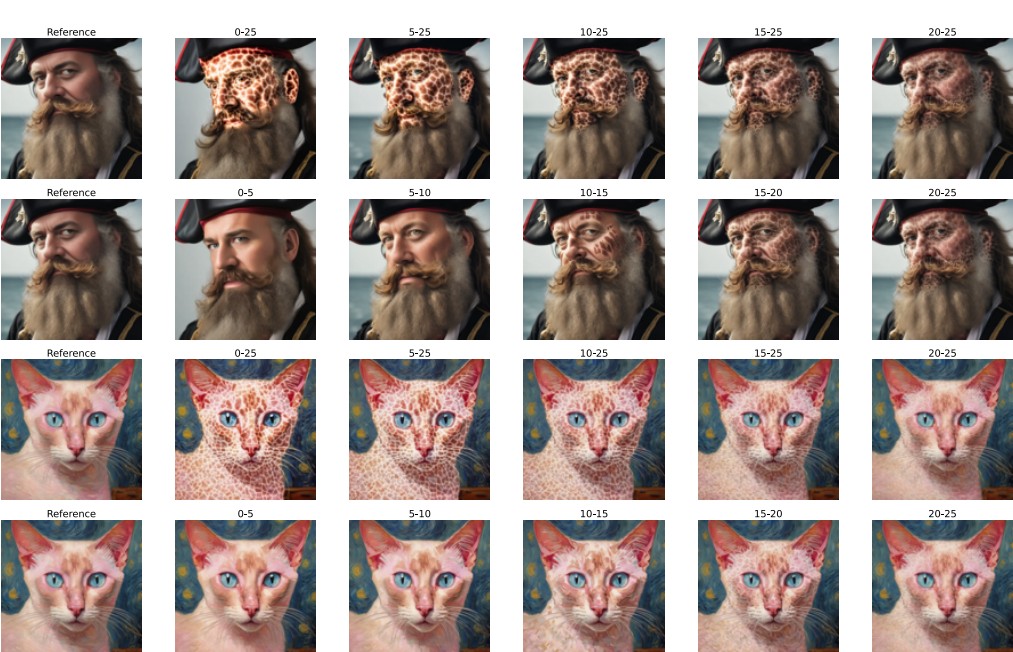

Figure 25: Performing interventions across different time intervals. For each prompt there are two rows, the first row contains ranges 0-25, 5-25, 10-25, 15-25, 20-25 and the second one 0-5, 5-10, 10-15, 15-20, 20-25. We would describe this feature as "giraffe pattern feature". We intervened with this feature across the entire face of the pirate and across the entire cat except its ears. *These results are from our first working SAE's with $k = 10$ and $n_f = 5120$.*

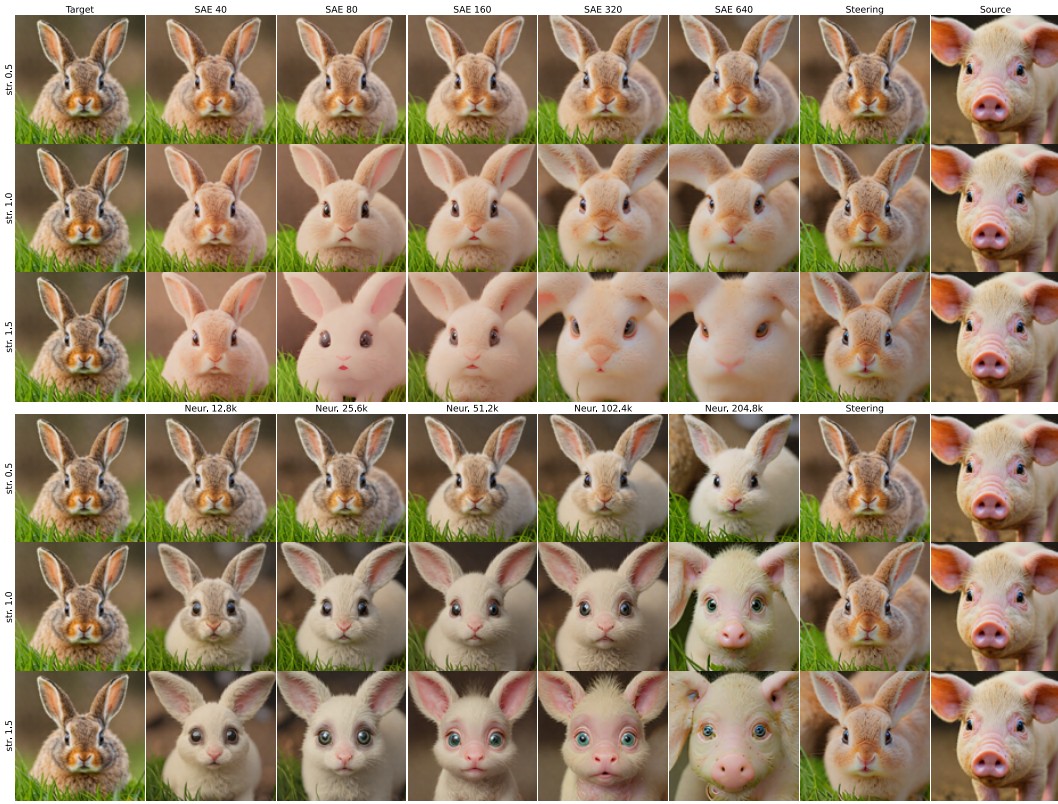

Figure 26: SDXL Turbo 4-step interventions; Example for edit category 1: "change object". Original prompt (target): "a cute little bunny with big eyes", edit prompt (source): "a cute little pig with big eyes". Source and target refers to from where we extract features (source) and where we insert them (target). Grounded SAM2 masks used to collect the features are not shown but in this example they would select the entire foreground objects respectively.

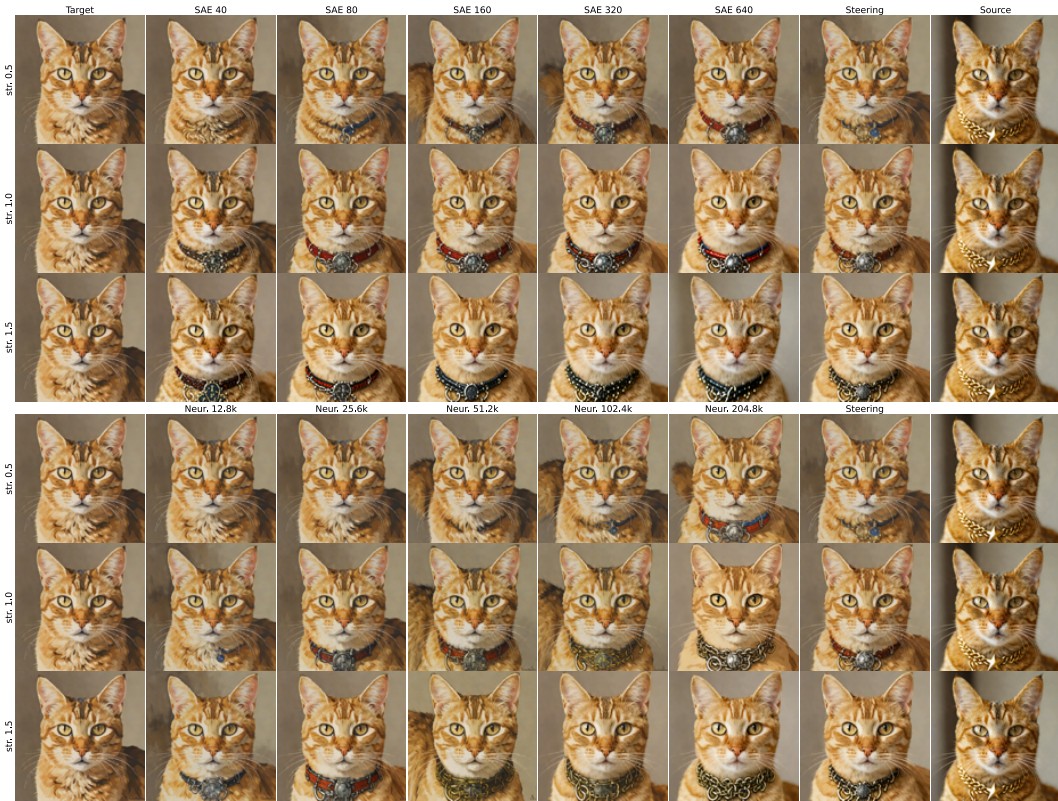

Figure 27: SDXL Turbo 4-step interventions; Example for edit category 2: "add object". Original prompt (target): "a cat", edit prompt (source): "a cat with a gold chain and a star on its head". Source and target refers to from where we extract features (source) and where we insert them (target). Grounded SAM2 masks used to collect the features are not shown but in this example they would select the cat's gold chain from the source forward and also use the same area in the target forward pass.

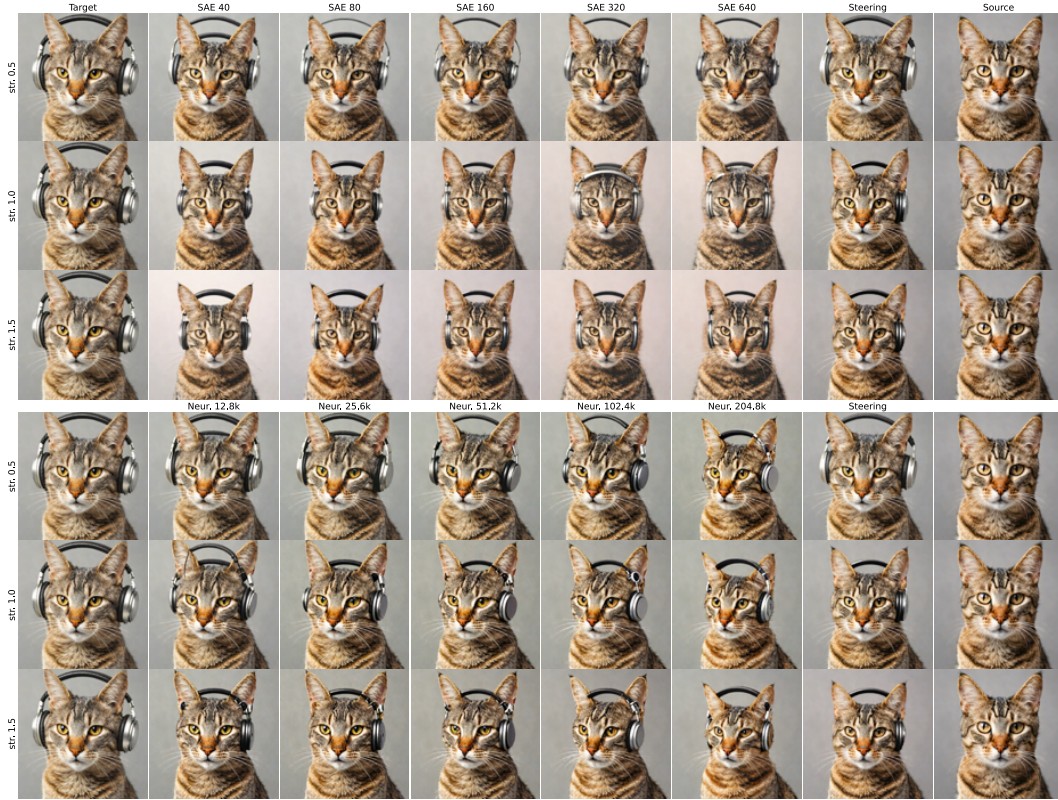

Figure 28: SDXL Turbo 4-step interventions; Example for edit category 3: "delete object". Original prompt (target): "a cat wearing headphones on a gray background", edit prompt (source): "a cat on a gray background". Source and target refers to from where we extract features (source) and where we insert them (target). Grounded SAM2 masks used to collect the features are not shown but in this example they would select the headphones in the target forward and the same area in the source forward pass. This example showcases a frequent failure mode of our intervention where the deleted object (re)appears in a different location in the image.

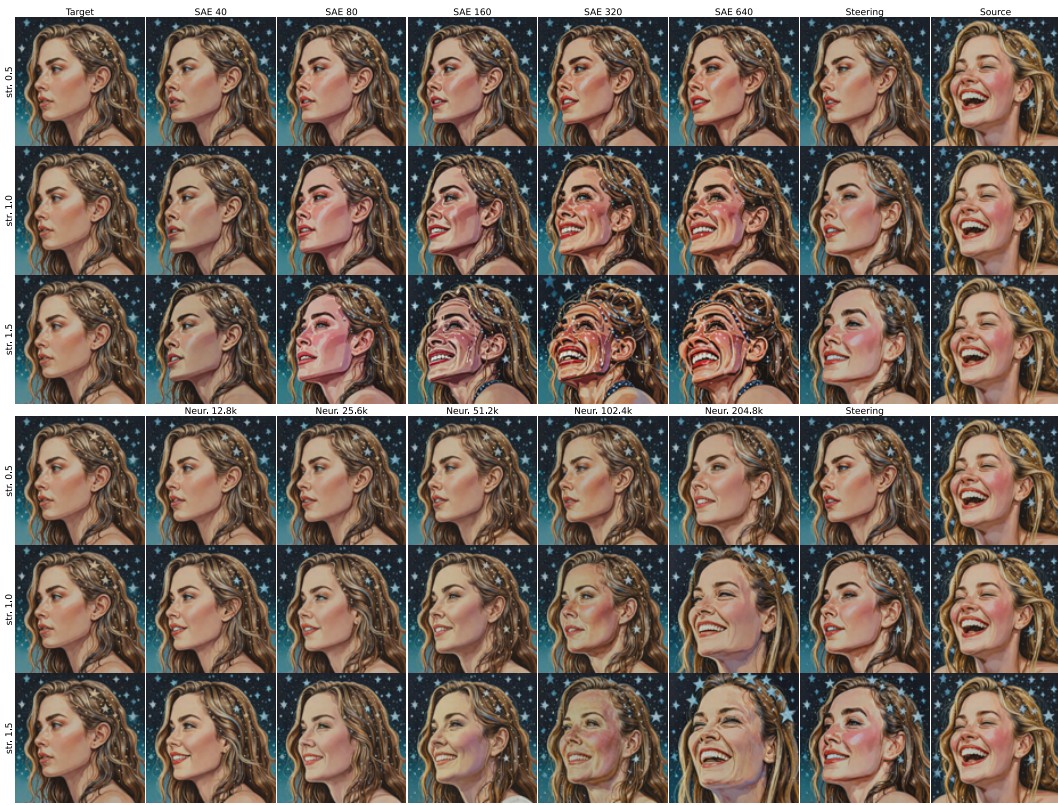

Figure 29: SDXL Turbo 4-step interventions; Example for edit category 4: "change content". Original prompt (target): "a detailed oil painting of a calm beautiful woman with stars in her hair", edit prompt (source): "a detailed oil painting of a laughing beautiful woman with stars in her hair". Source and target refers to from where we extract features (source) and where we insert them (target). Grounded SAM2 masks used to collect the features are not shown but in this example they would select the woman's face in both forward passes.

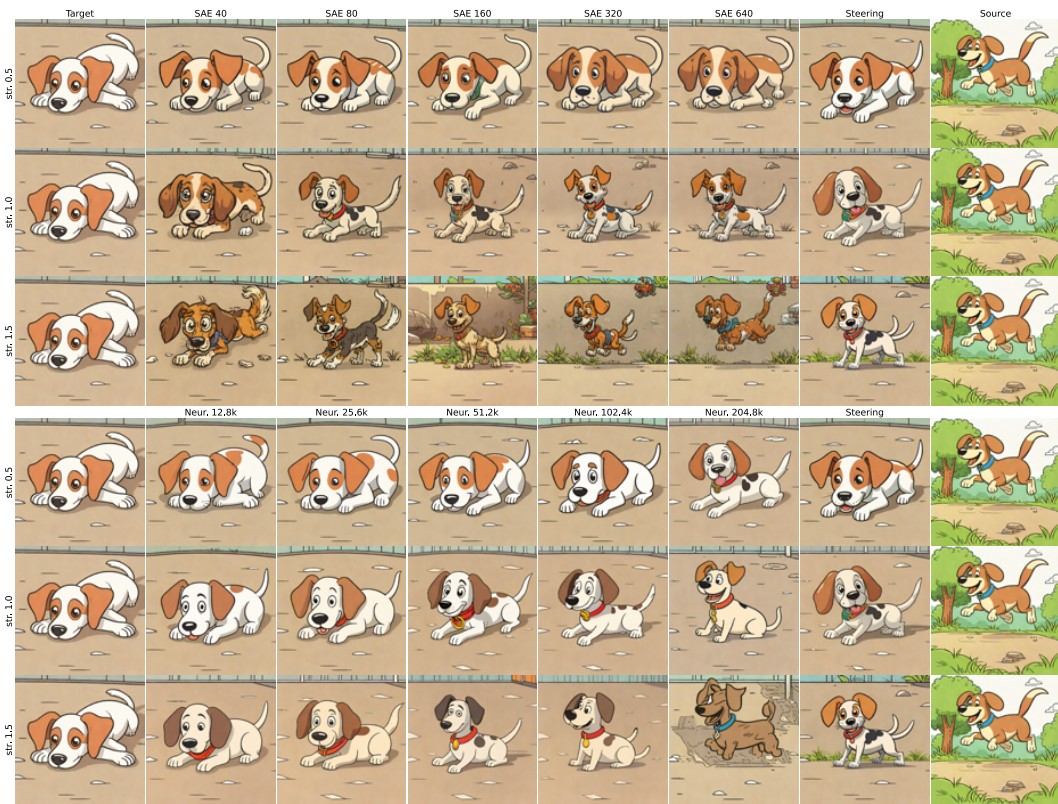

Figure 30: SDXL Turbo 4-step interventions; Example for edit category 5: "change pose". Original prompt (target): "a cartoon dog laying down on the ground", edit prompt (source): "a cartoon dog jumping up from the ground". Source and target refers to from where we extract features (source) and where we insert them (target). Grounded SAM2 masks used to collect the features are not shown but in this example they would select the dogs in both forward passes.

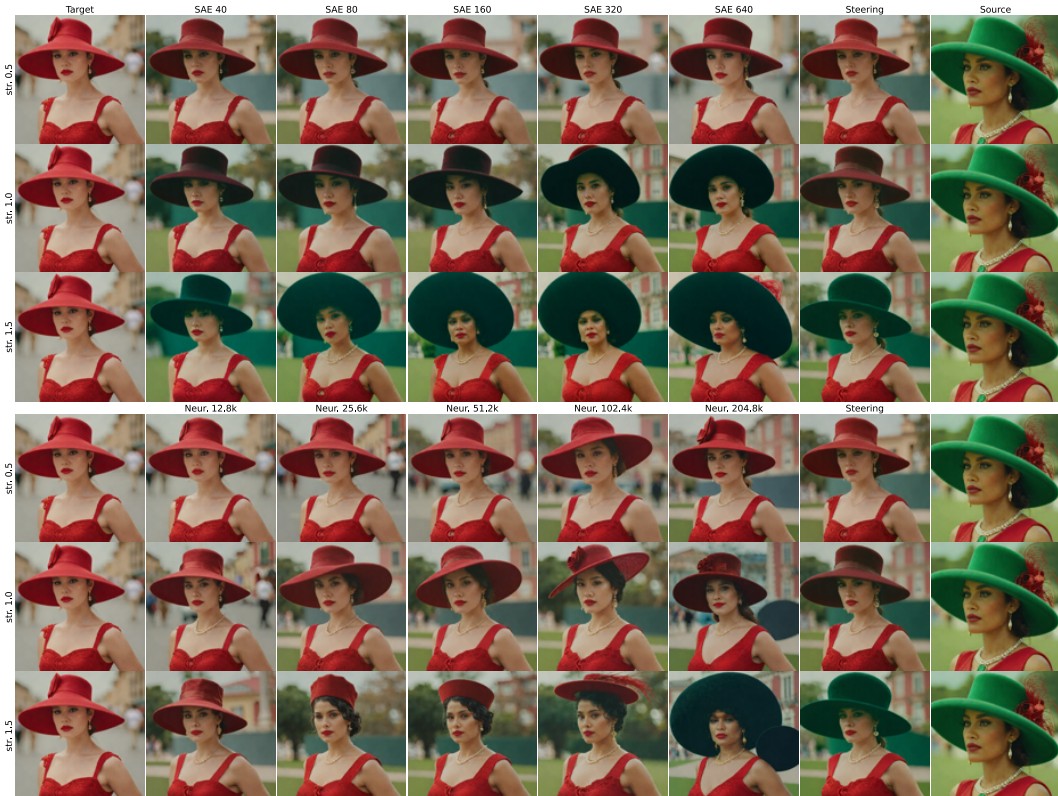

Figure 31: SDXL Turbo 4-step interventions; Example for edit category 6: "change color". Original prompt (target): "a woman wearing a red hat and a red dress", edit prompt (source): "a woman wearing a green hat and a red dress". Source and target refers to from where we extract features (source) and where we insert them (target). Grounded SAM2 masks used to collect the features are not shown but in this example they would select the hats in both forward passes.

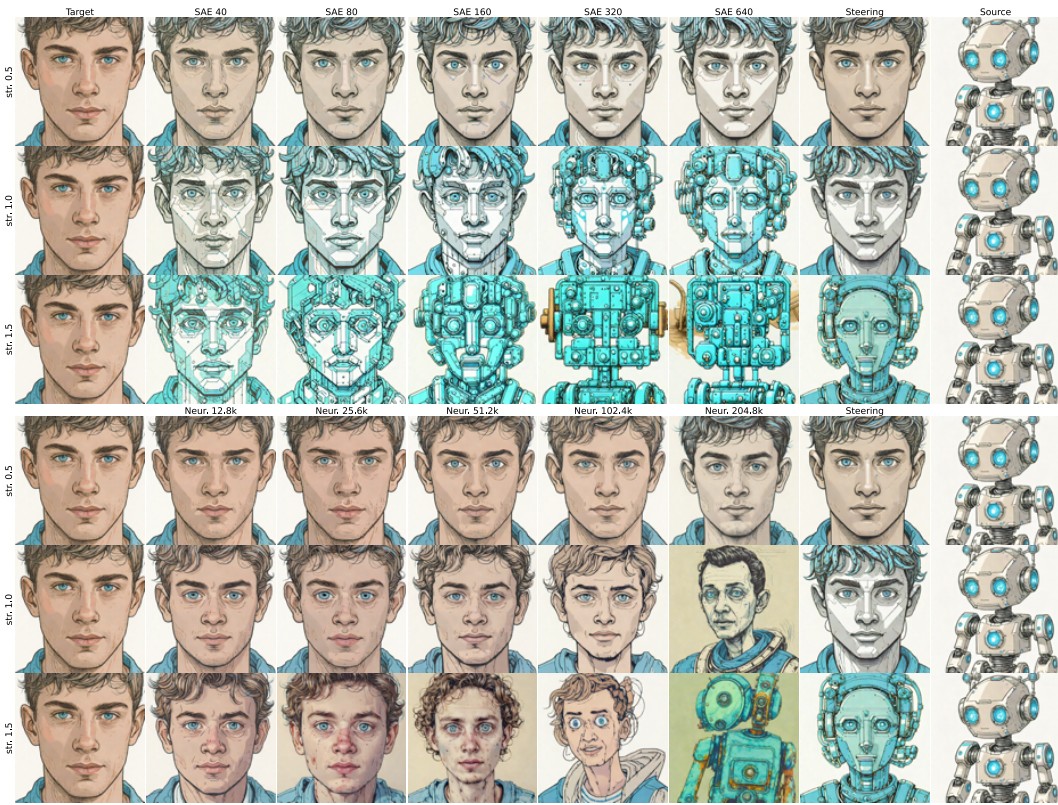

Figure 32: SDXL Turbo 4-step interventions; Example for edit category 7: "change material". Original prompt (target): "a drawing of a young man with blue eyes", edit prompt (source): "a drawing of a young robot with blue eyes". Source and target refers to from where we extract features (source) and where we insert them (target). Grounded SAM2 masks used to collect the features are not shown but in this example they would select the face of the man in the target and the robot in the source forward pass.

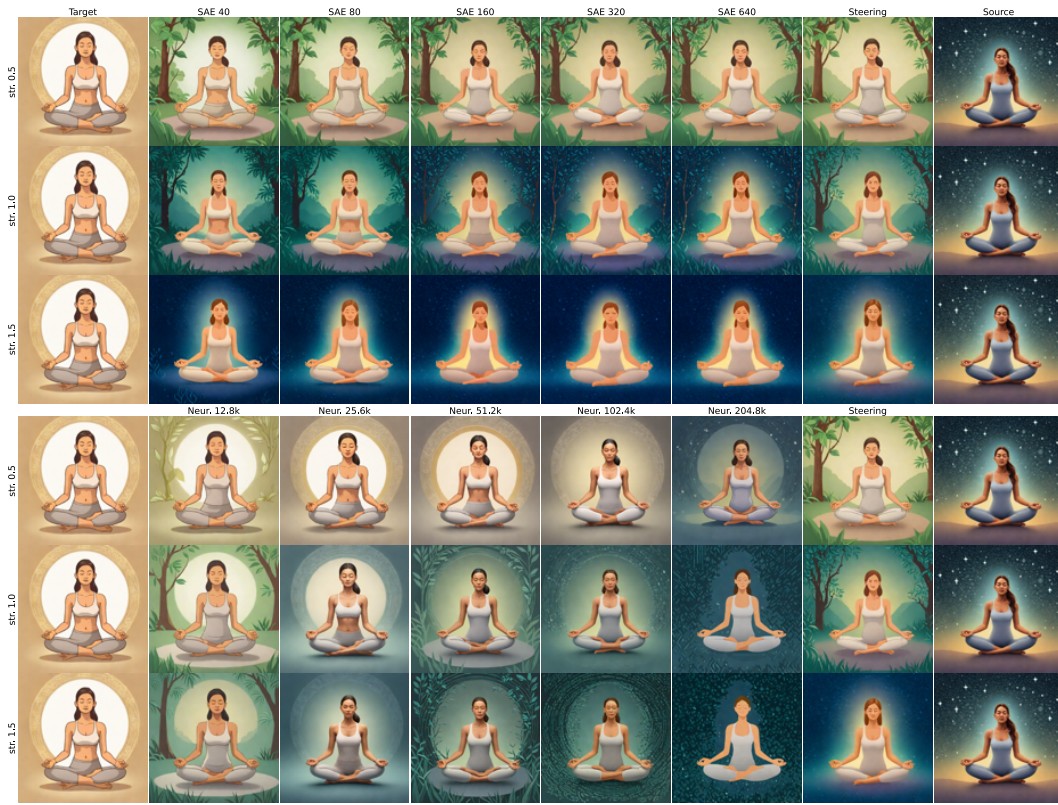

Figure 33: SDXL Turbo 4-step interventions; Example for edit category 8: "change background". Original prompt (target): "illustration of a woman meditating in a yoga pose", edit prompt (source): "illustration of a woman meditating in a yoga pose in the sky with stars". Source and target refers to from where we extract features (source) and where we insert them (target). Grounded SAM2 masks used to collect the features are not shown but in this example they would select the backgrounds in both forward passes.

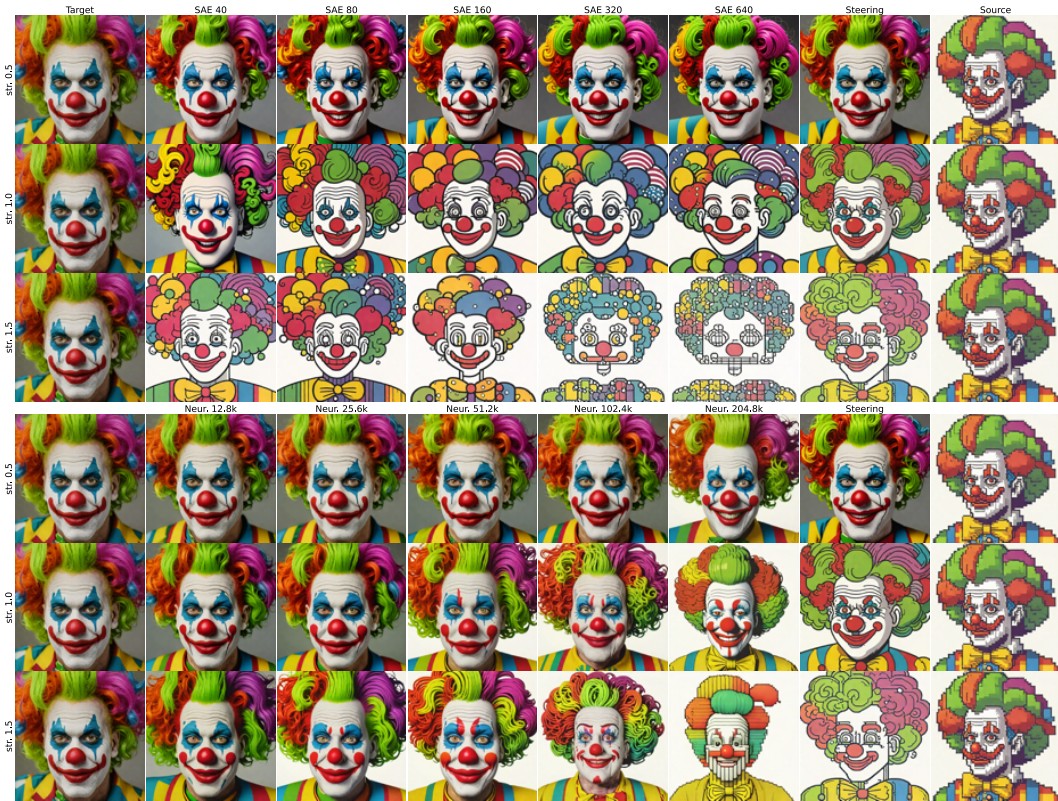

Figure 34: SDXL Turbo 4-step interventions; Example for edit category 9: "change style". Original prompt (target): "a photograph a clown with colorful hair", edit prompt (source): "a clown in pixel art style with colorful hair". Source and target refers to from where we extract features (source) and where we insert them (target). In this edit category we used features from the entire spatial grid. From this example it becomes clear that for this edit category we should select fewer features and probably a lower strength.

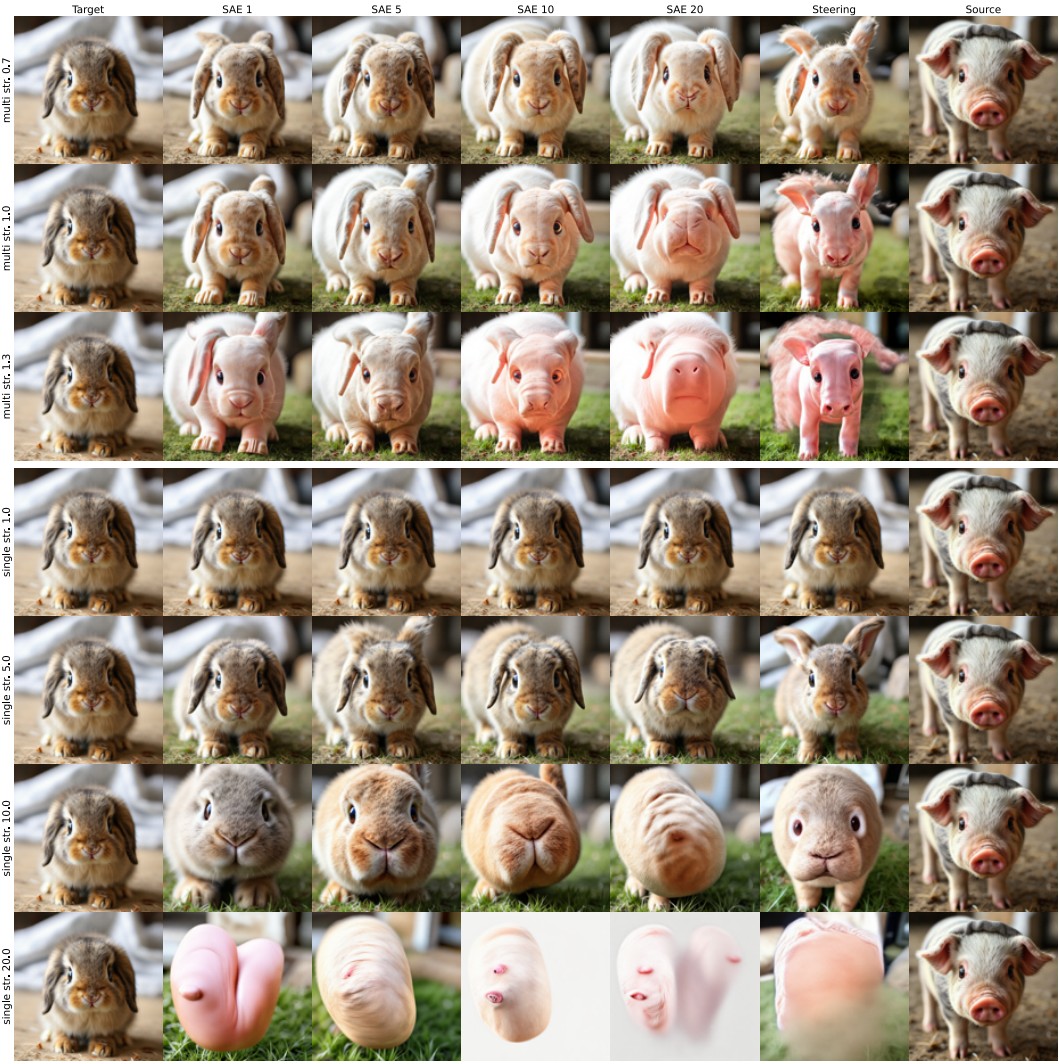

Figure 35: FLUX Schnell 1-step interventions; Example for edit category 1: "change object". Original prompt (target): "a cute little bunny with big eyes", edit prompt (source): "a cute little pig with big eyes". Source and target refers to from where we extract features (source) and where we insert them (target). Grounded SAM2 masks used to collect the features are not shown but in this example they would select the entire foreground objects respectively. The y-labels indicate interventions strength and whether single- or multi-layer interventions are used. The column titles indicate the intervention types and numbers of features transported.

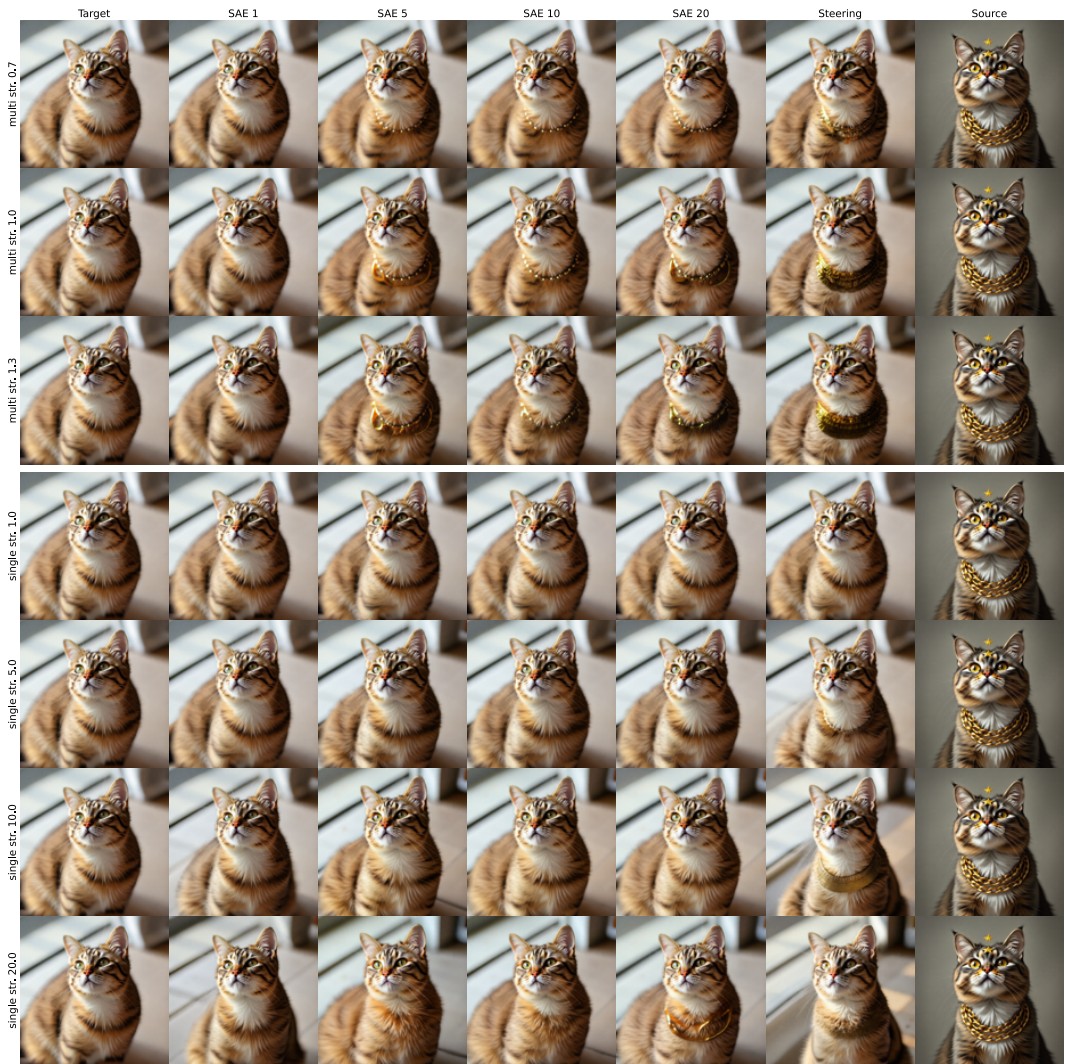

Figure 36: FLUX Schnell 1-step interventions; Example for edit category 2: "add object". Original prompt (target): "a cat", edit prompt (source): "a cat with a gold chain and a star on its head". Source and target refers to from where we extract features (source) and where we insert them (target). Grounded SAM2 masks used to collect the features are not shown but in this example they would select the cat's gold chain from the source forward and also use the same area in the target forward pass. The y-labels indicate interventions strength and whether single- or multi-layer interventions are used. The column titles indicate the intervention types and numbers of features transported.

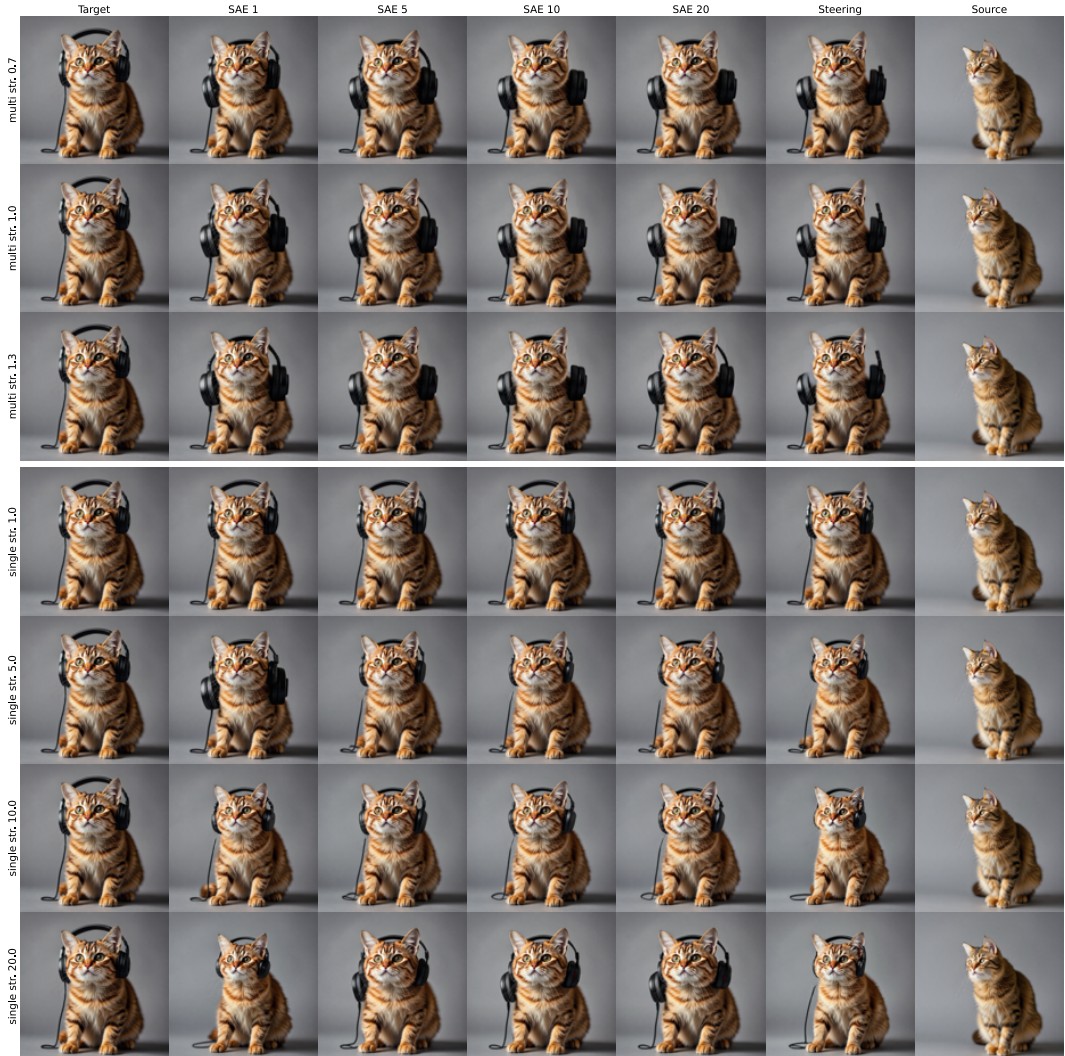

Figure 37: FLUX Schnell 1-step interventions; Example for edit category 3: "delete object". Original prompt (target): "a cat wearing headphones on a gray background", edit prompt (source): "a cat on a gray background". Source and target refers to from where we extract features (source) and where we insert them (target). Grounded SAM2 masks used to collect the features are not shown but in this example they would select the headphones in the target forward and the same area in the source forward pass. This example showcases a frequent failure mode of our intervention where the deleted object (re)appears in a different location in the image. The y-labels indicate interventions strength and whether single- or multi-layer interventions are used. The column titles indicate the intervention types and numbers of features transported.

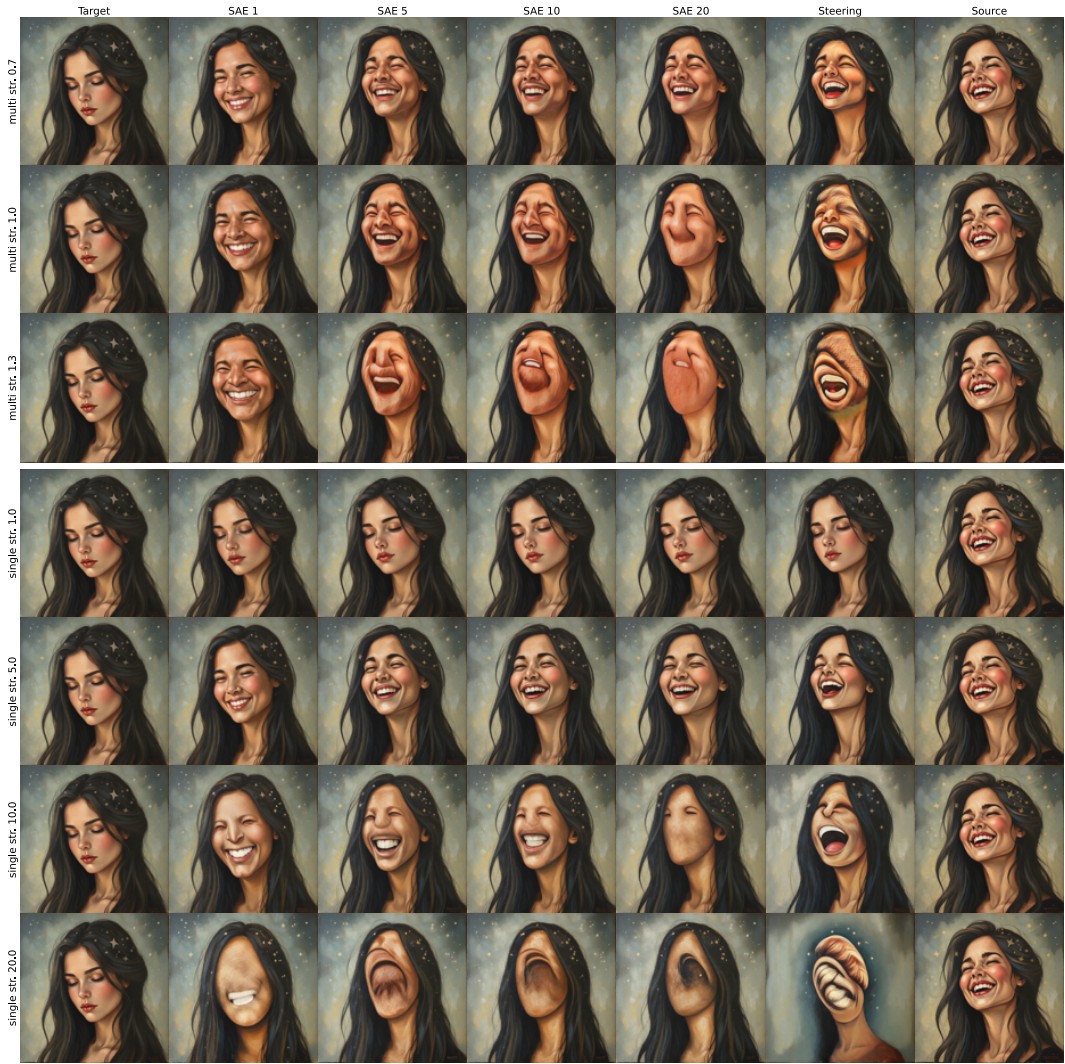

Figure 38: FLUX Schnell 1-step interventions; Example for edit category 4: "change content". Original prompt (target): "a detailed oil painting of a calm beautiful woman with stars in her hair", edit prompt (source): "a detailed oil painting of a laughing beautiful woman with stars in her hair". Source and target refers to from where we extract features (source) and where we insert them (target). Grounded SAM2 masks used to collect the features are not shown but in this example they would select the woman's face in both forward passes. The y-labels indicate interventions strength and whether single- or multi-layer interventions are used. The column titles indicate the intervention types and numbers of features transported.

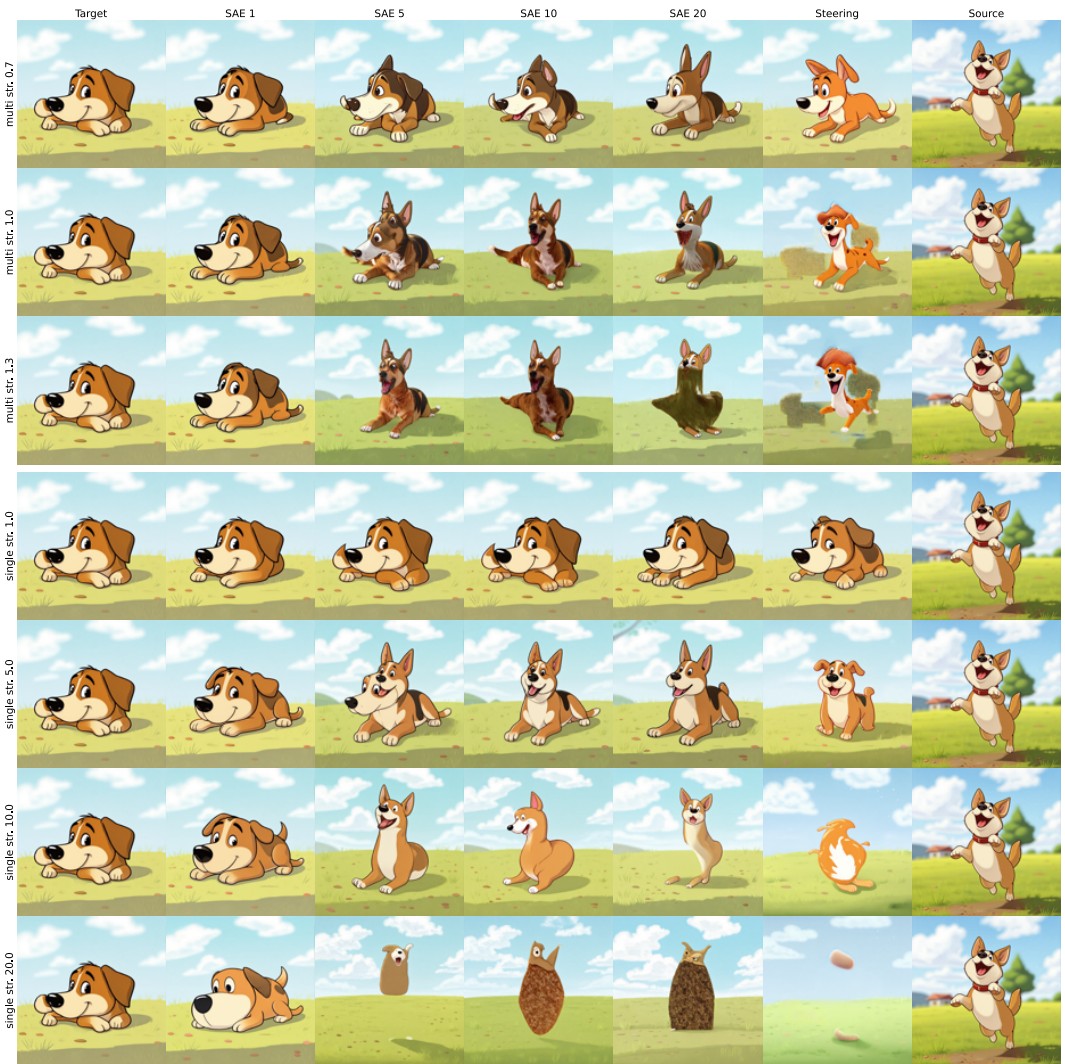

Figure 39: FLUX Schnell 1-step interventions; Example for edit category 5: "change pose". Original prompt (target): "a cartoon dog laying down on the ground", edit prompt (source): "a cartoon dog jumping up from the ground". Source and target refers to from where we extract features (source) and where we insert them (target). Grounded SAM2 masks used to collect the features are not shown but in this example they would select the dogs in both forward passes. The y-labels indicate interventions strength and whether single- or multi-layer interventions are used. The column titles indicate the intervention types and numbers of features transported.

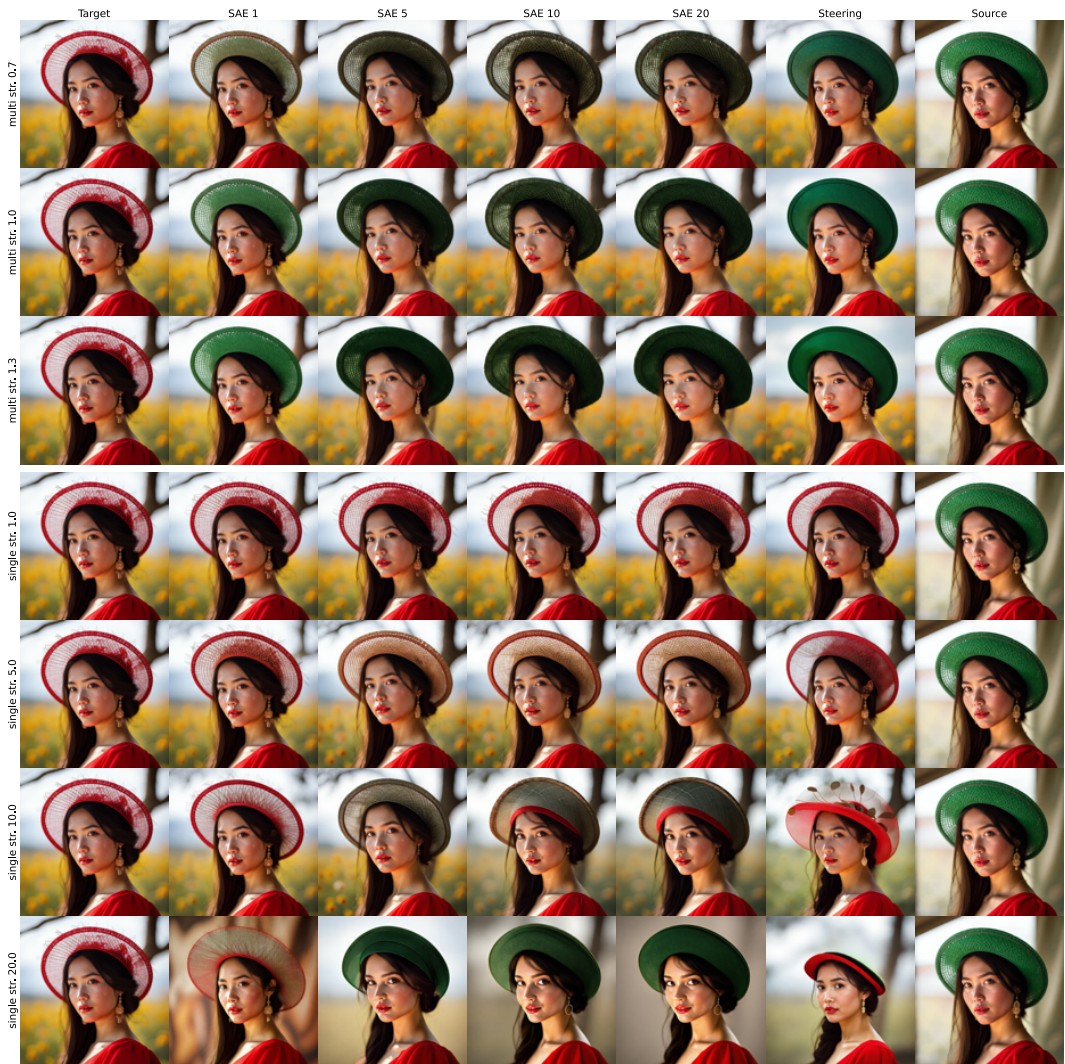

Figure 40: FLUX Schnell 1-step interventions; Example for edit category 6: "change color". Original prompt (target): "a woman wearing a red hat and a red dress", edit prompt (source): "a woman wearing a green hat and a red dress". Source and target refers to from where we extract features (source) and where we insert them (target). Grounded SAM2 masks used to collect the features are not shown but in this example they would select the hats in both forward passes. The y-labels indicate interventions strength and whether single- or multi-layer interventions are used. The column titles indicate the intervention types and numbers of features transported.

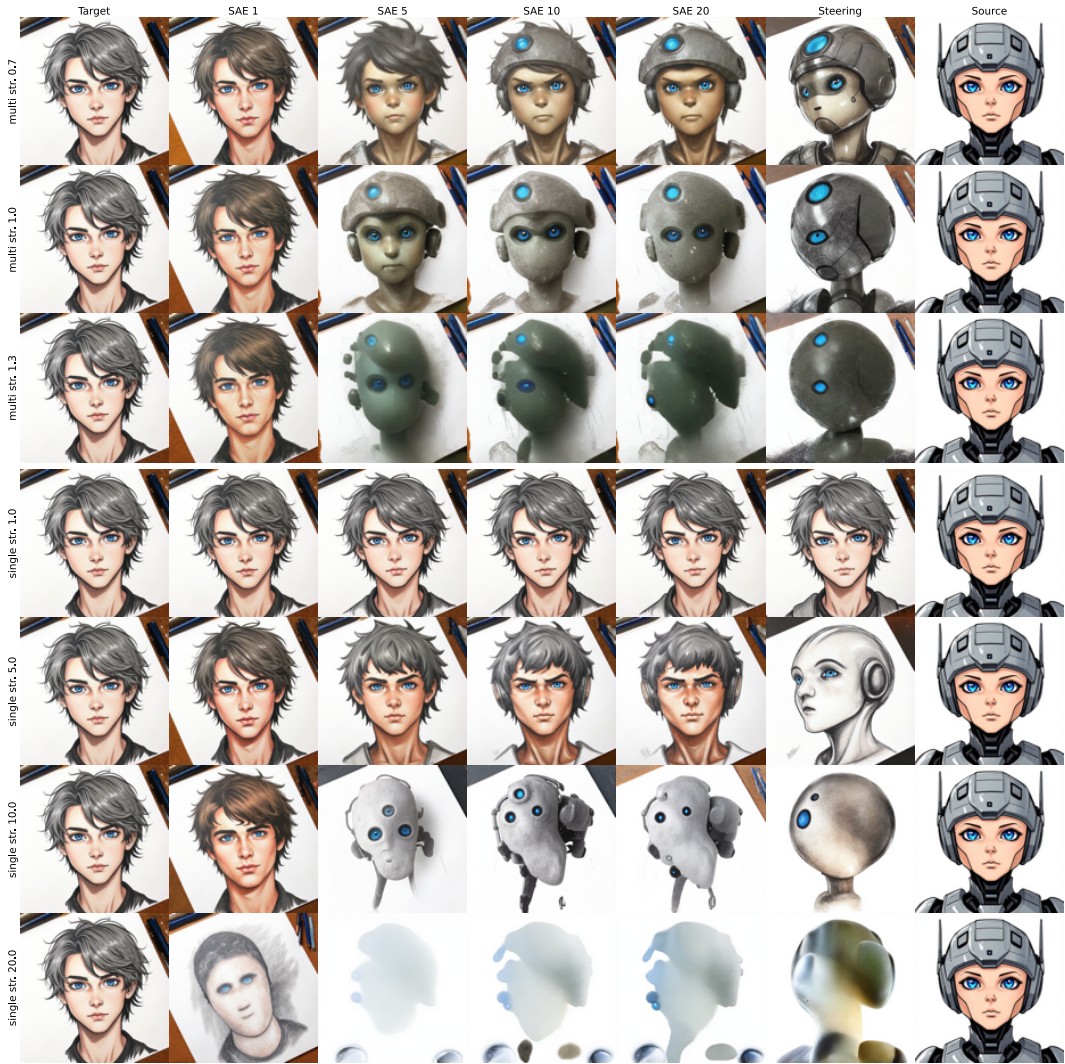

Figure 41: FLUX Schnell 1-step interventions; Example for edit category 7: "change material". Original prompt (target): "a drawing of a young man with blue eyes", edit prompt (source): "a drawing of a young robot with blue eyes". Source and target refers to from where we extract features (source) and where we insert them (target). Grounded SAM2 masks used to collect the features are not shown but in this example they would select the face of the man in the target and the robot in the source forward pass. The y-labels indicate interventions strength and whether single- or multi-layer interventions are used. The column titles indicate the intervention types and numbers of features transported.

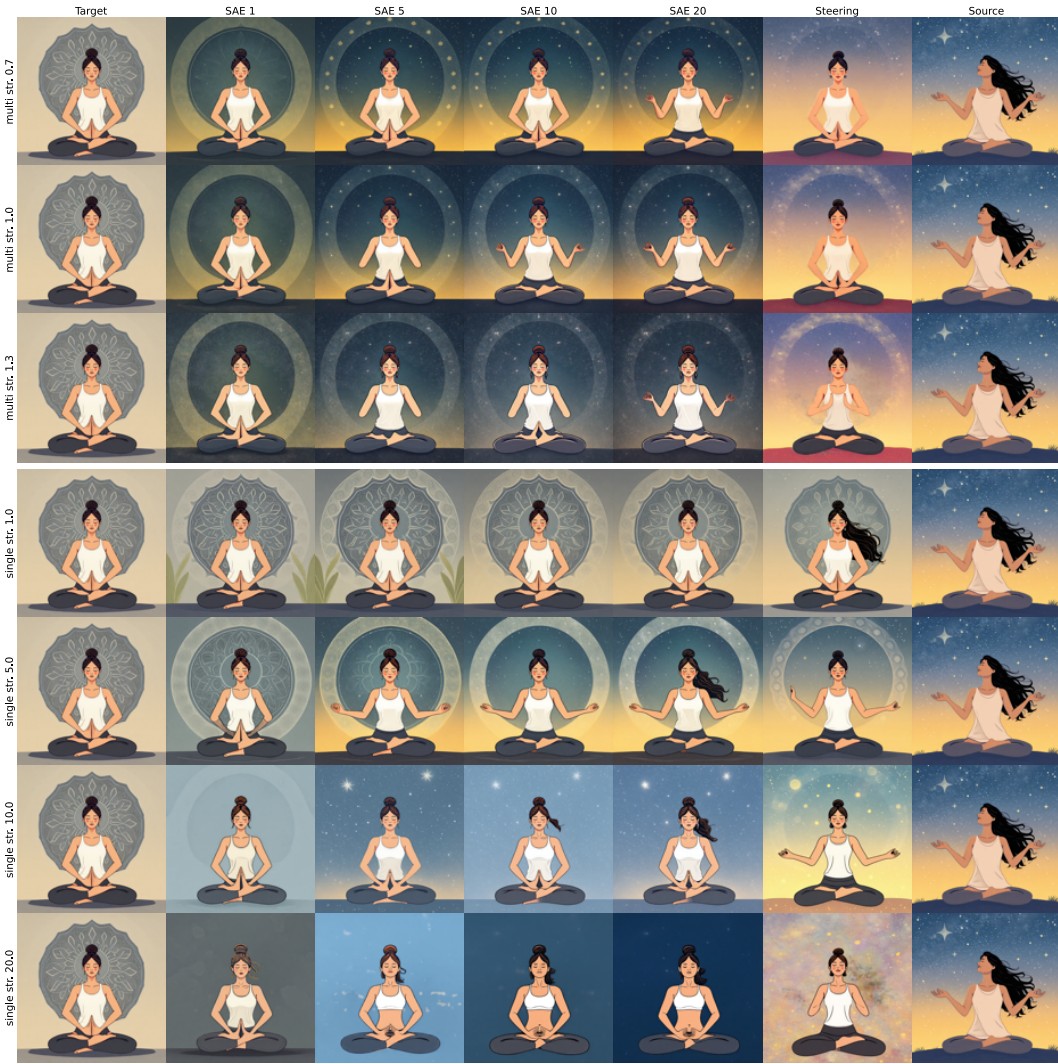

Figure 42: FLUX Schnell 1-step interventions; Example for edit category 8: "change background". Original prompt (target): "illustration of a woman meditating in a yoga pose", edit prompt (source): "illustration of a woman meditating in a yoga pose in the sky with stars". Source and target refers to from where we extract features (source) and where we insert them (target). Grounded SAM2 masks used to collect the features are not shown but in this example they would select the backgrounds in both forward passes. The y-labels indicate interventions strength and whether single- or multi-layer interventions are used. The column titles indicate the intervention types and numbers of features transported.

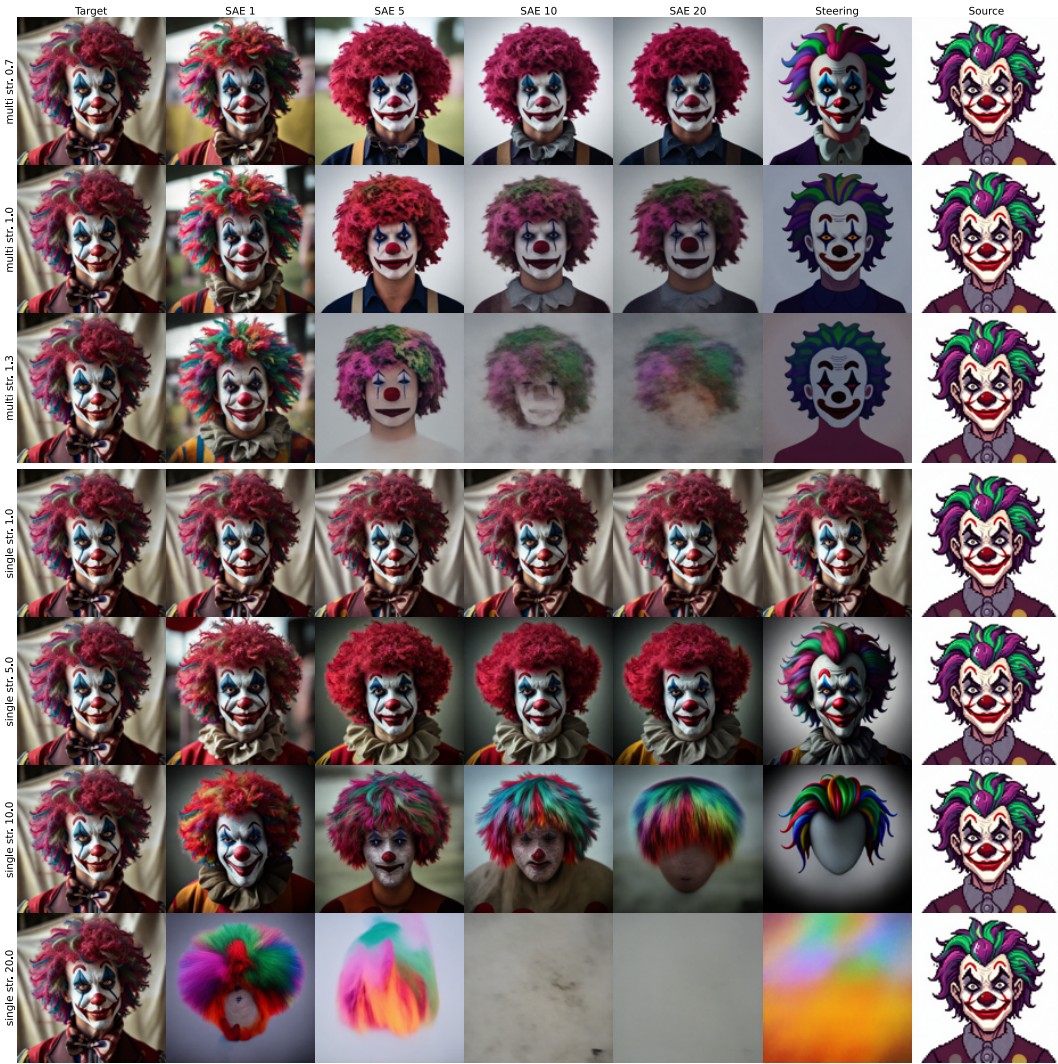

Figure 43: FLUX Schnell 1-step interventions; Example for edit category 9: "change style". Original prompt (target): "a photograph a clown with colorful hair", edit prompt (source): "a clown in pixel art style with colorful hair". Source and target refers to from where we extract features (source) and where we insert them (target). In this edit category we used features from the entire spatial grid. From this example it becomes clear that for this edit category we should select fewer features and probably a lower strength. The y-labels indicate interventions strength and whether single- or multi-layer interventions are used. The column titles indicate the intervention types and numbers of features transported.

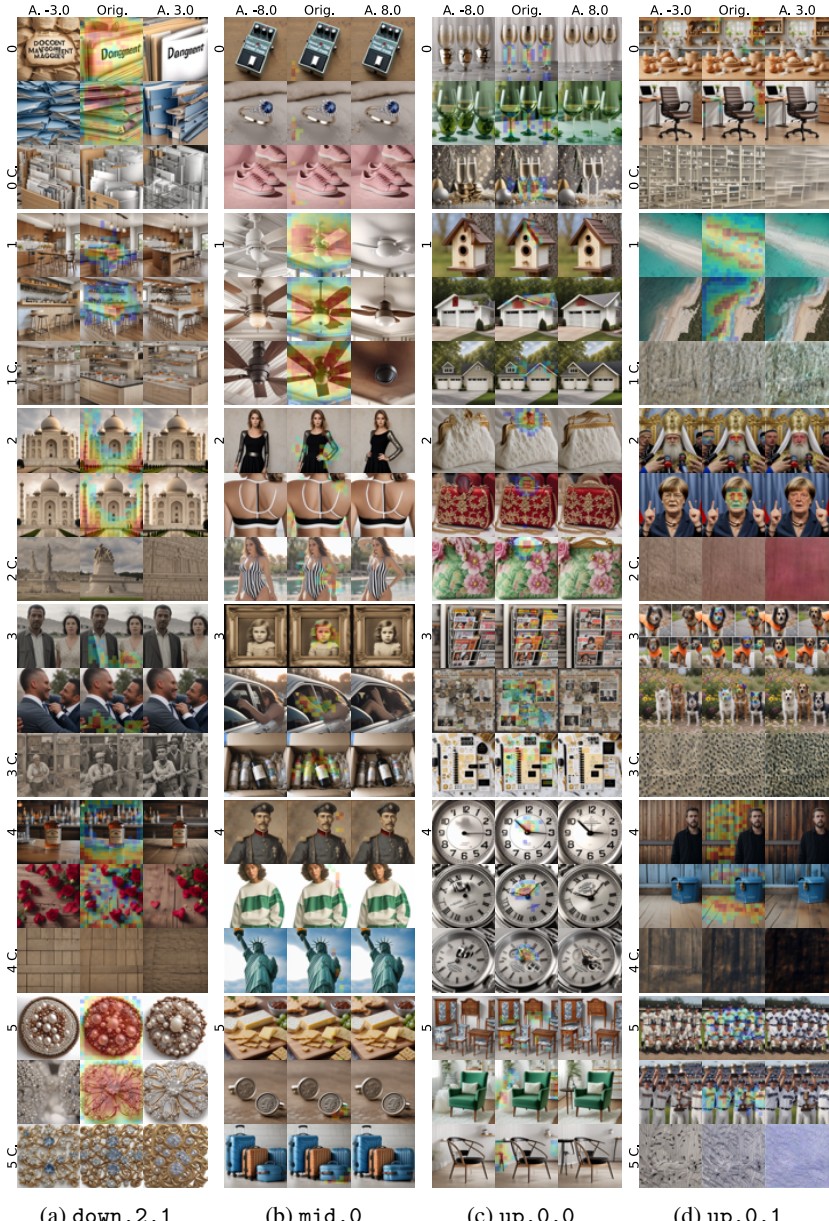

(a) down.2.1          (b) mid.0          (c) up.0.0          (d) up.0.1

Figure 44: We visualize 6 features for down.2.1 (a), mid.0 (b), up.0.0, and up.0.1. We use three columns for each transformer block and three rows for each feature. For down.2.1 and up.0.1 we visualize the two samples from the top 5% quantile of activating dataset examples (middle) together a feature ablation (left) and a feature enhancement (right), and, activate the feature on the empty prompt with $\gamma = 0.5, 1, 2$ from left to right. For mid.0 and up.0.0 we display three samples with ablation and enhancement. Captions are in Table 6. *These results are from our first working SAE's with $k = 10$ and $n_f = 5120$.*

Table 6: Prompts for the top 5% quantile examples in Fig. 44

| Block | Feature | Prompt |
|---|---|---|
| down.2.1 | 0 | A file folder with the word document management on it. |
| | 0 | Two blue folders filled with dividers. |
| | 1 | A kitchen with an island and bar stools. |
| | 1 | An unfinished bar with stools and a wood counter. |
| | 2 | The Taj Mahal, or a white marble building in India. |
| | 2 | The Taj Mahal, or a white marble building in India. |
| | 3 | A man and woman standing next to each other. |
| | 3 | Two men in suits hugging each other outside. |
| | 4 | An old Forester whiskey bottle sitting on top of a wooden table. |
| | 4 | Red roses and hearts on a wooden table. |
| | 5 | A beaded brooch with pearls and copper. |
| | 5 | An image of a brooch with diamonds. |
| mid.0 | 0 | The Boss TS-3W pedal has an electronic tuner. |
| | 0 | An engagement ring with blue sapphire and diamonds. |
| | 0 | The women's pink sneaker is shown. |
| | 1 | A white ceiling fan with three blades. |
| | 1 | A ceiling fan with three blades and a light. |
| | 1 | The ceiling fan is dark brown and has two wooden blades. |
| | 2 | The black dress is made from knit and has metallic sleeves. |
| | 2 | The back view of a woman wearing a black and white sports bra. |
| | 2 | The woman is wearing a striped swimsuit. |
| | 3 | An old-fashioned photo frame with a little girl on it. |
| | 3 | The woman is sitting in her car with her head down. |
| | 3 | The contents of an empty bottle in a box. |
| | 4 | An old painting of a man in uniform. |
| | 4 | The model wears an off-white sweatshirt with green panel. |
| | 4 | The Statue of Liberty stands tall in front of a blue sky. |
| | 5 | Cheese and crackers on a cutting board. |
| | 5 | Two cufflinks with coins on them. |
| | 5 | Three pieces of luggage are shown in blue. |
| up.0.0 | 0 | Three wine glasses with gold and silver designs. |
| | 0 | Three green wine glasses sitting next to each other. |
| | 0 | New Year's Eve with champagne, gold, and silver. |
| | 1 | The birdhouse is made from wood and has a brown roof. |
| | 1 | The garage is white with red shutters. |
| | 1 | Two garages with one attached porch and the other on either side. |
| | 2 | An elegant white lace purse with gold clasp. |
| | 2 | The red handbag has gold and silver designs. |
| | 2 | A pink and green floral-colored purse. |
| | 3 | A magazine rack with magazines on it. |
| | 3 | The year-in-review page for this digital scrap. |
| | 3 | The planner sticker kit is shown with gold and black accessories. |
| | 4 | A clock with numbers on the face. |
| | 4 | A silver watch with roman numerals on the face. |
| | 4 | An automatic watch with a silver dial. |
| | 5 | Four pieces of wooden furniture with blue and white designs. |
| | 5 | The green chair is in front of a white rug. |
| | 5 | The wish chair with a black seat. |
| up.0.1 | 0 | The wooden toy kitchen set includes bread, eggs, and flour. |
| | 0 | The office chair is brown and black. |
| | 1 | An aerial view of the white sand and turquoise water. |
| | 1 | An aerial view of the beach and ocean. |
| | 2 | The patriarch of Ukraine is shown speaking to reporters. |
| | 2 | German Chancellor Merkel gestures as she speaks to the media. |
| | 3 | Four pictures showing dogs wearing orange vests. |
| | 3 | Two dogs are standing on the ground next to flowers. |
| | 4 | A man standing in front of a wooden wall. |
| | 4 | A blue mailbox sitting on top of a wooden floor. |
| | 5 | The baseball players are posing for a team photo. |
| | 5 | The baseball players are holding up their trophies. |

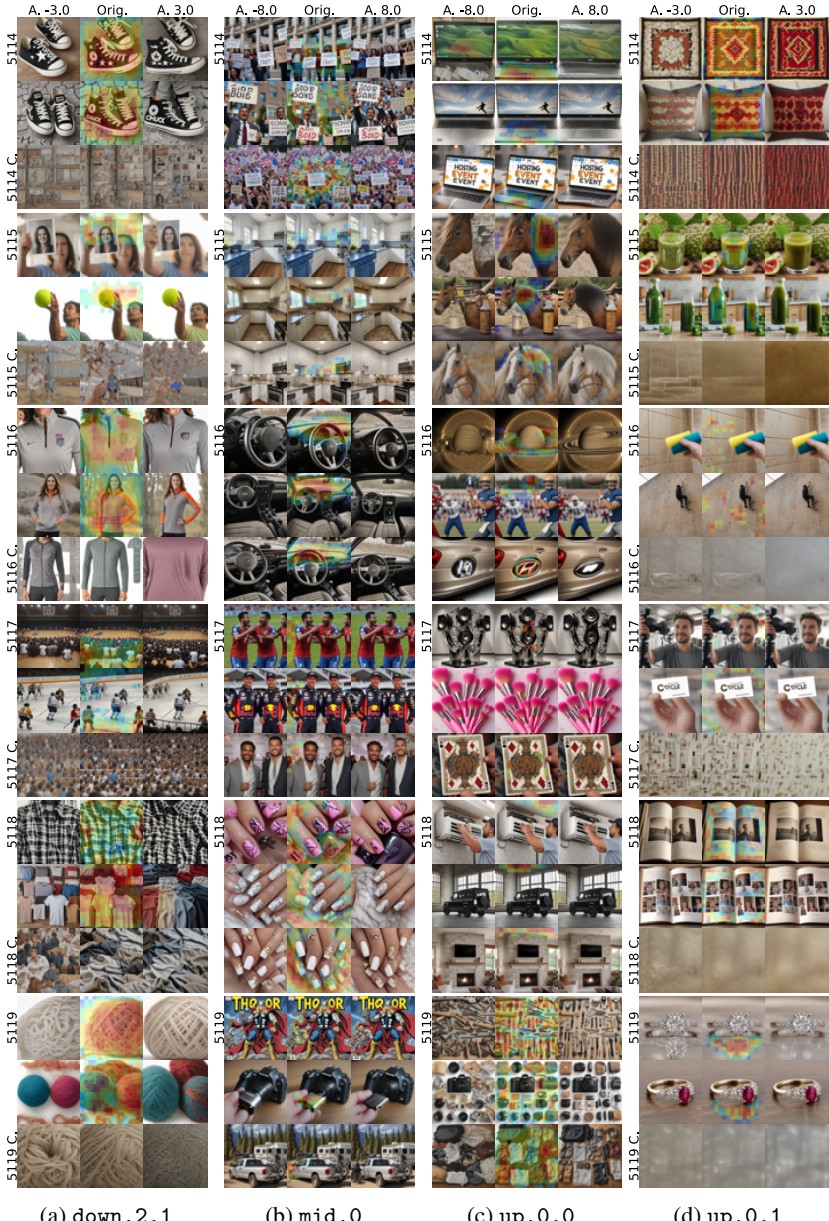

(a) down.2.1     (b) mid.0     (c) up.0.0     (d) up.0.1

Figure 45: We visualize last 6 features for `down.2.1` (a), `mid.0` (b), `up.0.0`, and `up.0.1`. We use three columns for each transformer block and three rows for each feature. For `down.2.1` and `up.0.1` we visualize two samples from the top 5% quantile of activating dataset examples (middle) together a feature ablation (left) and a feature enhancement (right), and, activate the feature on the empty prompt with $\gamma = 0.5, 1, 2$ from left to right. For `mid.0` and `up.0.0` we display three samples with ablation and enhancement. Captions are in Table 7. *These results are from our first working SAE's with $k = 10$ and $n_f = 5120$.*

Table 7: Prompts for the top 5 % quantile examples in Fig. 45

| Block | Feature | Prompt |
|---|---|---|
| down.2.1 | 5114 | Black and white Converse sneakers with the word black star. |
| | 5114 | Black and white Converse sneakers with the word Chuck. |
| | 5115 | A woman holding up a photo of herself. |
| | 5115 | A man holding up a tennis ball in the air. |
| | 5116 | The Nike Women's U.S. Soccer Team DRI-Fit 1/4 Zip Top. |
| | 5116 | The women's gray and orange half-zip sweatshirt. |
| | 5117 | A large group of people sitting in front of a basketball court. |
| | 5117 | Hockey players are playing in an arena with spectators. |
| | 5118 | The black and white plaid shirt is shown. |
| | 5118 | The different colors and sizes of t-shirts. |
| | 5119 | A ball of yarn on a white background. |
| | 5119 | Two balls of colored wool are on the white surface. |
| mid.0 | 5114 | People holding signs in front of a building. |
| | 5114 | Two men dressed in suits and ties are holding up signs. |
| | 5114 | A large group of people holding flags and signs. |
| | 5115 | A kitchen with white cabinets and a blue stove. |
| | 5115 | The kitchen is clean and ready for us to use. |
| | 5115 | A kitchen with white cabinets and stainless steel appliances. |
| | 5116 | The steering wheel and dashboard in a car. |
| | 5116 | The interior of a car with dashboard controls. |
| | 5116 | The dashboard and steering wheel in a car. |
| | 5117 | Three men are celebrating a goal on the field. |
| | 5117 | Two men in Red Bull racing gear standing next to each other. |
| | 5117 | Two men are posing for the camera at an event. |
| | 5118 | Someone is holding up their nail polish with pink and black designs. |
| | 5118 | The nail is very cute and looks great with marble. |
| | 5118 | White stily nails with gold and diamonds. |
| | 5119 | The Mighty Thor comic book. |
| | 5119 | The camera is showing its flash drive. |
| | 5119 | A truck with bikes on the back parked next to a camper. |
| up.0.0 | 5114 | The Acer laptop is open and ready to use. |
| | 5114 | The Lenovo S13 laptop is open and has an image of a person jumping off the keyboard. |
| | 5114 | A laptop with the words Hosting Event on it. |
| | 5115 | A horse with a black nose and brown mane. |
| | 5115 | The horse leather oil is being used to protect horses. |
| | 5115 | An oil painting on a canvas of a horse. |
| | 5116 | The sun is shining brightly over Saturn. |
| | 5116 | A football player throws the ball to another team. |
| | 5116 | Car door light logo sticker for Hyundai. |
| | 5117 | An artistic black and silver sculpture with speakers. |
| | 5117 | The pink brushes are sitting on top of each other. |
| | 5117 | Four kings playing cards in the hand. |
| | 5118 | A man is fixing an air conditioner. |
| | 5118 | The black Land Rover is parked in front of a large window. |
| | 5118 | A flat screen TV mounted on the wall above a fireplace. |
| | 5119 | A table with many different tools on it. |
| | 5119 | A camera with many different items including flash cards, lenses, and other accessories. |
| | 5119 | The contents of an open suitcase and some clothes. |
| up.0.1 | 5114 | An old Navajo rug with multicolored designs. |
| | 5114 | The pillow is made from an old kilim. |
| | 5115 | An image of noni juice with some fruits. |
| | 5115 | A bottle and glass on the counter with green juice. |
| | 5116 | Someone cleaning the shower with a sponge. |
| | 5116 | A man on a skateboard climbing a wall with ropes. |
| | 5117 | A man taking a selfie in front of some camera equipment. |
| | 5117 | A person holding up a business card with the words cycle transportation. |
| | 5118 | Two photos are placed on top of an open book. |
| | 5118 | An open book with pictures of children and their parents. |
| | 5119 | An engagement ring with diamonds on top. |
| | 5119 | An oval ruby and diamond ring. |

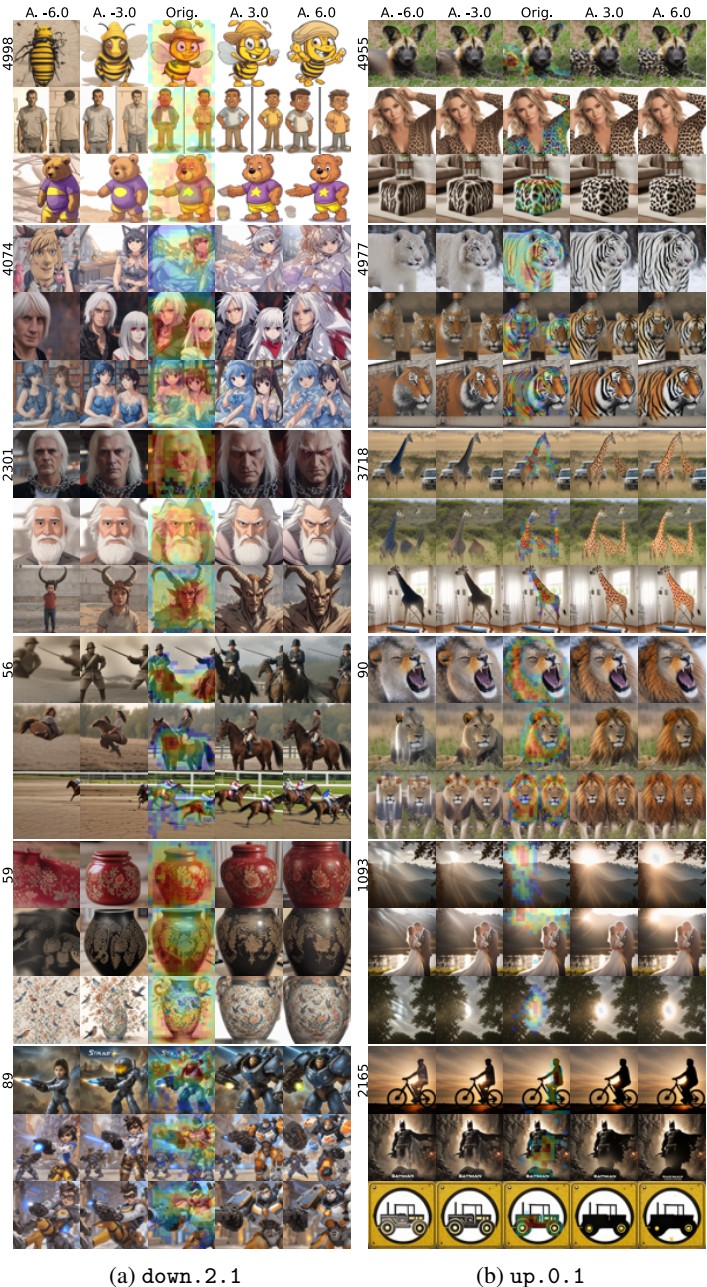

(a) `down.2.1`                                        (b) `up.0.1`

Figure 46: We visualize 6 features for `down.2.1` (a) and `up.0.1` (b). For each feature, we use 5 columns showing ablations (left), activating examples (middle), enhancements (right) and 3 rows with different samples from the top 5% quantile of activating examples. Captions are in Table 8. *These results are from our first working SAE's with $k = 10$ and $n_f = 5120$.*

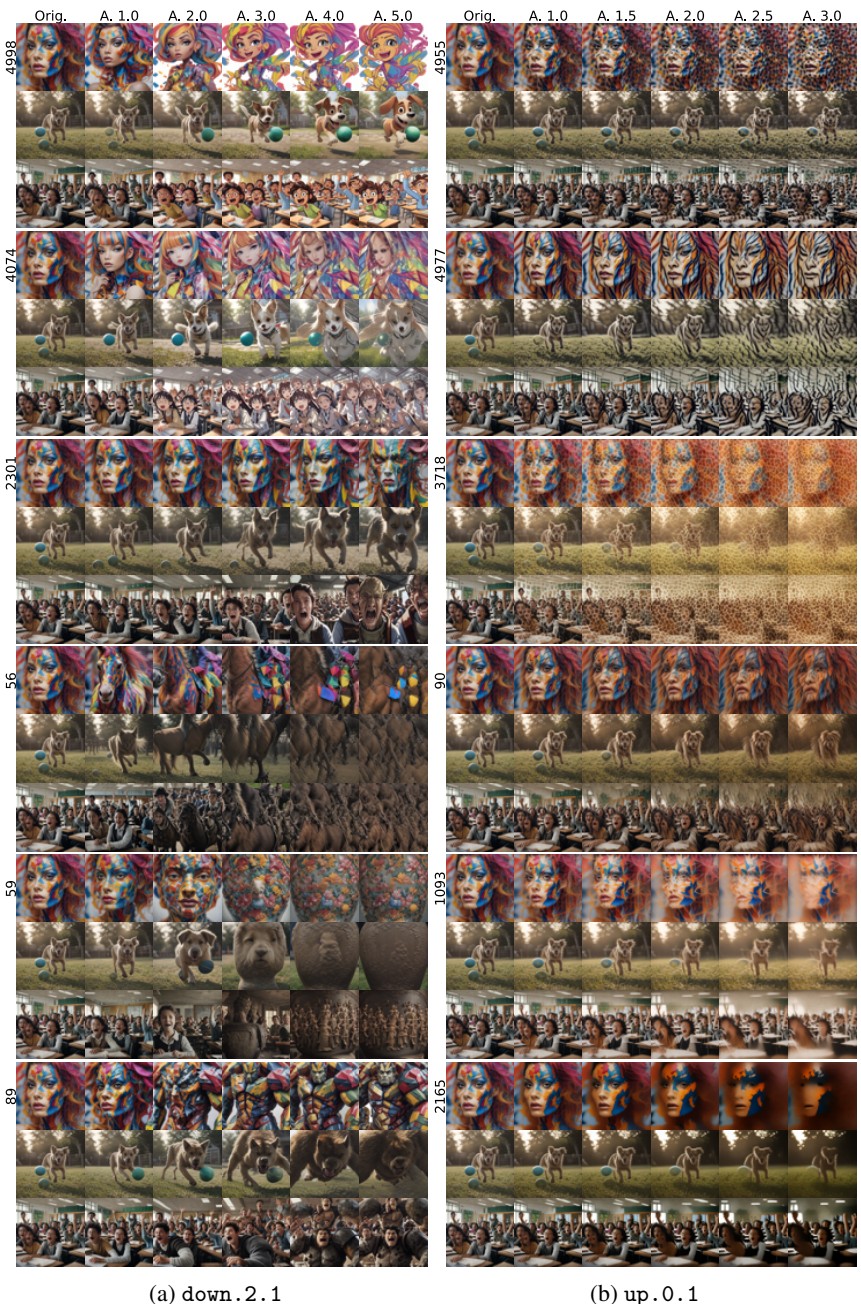

(a) `down.2.1`           (b) `up.0.1`

Figure 47: We turn on the features from Fig. 46 on three unrelated prompts "a photo of a colorful model", "a cinematic shot of a dog playing with a ball", and "a cinematic shot of a classroom with excited students". *These results are from our first working SAE's with $k = 10$ and $n_f = 5120$.*

Table 8: Prompts for the top 5% quantile examples in Fig. 46

| Block | Feature | Prompt |
|-------|---------|--------|
| down.2.1 | 4998 | A cartoon bee wearing a hat and holding something. |
| | 4998 | Two cartoon pictures of the same man with his hands in his pockets. |
| | 4998 | A cartoon bear with a purple shirt and yellow shorts. |
| | 4074 | An anime character with cat ears and a dress. |
| | 4074 | Two anime characters, one with white hair and the other with red eyes. |
| | 4074 | An anime book with two women in blue dresses. |
| | 2301 | A man with white hair and red eyes holding a chain. |
| | 2301 | An animated man with white hair and a beard. |
| | 2301 | The character is standing with horns on his head. |
| | 56 | Two men in uniforms riding horses with swords. |
| | 56 | A woman riding on the back of a brown horse. |
| | 56 | Two jockeys on horses racing down the track. |
| | 59 | A red jar with floral designs on it. |
| | 59 | An old black vase with some design on it. |
| | 59 | A vase with birds and flowers on it. |
| | 89 | StarCraft 2 is coming to the Nintendo Wii. |
| | 89 | Overwatch is coming to Xbox and PS3. |
| | 89 | The hero in Overwatch is holding his weapon. |
| up.0.1 | 4955 | An African wild dog laying in the grass. |
| | 4955 | The woman is posing for a photo in her leopard print top. |
| | 4955 | An animal print cube ottoman with brown and white fur. |
| | 4977 | A white tiger with blue eyes standing in the snow. |
| | 4977 | A bottle and tiger are shown next to each other. |
| | 4977 | A mural on the side of a building with a tiger. |
| | 3718 | Giraffes are standing in the grass near a vehicle. |
| | 3718 | Two giraffes standing next to each other in the grass. |
| | 3718 | A giraffe standing next to an ironing board. |
| | 90 | A lion is roaring its teeth in the snow. |
| | 90 | A lion sitting in the grass looking off into the distance. |
| | 90 | Two lions with flowers on their backs. |
| | 1093 | The sun is shining over mountains and trees. |
| | 1093 | Bride and groom in front of a lake with sun flare. |
| | 1093 | The milky sun is shining brightly over the trees. |
| | 2165 | The silhouette of a person riding a bike at sunset. |
| | 2165 | The Dark Knight rises from his cave in Batman's poster. |
| | 2165 | A yellow sign with black design depicting a tractor. |

