# OpenReview forum: "One-Step is Enough: Sparse Autoencoders for Text-to-Image Diffusion Models"
_NeurIPS.cc/2025/Conference — NeurIPS 2025 poster_

### Official Review · Reviewer_uRaJ · 2025-06-30

**Clarity:** 2
**Significance:** 3
**Originality:** 2
**Rating:** 4
**Confidence:** 4

**Summary:**

The proposed method provides an exploration over the learned features for SDXL and distilled diffusion/flow-matching models, where authors provide experiments mainly for SDXL-Turbo and demonstrate the extension on FLUX-schnell. Authors apply SAEs (sparse autoencoders) to the transformer layers of the models and provide analyses on the discovered representations over the features learned by the targeted transformer blocks. Following the SAE training, experiments analyze the explained variance by the targeted blocks across the timesteps. In addition to the variance analysis, authors also provide experiments on the significance of the discovered features on different types of edits, by utilizing a feature interpolation mechanism across the activations on the transformer blocks. Provided experiments show that features in specified blocks are responsible for certain generation patterns and utilized consistently for specified types of edits from PIE-Bench.

**Questions:**

- How do the authors decide on which transformer blocks to investigate?
- Are the discovered representations be labeled with text instructions, or are they more ambiguous?
- In the set of representations discovered, are there any overlaps, or does all of the representations discovered represent a distinct concepts?

These are the questions that I found to be beneficial for the clarity of the paper, and can improve the significance of the discovered representations. Answers to such questions in the rebuttal period will impact my judgement on the paper in a positive way.

**Ethical Concerns:**

["NO or VERY MINOR ethics concerns only"]

**Final Justification:**

My concerns regarding the novelty of the method and the types of representations discovered has been addressed by the rebuttal of the authors. After reviewing the responses of the other reviewers, I am leaning towards positive.

**Limitations:**

While the authors provide suggestions on possible future work, they have no claims on the limitations if their work. As an example, are there any types of features that require combination of different transformer blocks, that cannot be identified with the current method. Also the usability of these features is another question. While editing results are reported, we have no clue on how successful these edits are or if the representations achieve disentangled edits overall. In the examples provided in Fig. 1, we see local changes such as skin color, but have no idea on more global features can be identified. Such limitations should be addressed in a possible revision.

**Paper Formatting Concerns:**

The paper is in appropriate format, but the figures should be improved for readability. As an example, Figure 5 is very hard to read and understand. There are labels in the figures but they are indistinguishable, which makes it very hard to identify the patterns visually.

**Quality:**

2

**Strengths And Weaknesses:**

**Strengths**
- The paper provides the first study on the application of SAEs for the explainability of features learned by diffusion models and their distilled counterparts.
- The discovered features are investigated in terms of different types of editing tasks and their visualizations are provided to demonstrate their qualitative effectiveness (see Figure 1).
- The paper introduces a feature injection scheme to effectively use them for editing tasks

**Weaknesses**
- The organization of the paper should be improved. The narrative of the paper seems disconnected, which makes it hard to understand.As an example for this, the paper first explains the SAEs trained and the blocks that are selected for it. Then, in the analysis section, the example of feature transport is given. The connection here should be improved for writing, in the current form it feels disconnected (like saying that this is used to transport SAE features in the experiments etc.)
- The readability of the figures should be improved significantly. As an example for this, Figure 5 reports the impact on the editing performance. However, with the labels for SAE features the plot is almost unreadable. The authors shuld consider revising accordingly.
- In lines 117 and 118, the authors say that they select 4 layers for their analysis. However, there is no motivation on how they were selected. The ,methodology should either refer to previous studies on the investihgated blocks or provide a convincing reason on why these methods are investigated.
- While the authors provide editing results with discovered representations, their quality is uncertain. For example, we have no idea on if the representations discovered are repetitve and how unique they are. Furthermore, to show the effectiveness of such representations, the authors should provide qualitative editing results for the categories that the edits are reported.
- To demonstrate if the discovered representations are beneficial for image editing task, a comparison with existing editing methods is essential (for distilled methods, a comparison with [1] can be provided). Conversely, if edits that cannot be performed with the existing methods can be performed with the proposed method, then this should be stated explicitly.
- As we have no explicit information on which types of representations are discovered, the limitations of the method is ambiguous. An analysis on the discovered representations will improve the significance of the methodology.
- With the current state of the paper, it appears that the authors get an existing interpretability method (SAEs) and plug it into an existing method. While this is okay for the explanation of the representations, it makes the novelty of the paper questionable. The authors should provide details on how the adaptation is accomplished (like the selection of the features, how it differs from applying SAEs to LLMs) to further demonstrate the contribution of the paper (or explain their analysis in detail).

[1] Deutch et. al. "TurboEdit: Text-Based Image Editing Using Few-Step Diffusion Models", https://arxiv.org/abs/2408.00735

---

> ### Author Rebuttal · Authors · 2025-07-31
>
> Thank you for your thoughtful and constructive feedback on our paper. We appreciate the time you invested in reviewing our work and the specific suggestions you provided to improve the clarity, organization, and significance of our contributions. We have carefully considered each of your concerns and will address them individually below. Your feedback has been instrumental in helping us strengthen both the presentation and the technical content of our work.
>
> ## Shared response
>
> We recommend having a look at our shared response (contained in our **responses to reviewer SVh8 and siKW**) for our new quantitative results on FLUX.
>
> ## Individual concerns
>
> **Q.1. / W.3.** How do the authors decide on which transformer blocks to investigate?
>
> **R.** We provide detailed analysis in Appendix C "Finding causally influential transformer blocks." We performed ablation studies on each individual transformer block and found that only these 4 blocks have significant effects on generated images when modified. This selection is also supported by prior work (particularly the P+ study) showing that blocks from the center of the U-Net architecture are responsible for high-level concepts like objects, composition, and colors. Also, in the scope of our rebuttal for **fSpm** (their W.2.) we provided additional results for resnet blocks.
>
>
> **Q.2.** Are the discovered representations labeled with text instructions, or are they more ambiguous?
>
> **R.** We used a comprehensive annotation pipeline to label features of one of the blocks (down.2.1), as detailed in Appendix H.1 with examples in Appendix I.3. To annotate features, we prompted GPT-4o with series of images including: (1) images where the feature is inactive, (2) progression of images with increasing average activation values, and (3) images with the highest average activation values. This approach allows us to generate meaningful text labels for discovered representations, making them interpretable and usable for targeted editing applications.
>
>
> **Q.3** In the set of representations discovered, are there any overlaps, or do all of the representations discovered represent distinct concepts?
>
> **R.** Learning features with top-k SAEs involves inherent trade-offs in choosing expansion factors and numbers of nonzero coefficients, which affect feature overlap properties. We observe some degree of feature splitting (where single concepts get divided into multiple features, e.g., separate features for cat ears, eyes, mouth, fur), but this rarely occurs with our chosen hyperparameters for our demo application (k=10, expansion factor=4). We also observe features for similar concepts across different blocks, but these are rarely redundant functionally. Instead, different blocks specialize in different aspects: down.2.1 may place a "cat" in the scene, up.0.0 may modulate details like facial expressions, and up.0.1 may determine color and texture details. Additionally, we find significant feature universality between FLUX and SDXL models, as demonstrated in Figure 1 and our demo applications (Appendix A).
>
>
> **W.1.** The organization and presentation of the paper should be improved.
>
> **R.** Since we found this concern highly important, we provide an overview of the changes to our manuscript that we have performed since the submission in the **shared response**.
>
>
> **W.2.** The readability of the figures should be improved significantly.
>
> **R.** We acknowledge this important concern and will significantly improve the figure readability for the camera-ready version. For Figure 5 specifically, we will implement a cleaner visualization approach that conveys the same information about the Pareto dominance of SAEs versus other methods without overwhelming the reader. Our planned improvements include: color-coding the number of features transported using a clear legend, using different symbols for different steering strengths, and potentially splitting complex plots into multiple subplots for better clarity. We are committed to ensuring all figures are easily interpretable and support the paper's narrative effectively.
>
>
> **W.4.** The quality of the editing results via the discovered features is uncertain.
>
> **R.** We provide comprehensive case studies in our appendices that address these concerns. Specifically, Appendix A contains an installation free demo application to try out our brush tool. Appendix B contains additional results for FLUX. Appendix D contains qualitative results for applying interventions during different ranges of the multi-step setting. Appendix E contains qualitative examples for each edit category in RIEBench. Appendix F presents "Case study: Most active features on a prompt," Appendix G covers "Case study: random features," and Appendix H provides "Quantitative evaluation of the roles of the blocks." These studies demonstrate the quality, uniqueness and diversity of discovered representations. Our analysis shows that while some degree of feature overlap exists (particularly feature splitting with higher expansion factors), the features we discovered with our chosen hyperparameters for the demo application (k=10, expansion factor=4) are largely distinct and functionally non-redundant across different blocks.
>
>
> **W.5.** To demonstrate if the discovered representations are beneficial for image editing tasks, a comparison with existing editing methods is essential.
>
> **R.** To the best of our knowledge, the suggested related works like TurboEdit don't fall within the same category of method. Our interventions currently can only modify images while they are generated, whereas, TurboEdit is meant for editing real images. We are not aware of related works in the same category. Our SAE features and brush tools enable customization of text-to-image generation in ways that are not achievable through prompting-based techniques. While existing methods could potentially be adapted through masking restrictions, the diversity of SAE features learned in an unsupervised manner makes it challenging to match their effects with competing approaches using similar resources. For example, we can locally apply animal skin/fur textures onto human faces without perturbing the underlying facial structure—our brush tools can apply 5 different textures as demonstrated in our supplementary materials. Additionally, our multi-timestep approach with timed-hooks that act only on subsets of timesteps provides novel means of trading off similarity to the original image with edit strength. Real image editing is also possible when combining our SAE features with diffusion inversion techniques, though this area requires further exploration beyond the current scope.
>
>
> **W.6.** As we have no explicit information on which types of representations are discovered, the limitations of the method is ambiguous. An analysis on the discovered representations will improve the significance of the methodology.
>
> **R.** We provide detailed analysis of discovered representations throughout our work (Section 4.2., Appendix F, G, and H). Our key findings include: (1) different transformer blocks specialize in different aspects of image generation (e.g., down.2.1 places objects in scenes, up.0.0 modulates details like facial expressions, up.0.1 determines colors and textures); (2) features remain stable across timesteps, enabling one-step learning to transfer to multi-step generation; (3) we observe feature universality across different model architectures (SDXL and FLUX), as demonstrated in Figure 1 and our demo applications. The discovered representations span various semantic levels from high-level concepts to fine-grained textural details, providing comprehensive control over generation aspects.
>
>
> **W.7.** The novelty of the paper is questionable.
>
> **R.** Our work presents several significant novel contributions beyond simply applying existing methods. Key innovations include: (1) the crucial finding that one-step activation patterns can be applied to multi-step generation without retraining, which is essential for both understanding representation spaces and practical applications; (2) addressing unique challenges in the denoising process that differ substantially from LLM applications, including how activations evolve across different forward passes; (3) developing empty prompt interventions and specialized techniques for classifier-free guidance models; (4) creating image editing methods with semantic segmentation masks and brush tools that are unique to the visual modality and address spatial localization challenges not present in text applications. Furthermore, when applying SAEs to DiTs like FLUX, we found it of key importance to distinguish our "feature extraction" phase from "feature injection", which is in stark contrast to how things are done in LLMs. We find that those two are not strictly connected, and our injection methods represent novel contributions to visual generation control.
>
>
> **Regarding missing limitations**
>
> **R.** We acknowledge this important point and will add a comprehensive limitations section to our revised paper. Key limitations include: (1) our current method focuses on individual block analysis and may miss features that require combinations of multiple transformer blocks; (2) the success rate and disentanglement quality of edits varies depending on the feature type and target image content; (3) while we demonstrate local changes effectively (like skin color modifications), the identification and control of more global compositional features remains challenging; (4) our approach is currently limited to the specific architectures we studied (SDXL and FLUX), and generalization to other diffusion models requires further validation; (5) the hyperparameter choices for SAE training (expansion factor, sparsity level) significantly impact the types and granularity of discovered features, requiring careful tuning for optimal results.

---

> > ### Comment · Reviewer_uRaJ · 2025-08-07
> > **Thank you for the rebuttal**
> >
> > Thanks to the authors for the rebuttal. My concerns regrading the novelty of the method and the types of the representations discovered has been addressed. In the camera ready version of the paper, the authors are strongly encouraged to show the different types of representations visually. After reviewing the other reviews and raised concerns, I am also raising my score. However, as a common concern raised by all of the reviewers, the organization of the paper and figures such as Fig. 5 can make use a revision that will be beneficial for the readability of the paper.

---

> ### Comment · Area_Chair_vDBX · 2025-08-04
> **post-rebuttal comments**
>
> Dear reviewer uRaJ,
>
> As you have seen already, the authors have responded to the questions in your initial review. Can you please share your thoughts post rebuttal, once you've had a chance to see the author responses and also the other reviews?
>
> Best,
> AC

---

### Official Review · Reviewer_siKW · 2025-07-02

**Clarity:** 2
**Significance:** 2
**Originality:** 3
**Rating:** 4
**Confidence:** 3

**Summary:**

This paper explores the use of Sparse Autoencoders (SAEs) for interpreting and controlling text-to-image models, specifically SDXL Turbo. The authors train SAEs on updates made by transformer blocks within SDXL Turbo’s denoising U-net and find that they generalize well to different versions of SDXL models. The learned features from SAEs are interpretable, causally affect the generation process, and reveal block specialization. This is the first study to apply SAEs for interpretability in text-to-image models, demonstrating their potential for understanding and manipulating these models.

**Questions:**

Please refer to the weaknesses.

**Ethical Concerns:**

["NO or VERY MINOR ethics concerns only"]

**Final Justification:**

After reading the author's rebuttal, my concerns have been addressed. I raised my score to ba.

**Limitations:**

yes

**Quality:**

2

**Strengths And Weaknesses:**

Strengths:
1. The proposed method that leverages a sparse autoencoder to interpret and control diffusion models is interesting.
2. The paper is easy to understand. The experiments are detailed and comprehensive.

Weakness:
1. The generalizability of the proposed method in this paper needs further analysis. What conclusions would arise if the original SDXL model or other U-net-based diffusion models, such as SD 1.5, were used? Currently, many diffusion models are based on transformer architectures, so how can this type of analysis be applied to transformer-based models? Analyzing only the SDXL-Turbo model may not provide enough of a technical contribution to be widely applicable.
2. The relationship with other existing works. The existing research[1] also draws similar conclusions, and the author needs to further explain the connections and differences between their work and the existing literature.
3. Can the conclusions drawn from these observations further enhance the generation capabilities of diffusion models or expand their potential applications?
4. Can these properties be leveraged to further enhance these capabilities during the training of diffusion models to improve the generative performance of the diffusion model?
5. The authors should check the paper carefully before submmision to avoid the typo error.

[1] P+: Extended Textual Conditioning in Text-to-Image Generation

---

> ### Author Rebuttal · Authors · 2025-07-31
>
> We thank the reviewer for their thoughtful evaluation of our work and for recognizing the interesting nature of our proposed method and the comprehensive experimental design. We appreciate the constructive feedback! The rebuttal is structured by first responding to common concerns raised by multiple reviewers and subsequently will address each of the raised concerns individually below.
>
> ## Shared response
>
> **Improved writing** We significantly improved their manuscript's organization and clarity through several key updates:
>
> - **Enhanced Structure & Contributions**
>     - Added clear paragraph headings for their four main contributions: SDLens (caching library), SAEPaint (visualization demo), RIEBench (editing benchmark), and findings on generalization across steps/architectures
>     - Updated abstract and section headers to match the new organization
>
> - **Content Consolidation & Clarification**
>     - **Section 3**: Merged spatial activation and sparse feature maps into a single "feature interventions" paragraph; added details on SDXL base interventions and classifier-free guidance handling
>     - **Section 4**: Renamed subsection 4.1 to better reflect the RIEBench benchmark; fixed a bug in equation (9); expanded details on benchmark construction and edit categories
>
> - **New Content & References**
>     - Added a new **Appendix B** specifically covering SDXL Base implementation details, including optimal strategies for feature addition during conditional/unconditional forward passes
>     - Systematically added appendix references throughout the paper to provide readers with additional technical details and examples
>     - Enhanced cross-referencing between sections and appendices (with updated lettering after inserting the new appendix)
>
> **Quantitative results for FLUX** In addition, to address the common concern of overly focusing our paper on SDXL Turbo, we performed an evaluation of FLUX Schnell on our representation-based editing benchmark (RIEBench). We will add these results to the camera-ready version of our paper in case of acceptance. For ease of presentation in the rebuttal format, we averaged across tasks here. We provide mean and Gaussian 95% confidence intervals of LPIPS between our edited-image and original-image CLIP similarity of our edited-image and the edit-prompt averaged over our 9 edit categories of which each contains about 30-50 examples.
>
> | Method | LPIPS Original | CLIP Text Edited |
> |--------|----------------|------------------|
> | sae_5_2 | 0.4436 ± 0.0272 | 0.2549 ± 0.0046 |
> | sae_5_1.5 | 0.3856 ± 0.0254 | 0.2534 ± 0.0046 |
> | sae_5_1.0 | 0.3085 ± 0.0225 | 0.2496 ± 0.0044 |
> | steering_0.4 | 0.4309 ± 0.0252 | 0.2475 ± 0.0041 |
> | steering_0.3 | 0.3712 ± 0.0217 | 0.2444 ± 0.0042 |
> | sae_5_0.5 | 0.1934 ± 0.0180 | 0.2433 ± 0.0043 |
> | steering_0.5 | 0.4686 ± 0.0269 | 0.2431 ± 0.0046 |
> | steering_1.0 | 0.5545 ± 0.0284 | 0.2243 ± 0.0052 |
> | baseline | 0.0000 ± 0.0000 | 0.2398 ± 0.0043 |
>
> As can be seen, our FLUX SAE (hyperparameters: k=20, expansion factor=4) interventions, in which we transport 5 features from the mask within the edit-image into the mask of the original-image with varying strength, dominate the activation steering baseline (higher CLIP similarity with edit-prompt, while achieving lower perturbation of the original-image in terms of LPIPS distance). For reference we also provide a baseline in which we compute LPIPS between original image and itself (= 0) and CLIP similarity between the unmodified original-image and the edit-image.
>
> **Qualitative results and demo** It seems like reviewers did not find many of our provided qualitative results in our supplementary material as well as our installation-free demo in Appendix A. We highly recommend checking out the demo application if time permits it.
>
> ## Individual concerns
>
> Below, we address each of the raised concerns (prefixed with “**W**” when raised in weaknesses and with “**Q**” when raised in questions) individually. Responses are prefixed with “**R**”.
>
> **W.1.** The generalizability of the proposed method in this paper needs further analysis. What conclusions would arise if the original SDXL model or other U-net-based diffusion models, such as SD 1.5, were used? Currently, many diffusion models are based on transformer architectures, so how can this type of analysis be applied to transformer-based models? Analyzing only the SDXL-Turbo model may not provide enough of a technical contribution to be widely applicable.
>
> **R.** We address the generalizability concern through multiple demonstrations in our work. First, in the paper and supplementary materials, we provide qualitative analysis for Flux-schnell (Figure 1 and Appendix B for more details), showing that our approach extends beyond SDXL-Turbo to transformer-based diffusion models. Additionally, we demonstrate that SAEs trained on SDXL-Turbo can be successfully applied to SDXL - the full multistep model - indicating cross-model transferability within the same architecture family. For our SDXL base results consider (Fig. 1 and Fig. 3 in the main paper, as well as Fig. 11, 17, 18, 19, 20, 21, 22, and 23 in the appendix provided in our supplementary material) or check out our installation-free demo application that we provide in Appendix A. While we acknowledge that broader evaluation across different architectures (SD 1.5, various transformer-based models) would strengthen our claims, our current results provide strong evidence for the method's generalizability across both distilled and full models, as well as different architectural paradigms (U-Net vs. transformer-based). Based on the fact that our SAE features transfer from SDXL Turbo to SDXL base without retraining, we are confident that they also transfer to other SDXL finetunes, of which there are about 10.000 on huggingface today 06/31/25.
>
> **W.2.** The relationship with other existing works. The existing research [P+: Extended Textual Conditioning in Text-to-Image Generation] also draws similar conclusions, and the author needs to further explain the connections and differences between their work and the existing literature.
>
> **R.** We discuss this work comprehensively in our "Related Work" section. Here is the key excerpt: "In contrast to these prior works, our approach takes a fundamentally different perspective and does not rely on either handcrafted attributes or prompts. Instead, we introduce a new lens for analyzing SDXL Turbo's transformer blocks, which reveals specialization among the blocks as well. Interestingly, our findings on SDXL Turbo, which is a distilled, few-step diffusion model parallel [56]'s observations, which identify that composition precedes material and style. In SDXL Turbo, this progression occurs across the layers instead of the denoising timestamps." The key distinction is that our method provides interpretation without requiring predefined concepts or prompts, offering a more general and unbiased approach to understanding model behavior.
>
>
> **W.3.** Can the conclusions drawn from these observations further enhance the generation capabilities of diffusion models or expand their potential applications?
>
> **R.** Our approach significantly enhances image editing capabilities through interpretable feature manipulation. We envision a system where users have access to thousands of conceptual "brushes," each corresponding to a specific SAE feature (color, texture, object, concept, detail, expression, illumination, shadow). Users can selectively apply these concepts to specific areas of an image, making targeted modifications with minimal impact on surrounding areas - a level of control not achievable through prompting alone. We provide a demonstration of this tool in Appendix A with an accompanying demo application. Furthermore, our method enables safety-oriented applications, such as modifying text-to-image diffusion models to adhere to safety standards by suppressing features associated with undesirable generations (NSFW, gore, etc.). The related work by Cywinski and Deja ([11] in our paper)  demonstrates this possibility for SD1.5, and our approach extends this capability with more interpretable and controllable mechanisms.
>
>
> **W.4.** Can these properties be leveraged to further enhance these capabilities during the training of diffusion models to improve the generative performance of the diffusion model?
>
> **R.** While our work primarily focuses on model inference and interpretation, these insights hold potential for constructing inherently interpretable architectures. We envision that our SAEs, particularly after labeling specific features, could enable novel distillation techniques that selectively transfer desirable features to student models while excluding undesirable ones. This could lead to more controllable and safer diffusion models from the training stage. However, we acknowledge that such applications represent a significant extension beyond our current scope and would require substantial additional research to validate these possibilities with certainty.
>
>
> **W.5.** The authors should check the paper carefully before submission to avoid the typo error.
>
> **R.** Thank you for bringing this to our attention and for helping maintain high scientific standards. We have thoroughly revised our submission (as mentioned in the **shared response**) and corrected all identified typographical errors. We greatly appreciate your diligence in pointing this out. If you have identified any specific typos, we would be grateful for your feedback so we can cross-reference them with our updated version to ensure complete accuracy.
>
> [11] Bartosz Cywi´ nski and Kamil Deja. Saeuron: Interpretable concept unlearning in diffusion models with sparse autoencoders, 2025

---

> > ### Comment · Reviewer_siKW · 2025-08-05
> >
> > After reading the author's rebuttal, my concerns have been addressed. I will raise my score.

---

> ### Comment · Area_Chair_vDBX · 2025-08-04
> **post-rebuttal comments**
>
> Dear reviewer siKW,
>
> As you have seen already, the authors have responded to the questions in your initial review. Can you please share your thoughts post rebuttal, once you've had a chance to see the author responses and also the other reviews?
>
> Best,
> AC

---

### Official Review · Reviewer_fSpm · 2025-07-03

**Clarity:** 2
**Significance:** 2
**Originality:** 3
**Rating:** 4
**Confidence:** 5

**Summary:**

This paper investigates the application of sparse autoencoders (SAEs) to interpret and manipulate the intermediate representations of text-to-image diffusion models, specifically SDXL Turbo. By training SAEs on the residual updates of transformer blocks in SDXL Turbo’s one-step generation process, the authors successfully decompose complex hidden states into sparse and interpretable features. These features are shown to generalize across multiple denoising steps and even to other models like SDXL Base and FLUX Schnell. Using both qualitative visualizations and a quantitative benchmark adapted from PIEBench, the paper demonstrates that the learned features are not only interpretable but also causally influence the image generation process. Furthermore, the study reveals functional specialization among different transformer blocks, such as composition handling, object editing, and style refinement. This work represents a novel contribution to the mechanistic interpretability of diffusion models and establishes SAEs as a promising tool for understanding and editing their internal computations.

**Questions:**

1) While the proposed method is demonstrated on SDXL Turbo, which is a U-Net-based model, it remains unclear whether the approach can be applied to other one- or few-step generation models with different architectures, such as Diffusion Transformers. Have the authors considered how architectural differences may affect the applicability or reliability of SAE-based interpretability?

2) The paper shows that SAE-learned features can influence the generated image, but it is not entirely clear what kinds of semantic or structural properties these features capture. Beyond the ability to manipulate generation, what specific insights into the model's behavior or representations were gained through this analysis?

3) The caption for Figure 1 is somewhat minimal, especially concerning the Flux-schnell example. What do the numbers below the images represent, and how exactly does the image change as a result of the SAE intervention? It would be helpful to explain whether the SAE was trained using one-step intermediate features and then applied during a four-step or multi-step generation process for Flux-schnell, and how this intervention was performed.

4) The authors state that the learned features “generalize” across different denoising schedules and models (from 1-step SDXL Turbo to 4-step and 20-step SDXL). Could the authors clarify what specifically is being generalized? Does this refer to the preservation of semantic meaning in SAE features, reconstruction quality, or transferability of causal influence?

5) The analysis focuses solely on transformer blocks within the SDXL Turbo U-Net, even though the architecture also includes residual convolutional blocks. Given that convolutional and transformer features are known to exhibit different representational biases (e.g., texture vs. shape), it would be informative to know whether similar interpretability or control can be achieved through SAEs trained on convolutional features. Could the authors comment on why these components were not analyzed?

6) The description of the generalization study (e.g., L133 and Figure 1) lacks sufficient detail. Could the authors explain more clearly how the visual examples were constructed? Specifically, how were SAE features trained in the one-step setting reused during multi-step generation, and how was the intervention performed?

**Ethical Concerns:**

["NO or VERY MINOR ethics concerns only"]

**Final Justification:**

After reviewing the authors’ rebuttal and additional experiments, I have decided to maintain my original score of 4 (Borderline Accept). The paper presents a novel and timely contribution by applying sparse autoencoders (SAEs) to the interpretability of diffusion models, and I appreciate the authors’ detailed and thoughtful responses to my concerns. Below, I summarize my reasoning:
- Evaluation on Diffusion Transformer (DiT) architectures:
The authors have extended their analysis to a DiT-based model (FLUX-Schnell), including both qualitative results and a commitment to include quantitative evaluation (via RIEBench) in the camera-ready version. While it would have been stronger to include more diverse DiT models and quantitative results in the current submission, the effort to test on a substantially different architecture is appreciated and provides partial evidence of generalizability.
- Missing analysis of convolutional components:
The authors conducted new ablation experiments on ResNet blocks within SDXL-Turbo, showing that several convolutional layers have a substantial causal influence on the generated image. Although interpretability analysis using SAEs was not extended to these layers, the added results help paint a more complete picture of the U-Net architecture and directly address my concern.
- Clarification of generalization claims and intervention methodology:
The rebuttal includes clear explanations of what is meant by “generalization” (semantic meaning, causal influence, and reconstruction quality), and provides a step-by-step description of the SAE intervention process during multi-step generation. These clarifications significantly improve the methodological transparency of the work.
- Title and scope alignment:
The authors clarified the rationale behind the title “One Step is Enough” and acknowledged potential misunderstandings. They plan to revise the introduction to better communicate the intended message.

Overall, the authors have addressed my main concerns to a reasonable extent. Some aspects—particularly broader quantitative evaluation on DiT models and analysis of convolutional layers using SAEs—remain limited, which is why I maintain a borderline rating. However, I believe this paper makes a meaningful contribution and opens up promising directions for future research on interpretability in generative models.

**Limitations:**

"No" in the main paper.
--
- The analysis is limited to transformer blocks within the U-Net architecture, leaving out other critical components such as convolutional blocks, which may have distinct representational characteristics and contribute meaningfully to the generation process.
- Although the authors claim that the learned features generalize to multi-step generation and even to different models (e.g., SDXL Base, Flux-schnell), these claims are primarily supported by qualitative results, with limited quantitative evaluation or systematic comparison across architectures.
- The study is conducted on a relatively small number of U-Net blocks and features (e.g., four transformer blocks), raising the question of whether the observed interpretability and causal effects extend uniformly across the entire network.

**Quality:**

2

**Strengths And Weaknesses:**

# Strengths

### First application of SAEs to diffusion models:
This is the first work to apply sparse autoencoders (SAEs) for interpretability in text-to-image diffusion models, filling a significant gap in the mechanistic understanding of such models, which has previously focused mainly on language models.

### Causal and interpretable feature discovery:
The SAE-learned features are shown to be semantically meaningful and causally affect the image generation process. Manipulating individual features results in consistent and interpretable changes in the generated images.

### Feature-based image editing:
The authors demonstrate that SAE features can be used as an effective image editing interface during generation, by activating or suppressing specific features to achieve desired visual modifications.

### Block-level functional specialization:
The analysis reveals that different transformer blocks in SDXL Turbo (e.g., down.2.1, up.0.0, up.0.1) specialize in different roles, such as layout composition, object detail, or style. This insight contributes to a finer understanding of the generation pipeline.

# Weakness

### Limited baseline & Insufficient evaluation on Diffusion Transformers:
The study focuses primarily on SDXL Turbo, which, while compact and accessible, is not representative of the latest advancements in text-to-image generation. As newer models based on Diffusion Transformers—such as PixArt-series [1,2], SANA-series [3,4], or Stable Diffusion 3 [5]—have demonstrated strong performance, it would be valuable to assess whether the proposed SAE-based interpretability framework generalizes to these architectures. Although the authors include some qualitative examples on FLUX Schnell, the absence of quantitative results (e.g., explained variance, feature transport success) on these models limits the strength of the generalization claims. A more thorough evaluation on modern DiT-based models would significantly improve the impact and applicability of the approach. Including quantitative evaluations on one or more recent DiT-based models would significantly enhance the generality and relevance of the proposed approach, and help demonstrate whether the observed interpretability patterns hold across different model families.

### Missing analysis of convolutional components in SDXL-Turbo:
The paper focuses its analysis exclusively on transformer blocks within the U-Net of SDXL Turbo. However, the architecture also contains convolutional blocks, which play a crucial role in feature extraction and image synthesis. Understanding whether these convolutional components exhibit similarly interpretable or causally manipulable features is essential for a complete interpretability analysis of the model. As it stands, the findings may offer only a partial view of the internal mechanisms underlying image generation. Extending the SAE-based analysis to convolutional blocks would provide a more comprehensive understanding of the U-Net architecture and help reveal whether interpretability techniques can uncover complementary or distinct types of visual concepts encoded in different layer types.

# Minor point
The expression “One Step is Enough” in the title may not accurately reflect the core contributions of the paper. It could give the impression that the work is focused on accelerating diffusion models or improving generation efficiency, which is not the main focus. Instead, the paper is centered on interpretability using sparse autoencoders. A more precise title would help align reader expectations with the actual scope of the study.



---
[1] Chen et al. PixArt-α: Fast Training of Diffusion Transformer for Photorealistic Text-to-Image Synthesis. ICLR 2024.
[2] Chen et al. PixArt-Σ: Weak-to-Strong Training of Diffusion Transformer for 4K Text-to-Image Generation. ECCV 2024.
[3] Xie et al. SANA: Efficient High-Resolution Image Synthesis with Linear Diffusion Transformer. ICLR 2025.
[4] Xie et al. SANA 1.5: Efficient Scaling of Training-Time and Inference-Time Compute in Linear Diffusion Transformer. ICML 2025.
[5] Esser et al. Scaling Rectified Flow Transformers for High-Resolution Image Synthesis. ICML 2024.

---

> ### Author Rebuttal · Authors · 2025-07-31
>
> We sincerely thank the reviewer for their thorough evaluation and insightful feedback on our work. We are encouraged that the reviewer found our work to be the "first application of SAEs to diffusion models," filling a "significant gap," and appreciated the discovery of "causal and interpretable features" and the analysis of "block-level functional specialization."
>
> ## Shared response
> Please find our response addressing shared concerns in our **response to reviewer SVh8**.
>
> ## Individual concerns
> Below, we address each of the raised concerns (prefixed with “**W**” when raised in weaknesses and with “**Q**” when raised in questions) individually. Responses are prefixed with “**R**”.
>
> **W.1.** Limited baseline & Insufficient evaluation on Diffusion Transformers
>
> **R.** We agree that a more thorough evaluation of our method on DiT architectures would significantly strengthen the paper. To this end we will include an evaluation of FLUX on our RIEBench benchmark into the camera-ready version of our paper (see shared response).
>
> **W.2.** Missing analysis of convolutional components in SDXL-Turbo
>
> **R.** The reviewer raises an excellent point regarding the role of other architectural components like ResNet blocks. Our initial focus was on the transformer blocks, given their established role in handling high-level semantic and compositional reasoning in generative models.
>
> However, to directly address the reviewer's comment, we ran an additional ablation study where we measured the causal impact of each ResNet block on the final image, similar to the analysis we performed for the attention blocks in our paper. The impact is quantified using the LPIPS distance between the original image and the image generated with a specific block ablated. The results are presented below (R stands for resnet and A for attention blocks):
>
> |Type|Block|Score|
> |-|-|-|
> |R|d0.r0|0.733|
> |R|d0.r1|0.415|
> |A|d1.a0|0.184|
> |A|d1.a1|0.160|
> |R|d1.r0|0.457|
> |R|d1.r1|0.305|
> |A|d2.a0|0.193|
> |A|d2.a1|0.512|
> |R|d2.r0|0.522|
> |R|d2.r1|0.269|
> |A|m.a0|0.345|
> |R|m.r0|0.239|
> |R|m.r1|0.131|
> |A|u0.a0|0.394|
> |A|u0.a1|0.520|
> |A|u0.a2|0.280|
> |R|u0.r0|0.347|
> |R|u0.r1|0.170|
> |R|u0.r2|0.156|
> |A|u1.a0|0.252|
> |A|u1.a1|0.217|
> |A|u1.a2|0.254|
> |R|u1.r0|0.198|
> |R|u1.r1|0.167|
> |R|u1.r2|0.201|
> |R|u2.r0|0.274|
> |R|u2.r1|0.702|
> |R|u2.r2|0.851|
>
>
> This new analysis confirms that several ResNet blocks indeed have a substantial causal effect on the final output. Specifically, blocks such as down_blocks.0.resnets.1, down_blocks.1.resnets.0, and down_blocks.2.resnets.0. exhibit an impact comparable to or even greater than the most significant attention blocks we studied. There are resnet layers at the beginning and end of the unet that have an even larger LPIPS score but those destroy the entire image (ablating them results in pure noise), we think they encode/decode specifically to the noise and won’t yield interpretable features.
>
> While we find it highly interesting and agree that decomposing features from these ResNet blocks is a valuable research direction, we believe it is out of the scope of the current work.
>
> **W.3.** Minor point The expression “One Step is Enough” in the title may not accurately reflect the core contributions of the paper.
>
> **R.** The idea behind our title is to highlight a key and, we believe, surprising finding of our work: SAE features learned exclusively from the intermediate states of a one-step distilled model generalize remarkably well. They retain their semantic meaning and causal influence when applied in a multi-step generation process with the larger, non-distilled base models. We consider this a crucial result because it demonstrates that meaningful, interpretable features can be extracted with significantly lower computational resources (by training on the distilled model) and then effectively deployed to understand and edit more powerful, multi-step models. We will clarify this motivation in the introduction to better align reader expectations from the outset.
>
>
> **Q.1.** What about diffusion transformers?
>
> **R.** This is an important question that touches upon the generality of our findings. As mentioned in our response R.1, we have taken steps to explore the applicability of our method to other architectures. We have successfully trained SAEs on FLUX-Schnell, which is based on the Diffusion Transformer (DiT) architecture. Our results show that our method effectively learns interpretable and causally effective features in this different architectural paradigm as well. In our revised manuscript, we will present both qualitative examples and new quantitative results from our RIEBench benchmark on FLUX-Schnell to demonstrate that the approach is not limited to U-Net-based models.
>
>
> **Q.2.** What semantic and structural properties are captured by the features and what insights were gained?
>
> **R.** We thank the reviewer for this question and take this opportunity to clarify two central insights of our work.
> First, as detailed in Section 4.2 of our paper, our analysis reveals a strong functional specialization among different transformer blocks in SDXL. For example:
> - The down.2.1 block handles high-level composition and object creation. Interventions on this layer can generate entire objects corresponding to prompt concepts and have a global effect on the image.
> - The up.0.1 block specializes in style and texture. Its features can apply localized effects like animal textures, specific colors, or illumination without altering the underlying object structure.
> - The up.0.0 block acts as a detail-oriented layer. Its features are highly context-dependent and can be used to add or remove specific semantic properties of existing objects, such as removing a bow tie from a cat (as shown in our figures).
> - For FLUX-Schnell, we observed that the learned features tend to be more spatially localized, which is likely attributable to the higher spatial resolution of its latent space. We invite the reviewer to explore these behaviors interactively in our supplementary demo application.
> Second, a key finding is the remarkable stability of these semantic features across different noise levels. Once a feature is activated during the denoising process, it tends to remain active. This stability is precisely what enables our SAEs, trained on one-step SDXL Turbo, to be transferred directly to the multi-step SDXL Base model without retraining, suggesting a shared semantic subspace between the distilled and base models.
>
>
> **Q.3.** The caption for Figure 1 is somewhat minimal.
>
> **R.** Thank you for pointing this out. We will revise the caption for Figure 1 and the corresponding methodology section to provide a more comprehensive explanation. The numbers below the images (e.g., #2301) are the indices of the specific features being activated from our learned SAE dictionary, which contains thousands of feature vectors. For the FLUX-Schnell example, the SAE was indeed trained on the intermediate features from its one-step generation process. The intervention was then performed during a multi-step generation to demonstrate the generalization of the feature's causal effect. We will make this process explicit in the revised manuscript.
>
>
> **Q.4.** Could the authors clarify what specifically is being generalized?
>
> **R.** The reviewer is correct in their interpretation. When we refer to "generalization," we are referring to the preservation of all three aspects when moving from the one-step distilled model (where the SAEs were trained) to the multi-step base model:
> - Semantic Meaning: The features correspond to the same visual concepts (e.g., a specific style, object part, or texture) in both settings.
> - Causal Influence: Activating a specific feature produces a consistent and predictable change in the final image in both the one-step and multi-step generation processes.
> - Reconstruction Quality: For SAEs that achieve good reconstruction of the hidden state in the one-step model, they continue to be effective in the multi-step context.
> This finding is intuitive, as the distilled model is initialized from and trained to mimic the base model, suggesting they share a significant portion of their representational structure. We will clarify this definition of "generalization" in the paper.
>
> **Q.5.** Missing analysis of convolutional components.
>
> **R.** See the response to W.2.
>
> **Q.6.** The description of the generalization study lacks sufficient detail.
>
> **R.** We apologize for the lack of clarity and thank the reviewer for highlighting it. As discussed in our responses R.6 and R.7, the process is as follows:
> Training: An SAE is trained on a specific layer's hidden state activations, collected from numerous prompts using the one-step distilled model (e.g., SDXL Turbo).
> Intervention: During a multi-step generation process with the base model (e.g., SDXL Base over 20 steps), we intervene at a chosen denoising step. The intervention consists of adding a scaled version of a specific SAE feature vector (e.g., the 2301st feature from the dictionary) to the model's hidden state at the corresponding layer before it is passed to the next block.
> We recognize this process is central to our claims and will revise the methodology section to include this detailed, step-by-step description to ensure it is perfectly clear and reproducible.
>
> **Regarding missing limitations**
>
> **R.** We thank the reviewer for this comment. We wish to clarify that a limitations section was indeed included in our original submission. As per NeurIPS guidelines, it is located within the checklist section, which appears after the references. We acknowledge that this placement can be easily overlooked. The key limitations discussed there include the points raised by the reviewer, namely the focus on transformer blocks (addressed in R.2) and the need for more quantitative evaluation on DiT models (addressed in R.1). We will ensure the limitations are more prominently signposted in the camera-ready version.

---

> > ### Comment · Reviewer_fSpm · 2025-08-06
> > **Official Comment by Reviewer**
> >
> > Thank you for your effort and additional experiments. Since the main concerns are resolved clearly, I would keep my initial rating.

---

> > > ### Author Response · Authors · 2025-08-07
> > > **Request for clarification**
> > >
> > > Thank you for your feedback on our revised submission. We appreciate your acknowledgment that "the main concerns are resolved clearly" following our additional experiments and revisions.
> > >
> > > However, we are puzzled by your statement that you would "keep your initial rating" despite confirming that the main concerns have been clearly resolved. Could you help us understand the reasoning behind maintaining the same score when the primary issues you raised have been addressed?
> > >
> > > We would like to understand how to make our paper more acceptable.

---

> > > > ### Comment · Reviewer_fSpm · 2025-08-09
> > > > **Comment by Reviewer**
> > > >
> > > > After reviewing the authors’ rebuttal, I acknowledge that my main concern has been adequately addressed. However, considering the proposed methodology, overall completeness, and the potential impact on the community, I do not believe the paper warrants a score of 5 in the NeurIPS main track. I therefore maintain my original score of 4 (Borderline Accept).

---

> ### Comment · Area_Chair_vDBX · 2025-08-04
> **post-rebuttal comments**
>
> Dear reviewer fSpm,
>
> As you have seen already, the authors have responded to the questions in your initial review. Can you please share your thoughts post rebuttal, once you've had a chance to see the author responses and also the other reviews?
>
> Best,
> AC

---

### Official Review · Reviewer_SVh8 · 2025-07-03

**Clarity:** 3
**Significance:** 3
**Originality:** 3
**Rating:** 5
**Confidence:** 3

**Summary:**

Inspired by the application of sparse autoencoders in interpreting large language models, this work applied it to text-to-image diffusion models by training sparse autoencoders over the cross-attention blocks of one-step residual updates. This work found that training on one-step diffusion is able to explain most of the variance even in 4-step and 20-step diffusion. By using a SAM segmentation model, this work also showed that they can rank the importance of different found features and steer such features to control the generation process.

**Questions:**

1. For Figure 1 can you show the effect of changing the steering coefficient smoothly from negative to positive? Would the steering effect change smoothly as well?
2. What's the benefit of training the sparse autoencoder on 1-step updates? If for most experiments you used 4-step updates, why not directly train it on 4-step updates? Is it an efficiency argument? (but I assume you can use 1/4 of the data to train on 4-step updates with the same cost?)

**Ethical Concerns:**

["NO or VERY MINOR ethics concerns only"]

**Final Justification:**

Thanks authors for their rebuttal. I think all my concerns have been answered, including the stability of the steering effect, the generalization ability, and also what Figure 1 means. Therefore, I'm recommending the acceptance of this paper and increasing my score to 5.

**Limitations:**

Yes

**Paper Formatting Concerns:**

No issues found.

**Quality:**

3

**Strengths And Weaknesses:**

Strengths:
1. Applying sparse autoencoders to text-to-image diffusion models is well-motivated to interpret features.
2. Experiments show interesting specialization among blocks.
3. The learned features can be used to steer the generation process.

Weaknesses:
1. It's not clear how stable the steering effect is. For example, if multiple samples were taken, would all show the same steering effect? Also, if the steering coefficient changes smoothly (from a large negative value to a large positive value), would the steering effect also change smoothly?
2. It's not clear how well this method generalizes as experiments are mostly done on SDXL-Turbo.
3. The results on Flux-schnell are not clear. Why does Figure 1 show "the generalizability of our approach to recent Diffusion-Transformers" (Line 146)? Why didn't you apply the same steering interventions as those applied to SDXL in each column (or did I misunderstand something here)?

---

> ### Author Rebuttal · Authors · 2025-07-31
>
> We thank the reviewer for their thoughtful evaluation of our work and for recognizing the motivation behind applying sparse autoencoders to text-to-image diffusion models. We appreciate the acknowledgment of our experimental findings showing interesting specialization among blocks and the demonstrated ability to use learned features for steering generation. We proceed by a shared response followed by addressing the individual concerns of the reviewer.
>
> ## Shared response
>
> **Improved writing** We significantly improved their manuscript's organization and clarity through several key updates:
>
> - **Enhanced Structure & Contributions**
>     - Added clear paragraph headings for their four main contributions: SDLens (caching library), SAEPaint (visualization demo), RIEBench (editing benchmark), and findings on generalization across steps/architectures
>     - Updated abstract and section headers to match the new organization
>
> - **Content Consolidation & Clarification**
>     - **Section 3**: Merged spatial activation and sparse feature maps into a single "feature interventions" paragraph; added details on SDXL base interventions and classifier-free guidance handling
>     - **Section 4**: Renamed subsection 4.1 to better reflect the RIEBench benchmark; fixed a bug in equation (9); expanded details on benchmark construction and edit categories
>
> - **New Content & References**
>     - Added a new **Appendix B** specifically covering SDXL Base implementation details, including optimal strategies for feature addition during conditional/unconditional forward passes
>     - Systematically added appendix references throughout the paper to provide readers with additional technical details and examples
>     - Enhanced cross-referencing between sections and appendices (with updated lettering after inserting the new appendix)
>
> **Quantitative results for FLUX** In addition, to address the common concern of overly focusing our paper on SDXL Turbo, we performed an evaluation of FLUX Schnell on our representation-based editing benchmark (RIEBench). We will add these results to the camera-ready version of our paper in case of acceptance. For ease of presentation in the rebuttal format, we averaged across tasks here. We provide mean and Gaussian 95% confidence intervals of LPIPS between our edited-image and original-image CLIP similarity of our edited-image and the edit-prompt averaged over our 9 edit categories of which each contains about 30-50 examples.
>
> | Method | LPIPS Original | CLIP Text Edited |
> |--------|----------------|------------------|
> | sae_5_2 | 0.4436 ± 0.0272 | 0.2549 ± 0.0046 |
> | sae_5_1.5 | 0.3856 ± 0.0254 | 0.2534 ± 0.0046 |
> | sae_5_1.0 | 0.3085 ± 0.0225 | 0.2496 ± 0.0044 |
> | steering_0.4 | 0.4309 ± 0.0252 | 0.2475 ± 0.0041 |
> | steering_0.3 | 0.3712 ± 0.0217 | 0.2444 ± 0.0042 |
> | sae_5_0.5 | 0.1934 ± 0.0180 | 0.2433 ± 0.0043 |
> | steering_0.5 | 0.4686 ± 0.0269 | 0.2431 ± 0.0046 |
> | steering_1.0 | 0.5545 ± 0.0284 | 0.2243 ± 0.0052 |
> | baseline | 0.0000 ± 0.0000 | 0.2398 ± 0.0043 |
>
> As can be seen, our FLUX SAE (hyperparameters: k=20, expansion factor=4) interventions, in which we transport 5 features from the mask within the edit-image into the mask of the original-image with varying strength, dominate the activation steering baseline (higher CLIP similarity with edit-prompt, while achieving lower perturbation of the original-image in terms of LPIPS distance). For reference we also provide a baseline in which we compute LPIPS between original image and itself (= 0) and CLIP similarity between the unmodified original-image and the edit-image.
>
> **Qualitative results and demo** It seems like reviewers did not find many of our provided qualitative results in our supplementary material as well as our installation-free demo in Appendix A. We highly recommend checking out the demo application if time permits it.
>
> ## Individual concerns
>
> Below, we address each of the raised concerns (prefixed with “**W**” when raised in weaknesses and with “**Q**” when raised in questions) individually. Responses are prefixed with “**R**”.
>
> **Q.1 & W.1** It's not clear how stable the steering effect is. For example, if multiple samples were taken, would all show the same steering effect? Also, if the steering coefficient changes smoothly (from a large negative value to a large positive value), would the steering effect also change smoothly?
>
> **R.** We demonstrate that the steering effect is both stable and smooth across multiple dimensions. Regarding smoothness, we provide comprehensive evidence in Figures 7 and 8 (App. B) for Flux and Figure 35 (App. G) for SDXL, which show smooth transitions as steering coefficients vary continuously from large negative to large positive values. Additionally, our demo application (App. A) allows interactive exploration of this smoothness property.
>
> For stability across multiple samples, Figure 36 (App. G) provides clear evidence that the steering effects are consistent across different generations. Furthermore, Appendix F presents a detailed case study demonstrating that empty-prompt interventions produce consistent image changes, reinforcing the reliability of our approach. We encourage the reviewer to explore our demo application, which provides additional interactive evidence of both the stability and smoothness of steering effects.
>
>
> **Q.2** What's the benefit of training the sparse autoencoder on 1-step updates? If for most experiments you used 4-step updates, why not directly train it on 4-step updates? Is it an efficiency argument? (but I assume you can use 1/4 of the data to train on 4-step updates with the same cost?)
>
> **R.** Training on one-step updates provides multiple advantages beyond simple efficiency considerations. While computational efficiency is indeed a primary benefit, the deeper insight lies in the nature of feature activation patterns across timesteps.
> As demonstrated in Figure 3, features activated across different timesteps exhibit remarkable consistency, with the quality of this consistency reflected in the explained variance metrics. Using 4-step generations would oversample correlated activations and repeat similar features/samples, effectively reducing the diversity of training samples within a fixed computational budget. This redundancy would not improve feature quality while increasing computational cost.
>
> More fundamentally, our key insight is the successful decoupling of feature extraction methodology from feature injection during denoising. The surprising and practically valuable finding is that features extracted from 1-step activations effectively generalize to multi-step generation (4-steps for SDXL-Turbo, and even 20-25 steps for SDXL base). This demonstrates that our simpler training scheme is sufficient for extracting interpretable features, significantly simplifying the workflow for practitioners who need to train SAEs on diffusion models.
>
> This finding challenges the intuitive assumption that training and inference should use matching step counts, instead revealing that the underlying feature representations are robust across different sampling strategies.
>
>
> **W.2** It's not clear how well this method generalizes as experiments are mostly done on SDXL-Turbo.
>
> **R.** Our work demonstrates generalizability across multiple model architectures and configurations. Specifically, we show that SDXL-Turbo features, without any additional training, successfully generalize to SDXL base models (as evidenced in Figures 1, 2, 3, and 36). This cross-model transferability is particularly noteworthy as it demonstrates the robustness of our learned representations.
> More importantly, we replicate our SAE training methodology on FLUX, a state-of-the-art diffusion transformer-based text-to-image model, achieving successful results (Figures 1, 7, 8). This represents a significant architectural leap from the U-Net based SDXL models to transformer-based architectures. At the time of our initial submission and arXiv publication, we were the first work to train SAEs on intermediate results of the visual backbone in diffusion models. Furthermore, at the time of this NeurIPS submission, we were the first to achieve successful FLUX SAEs with effective feature steering capabilities.
>
>
> **W.3** The results on Flux-schnell are not clear. Why does Figure 1 show "the generalizability of our approach to recent Diffusion-Transformers" (Line 146)? Why didn't you apply the same steering interventions as those applied to SDXL in each column (or did I misunderstand something here)?
>
> **R.** We applied the same SAE training methodology to Flux-schnell activations, but the learned features naturally differ between architectures, and one-to-one feature matching is not always present. In some cases, achieving specific concepts requires combining multiple features (for example, creating a Santa Claus hat involves two distinct learned features working in conjunction).
> Figure 1 serves as a qualitative showcase of representative features learned by our SAEs on Flux-schnell, demonstrating successful causal interventions through feature steering. While the specific features differ from those learned on SDXL-Turbo (as expected given architectural differences), we present features with similar semantic content (blue color, "evil" characteristics, cartoon style) to illustrate conceptual correspondence. In contrast, features extracted from SDXL-Turbo one-step training successfully transfer to both SDXL-Turbo 4-step and SDXL base 25-step generation.
>
> The generalizability of our approach refers to the effectiveness of training SAEs to extract interpretable features that enable causal interventions for activating high-level concepts across different architectures. This generalizability is non-trivial, particularly for Diffusion Transformer (DiT) architectures where the role of each layer and the causal relationship between directions in the representation space differ significantly from U-Net architectures.

---

> ### Comment · Area_Chair_vDBX · 2025-08-04
> **post-rebuttal comments**
>
> Dear reviewer SVh8,
>
> As you have seen already, the authors have responded to the questions in your initial review. Can you please share your thoughts post rebuttal, once you've had a chance to see the author responses and also the other reviews?
>
> Best,
> AC

---

### Note · Authors · 2025-08-14

We'd like to thank the AC and all four reviewers for their thoughtful and positive feedback on our submission. We are glad that there is a **consensus** among the reviewers on the contribution of our work, with reviewers SVh8 and fSpm awarded a **score of 4**, and reviewers siKW and uRaJ were convinced by our rebuttal and decided to raise their score. We especially appreciate the **constructive comments** provided by reviewer uRaJ.

Although we did not receive any further response from reviewer SVh8, given that reviewer SVh8 confirmed acknowledgement of our rebuttal, we interpret this as being satisfied or at least having no further doubts about our revisions.

---

### Decision · Program_Chairs · 2025-09-17

**Decision:**

Accept (poster)

**Comment:**

This paper focuses on the lack of interpretability of standard text-to-image diffusion models. Inspired by the use of sparse autoencoders (SAEs) in the context of LLMs to decompose intermediate representations sparse sums of interpretable features, this work proposes to train SAEs on the residual updates of transformer blocks in SDXL Turbo. This is first shown on the SDXL Turbo's one-step generation process. Without additional training, this 1-step trained model also applies to the 4-step and the 20-step cases.

This work has several strengths: good motivation for applying SAEs to text-to-image diffusion models, causal and interpretable features, generalization of 1-step to 4-step and 20-step model variants, feature-based image editing. After the reviewer-author and the reviewers-AC discussions, all the reviewers confirmed that their initial questions were clearly addressed by the authors. The main remaining question was on the organization and clarity of the paper. This can be addressed in a final revision of the paper.

The strengths, especially addressing an important problem of interpretability of large generative models such as text-to-image diffusion models, clearly outweigh the other minor negative points on organization and clarity of the paper. In view of this, the AC recommends the paper for acceptance. Addressing the remaining issue and including all the clarifications provided in the rebuttal and discussion phases will make for a more complete paper.